# Implicit Bias and Loss of Plasticity in Matrix Completion: Depth Promotes Low-Rankness

**Baekrok Shin, Chulhee Yun**
Kim Jaechul Graduate School of AI, KAIST
{br.shin,chulhee.yun}@kaist.ac.kr

## Abstract

We study matrix completion via deep matrix factorization (a.k.a. deep linear neural networks) as a simplified testbed to examine how network depth influences training dynamics. Despite the simplicity and importance of the problem, prior theory largely focuses on shallow (depth-2) models and does not fully explain the implicit low-rank bias observed in deeper networks. We identify *coupled dynamics* as a key mechanism behind this bias and show that it intensifies with increasing depth. Focusing on gradient flow under block-diagonal observations, we prove: (a) networks of depth $\geq 3$ exhibit coupling unless initialized diagonally, and (b) convergence to rank-1 occurs if and only if the dynamics is coupled—resolving an open question by Menon (2024) for a family of initializations. We also revisit the *loss of plasticity* phenomenon in matrix completion (Kleinman et al., 2024), where pre-training on few observations and resuming with more degrades performance. We show that deep models avoid plasticity loss due to their low-rank bias, whereas depth-2 networks pre-trained under decoupled dynamics fail to converge to low-rank, even when resumed training (with additional data) satisfies the coupling condition—shedding light on the mechanism behind this phenomenon.

## 1 Introduction

Overparameterized neural networks have the capacity to perfectly memorize the training data, even when they are given random labels (Zhang et al., 2017). Despite their large capacity, neural networks often generalize well to unseen data without any explicit regularization techniques, which challenges conventional statistical wisdom. Recent studies attribute this phenomenon to the implicit bias of neural networks, arguing that among the many possible global minima, first-order algorithms such as (stochastic) gradient descent favor solutions that generalize well (Neyshabur et al., 2014; 2017; Huh et al., 2021; Timor et al., 2023; Frei et al., 2023; Kou et al., 2023; Galanti et al., 2024; Jacot, 2022).

Matrix completion, a task with practical applications in areas like recommender systems and image restoration, provides a key framework for investigating these implicit biases, particularly the tendency towards low-rank solutions. While matrix completion can be viewed as a special case of the broader matrix sensing framework (Jin et al., 2023; Soltanolkotabi et al., 2023; Ma & Fattahi, 2023; Stöger & Soltanolkotabi, 2021; Li et al., 2018), which offers general tools for understanding recovery from limited data, specific challenges can emerge when applying these general theories directly. Notably, common theoretical assumptions prevalent in matrix sensing analyses, such as the Restricted Isometry Property (RIP) (Candes & Tao, 2005), often prove too stringent or may not adequately capture the nuances of many practical matrix completion tasks. For instance, even when completing the $2 \times 2$ matrix $M_C$ (introduced in Figure 1a), which can successfully converge to a low-rank solution, the RIP condition cannot be satisfied. Therefore, researchers have investigated implicit bias phenomena specifically within matrix completion, without assuming the RIP condition (Menon, 2024; Bai et al., 2024; Razin & Cohen, 2020; Ma & Fattahi, 2024; Kim & Chung, 2023).

The goal of the matrix completion task is to recover a low-rank ground truth matrix $W^*$ using only a subset of its entries. A common strategy for matrix completion involves matrix factorization, which can also be viewed as linear neural networks. These networks reparameterize the target matrix $X$ as a product of factor matrices, $X = W_L W_{L-1} \cdots W_1$, and optimize the factors $\{W_l\}_{l \in [L]}$ by

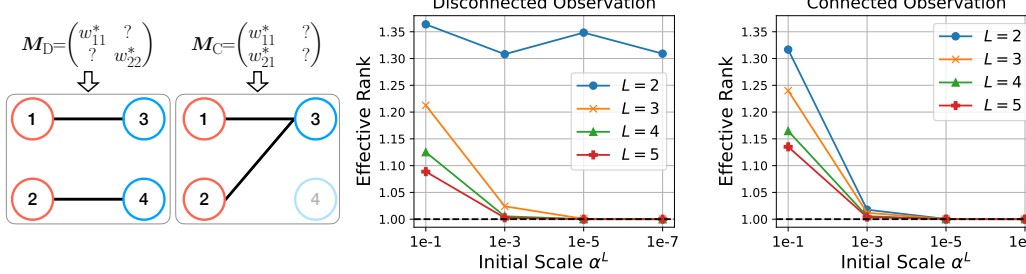

(a) Bipartite graph of $M_\text{D}$ & $M_\text{C}$     (b) Effective rank trained w/ $M_\text{D}$     (c) Effective rank trained w/ $M_\text{C}$

Figure 1: (**a**) Examples of bipartite graphs corresponding to observation patterns of $M_\text{D}$ (disconnected) and $M_\text{C}$ (connected). (**b-c**) Training results showing effective rank (*cf.* Roy & Vetterli (2007)) for completing rank-1 matrices $M_\text{D}$ and $M_\text{C}$, respectively. The rank-1 ground truth matrices were generated via $\boldsymbol{uv}^\top$, where $\boldsymbol{u}, \boldsymbol{v} \in \mathbb{R}^2$ with entries sampled i.i.d. from a standard normal distribution. We initialized each layer's entries by sampling from a Gaussian distribution with mean zero and standard deviation $\alpha$, chosen to ensure the initial scale of the product matrix $\boldsymbol{W}_{L:1}(0)$ is approximately invariant to depth $L$. Each result shows an average of 300 independent random trials.

minimizing the mean squared error over the observed entries using gradient descent. The observed entries constitute the training set, while the unobserved entries act as the test set.

The problem of predicting $\boldsymbol{W}^*$ is underdetermined, as infinitely many completions are possible. Nevertheless, both theory and experiments indicate that training even a simple two-layer factorization ($L = 2$) with gradient descent, without explicit rank constraints, typically yields a low-rank solution under reasonable assumptions (Razin & Cohen, 2020; Bai et al., 2024; Ma & Fattahi, 2024).

A recent work by Bai et al. (2024) formalizes this phenomenon using the concept of *data connectivity*. They demonstrate that if the observed entries form a connected bipartite graph (meaning any observed entry can be reached from any other via shared rows or columns), a depth-2 factorization initialized at an infinitesimally small scale converges to a low-rank solution. Conversely, the network may converge to a higher-rank matrix if the observations are disconnected (see Definition 1 and Figure 1a).

However, the situation changes significantly for deeper ($L \geq 3$) networks, as empirically demonstrated in Figure 1. Consider the task of completing the $2 \times 2$ matrix

$$\boldsymbol{M}_\text{D} = \begin{pmatrix} w_{11}^* & ? \\ ? & w_{22}^* \end{pmatrix} \tag{1}$$

where only the diagonal entries are observed. This observation pattern forms a disconnected graph as illustrated in Figure 1a. Consistent with the theory for disconnected graphs, $L = 2$ models fail to find a low-rank solution, empirically converging to rank-2 regardless of initialization scale. In contrast, deeper models ($L \geq 3$) with small initialization tend to converge to a rank-1 solution, as shown in Figure 1b. This specific example highlights that the implicit low-rank bias appears to be strengthened by depth, in a way that *cannot be explained solely by the data connectivity framework* developed for $L = 2$ models. Furthermore, considering connected cases as well, Figure 1c demonstrates that this strong low-rank bias is generally robust, tending to strengthen further as depth increases.

However, a theoretical understanding of this depth-induced bias remains elusive, largely due to the complex, coupled dynamics during training. While Arora et al. (2019) offer insights, their claim that the gap between two arbitrary singular values widens with depth is not fully formal. It stems largely from their analysis assuming stabilized singular vectors, which limits its scope. Indeed, Menon (2024) notes that even for a simple case like (1) with $w_{11}^* = w_{22}^* = 1$, proving that gradient descent with a deep factorization converges to a low-rank solution is still an open problem. Motivated by this gap in understanding, we theoretically analyze such settings, including the example (1).

Investigating the implicit low-rank bias in matrix completion can also shed light on the phenomenon of "*loss of plasticity*", a challenge widely observed in general neural network training (Shin et al., 2024; Ash & Adams, 2020; Achille et al., 2018; Berariu et al., 2021). The term loss of plasticity describes the tendency of neural networks, particularly after initial training, to lose their adaptability

to new information, hindering their generalization capabilities. A recent work by Kleinman et al. (2024) empirically reports this phenomenon even in matrix completion. They observe that models trained with insufficient data often yield high-rank solutions. If these models then warm-start using augmented data, they frequently struggle to achieve low-rank solutions. To provide a theoretical explanation for why this loss of plasticity occurs, this paper elucidates the phenomenon.

To summarize, here are the main research questions that we address throughout the paper:

- *What is the fundamental difference between deep ($L \geq 3$) and shallow ($L = 2$) factorizations regarding their implicit low-rank bias, particularly for disconnected observations?*

- *Can we theoretically establish that deeper models (i.e., with larger $L \geq 3$) exhibit a stronger implicit bias toward low-rank solutions?*

- *What is the underlying cause of the loss of plasticity phenomenon, and how does depth interplay with it?*

In Section 3.1, we begin by examining the depth-2 case to elucidate the key mechanism of connectivity. We find that *coupled training dynamics* induces a low-rank bias, a phenomenon generalizable to deeper networks. Section 3.2 further investigates this for all $L \geq 2$ using the block-diagonal observation case. Our analysis reveals that, for deep models, this bias distinctively promotes low-rank solutions compared to depth-2 models, strengthening with depth. Finally, Section 4 explores the loss of plasticity phenomenon in matrix completion. We observe that deep models typically avoid this phenomenon due to their low-rank bias. In contrast, we empirically observe and prove that depth-2 networks pre-trained with limited observations (yielding decoupled dynamics) and subsequently trained with augmented observations (yielding coupled dynamics) fail to find a low-rank solution. Please refer to Appendix A for further discussion of related work.

## 2 PROBLEM SETTING

We consider the problem of estimating a ground truth matrix $\boldsymbol{W}^* \in \mathbb{R}^{d \times d}$ based on observations of its entries $\{w_{ij}^*\}_{(i,j) \in \Omega}$, where $\Omega \subseteq [d] \times [d]$ is the set of observed indices. We model the estimate as a linear network $\boldsymbol{W}_{L:1} \triangleq \boldsymbol{W}_L \boldsymbol{W}_{L-1} \cdots \boldsymbol{W}_1$, where $\boldsymbol{W}_l \in \mathbb{R}^{d_l \times d_{l-1}}$ with $d_0 = d_L = d$. We denote the $(i,j)$-th entry of the matrix $\boldsymbol{W}_{L:1}$ as $w_{ij}$. The factor matrices $\{\boldsymbol{W}_l\}_{l=1}^L$ are trained by minimizing an objective function $\phi$, defined as the mean squared error $\ell$ over the observed entries in $\Omega$:

$$\phi(\boldsymbol{W}_1, \ldots, \boldsymbol{W}_L; \Omega) \triangleq \ell(\boldsymbol{W}_{L:1}; \Omega) = \frac{1}{2} \sum_{(i,j) \in \Omega} \left(w_{ij} - w_{ij}^*\right)^2. \tag{2}$$

We study the overparameterized regime where the intermediate dimensions satisfy $d_l \geq d$ for all $l \in [L-1]$, imposing no explicit rank constraints on the product model $\boldsymbol{W}_{L:1}$. Consistent with prior works, our analysis focuses on *gradient flow* dynamics (gradient descent with an infinitesimal step size) for a given objective function $\phi$. The dynamics for each layer $\boldsymbol{W}_l(t)$ evolve according to:

$$\dot{\boldsymbol{W}}_l(t) \triangleq \frac{d}{dt} \boldsymbol{W}_l(t) = -\frac{\partial}{\partial \boldsymbol{W}_l} \phi(\boldsymbol{W}_1(t), \boldsymbol{W}_2(t), \ldots, \boldsymbol{W}_L(t); \Omega), \quad l \in [L], \ t \geq 0. \tag{3}$$

For depth-2 networks ($L = 2$), the product of factor matrices $\boldsymbol{A} \in \mathbb{R}^{d \times d_1}$ (representing $\boldsymbol{W}_2$) and $\boldsymbol{B} \in \mathbb{R}^{d_1 \times d}$ (representing $\boldsymbol{W}_1$), we denote $\boldsymbol{W}_{\boldsymbol{A},\boldsymbol{B}} \triangleq \boldsymbol{A}\boldsymbol{B}$. We define the stable rank of a matrix $\boldsymbol{W}$ as $\mathrm{srank}(\boldsymbol{W}) \triangleq \|\boldsymbol{W}\|_F^2 / \|\boldsymbol{W}\|_2^2$. We also denote by $\boldsymbol{I}_d$ the $d \times d$ identity matrix and by $\boldsymbol{J}_d$ the $d \times d$ all-ones matrix.

Bai et al. (2024) introduce the concept of data connectivity for an incomplete matrix $\boldsymbol{M}$. Connectivity is characterized by its set of observed indices $\Omega \subseteq [d] \times [d]$ and the corresponding observation matrix $\boldsymbol{P}$ (where $P_{ij} = 1$ if $(i,j) \in \Omega$, and 0 otherwise). The formal definition is as follows:

**Definition 1** (Connectivity from Bai et al. (2024)). *An incomplete matrix $\boldsymbol{M}$ is **connected** if the bipartite graph $\mathcal{G}_{\boldsymbol{M}}$, constructed from its observation matrix $\boldsymbol{P}$ using the adjacency matrix $\begin{bmatrix} \boldsymbol{0} & \boldsymbol{P}^\top \\ \boldsymbol{P} & \boldsymbol{0} \end{bmatrix}$, is connected after removing isolated vertices. Otherwise, $\boldsymbol{M}$ is **disconnected**.*

## 3 IMPLICIT BIAS OF DEPTH INDUCED BY COUPLED TRAINING DYNAMICS

In this section, we extend the connectivity argument of Bai et al. (2024) to general depth factorizations. We first demonstrate how the *coupling of training dynamics* serves as the key mechanism explaining data connectivity's role in depth-2 models, through the completion of two previously introduced $2 \times 2$ matrices, $M_D$ and $M_C$, as illustrative examples. Building on the insights derived from these depth-2 model analyses, we hypothesize that deep networks exhibit an intrinsic low-rank bias because they maintain a high degree of coupled training dynamics, irrespective of observation patterns. This hypothesis is further corroborated by the block-diagonal observation results presented in Section 3.2.

### 3.1 WARM-UP: COUPLED DYNAMICS VS. DECOUPLED DYNAMICS IN DEPTH-2 NETWORKS

We focus on the simple $2 \times 2$ matrix completion of $M_D$ and $M_C$, using depth-2 models $W_{A,B}(t) = A(t)B(t)$. For brevity, let $a_i(t) \in \mathbb{R}^{d_1}$ be the transpose of the $i$-th row of $A(t)$, and let $b_j(t) \in \mathbb{R}^{d_1}$ be the $j$-th column of $B(t)$. Our aim is to see how training dynamics affect the *alignment* of the rows of $A(t)$ or the columns of $B(t)$, as such alignment leads to a rank-1 product matrix $W_{A,B}(t)$.

**Decoupled Dynamics.** In the $M_D$ case (disconnected observations $w_{11}^*, w_{22}^*$), the gradient flow using the objective defined in (2), results in independent dynamics for the pairs $(a_1, b_1)$ and $(a_2, b_2)$:

$$\dot{a}_i(t) = \left(w_{ii}^* - a_i(t)^\top b_i(t)\right) b_i(t), \quad \dot{b}_i(t) = \left(w_{ii}^* - a_i(t)^\top b_i(t)\right) a_i(t) \quad \text{for } i = 1, 2.$$

Note that while the dynamics couple $a_1(t)$ with $b_1(t)$ and $a_2(t)$ with $b_2(t)$ within each pair, the two pairs $(a_1, b_1)$ and $(a_2, b_2)$ are decoupled from each other. This decoupling means the overall system's dynamics separate into two independent systems. Consequently, there is no compelling reason to align vectors from different pairs, typically leading to high-rank solutions with generic initializations (Figure 1b). Indeed, we can obtain closed-form solutions solely dependent on initialization (see Proposition 4.1). For instance, with $A(0) = B(0) = \alpha I_2$, we have $W_{A,B}(\infty) = \text{diag}(w_{11}^*, w_{22}^*)$.

**Coupled Dynamics.** In contrast, for the $M_C$ case (connected observations $w_{11}^*, w_{21}^*$), the gradient flow on the objective (2) yields coupled dynamics that do not decompose into independent pairs:

$$\dot{a}_1(t) = \left(w_{11}^* - a_1(t)^\top b_1(t)\right) b_1(t), \quad \dot{a}_2(t) = \left(w_{21}^* - a_2(t)^\top b_1(t)\right) b_1(t),$$
$$\dot{b}_1(t) = \left(w_{11}^* - a_1(t)^\top b_1(t)\right) a_1(t) + \left(w_{21}^* - a_2(t)^\top b_1(t)\right) a_2(t). \tag{4}$$

An important observation from (4) is that $A(0) = 0$ ensures rank-1 $W_{A,B}(t)$ due to persistent alignment of $a_1(t), a_2(t)$ and $b_1(t)$. Although non-zero initialization leads to more complex behavior arising from coupled training dynamics, the following theorem demonstrates that sufficiently small initial norms in $A(0)$ also result in the alignment of $a_1(t)$ and $a_2(t)$ with $b_1(t)$.

**Theorem 3.1.** *For the product model $W_{A,B}(t) = A(t)B(t) \in \mathbb{R}^{2 \times 2}$, we consider the gradient flow dynamics (4), where the observations are $w_{11}^*(\neq 0)$ and $w_{21}^*(\neq 0)$. We assume convergence to the zero-loss solution (i.e., $w_{11}(\infty) = w_{11}^*, w_{21}(\infty) = w_{21}^*$). Defining $u^* = \frac{b_1(\infty)}{\|b_1(\infty)\|_2}$ and the orthogonal component $a_{i\perp}(\infty) = a_i(\infty) - (a_i(\infty)^\top u^*)u^*$, we have:*

$$\frac{\|a_{i\perp}(\infty)\|_2^2}{\|a_i(\infty)\|_2^2} \leq \frac{\|A(0)\|_F^2 \left(\sqrt{\|b_1(0)\|_2^4 + 4w_{11}^{*2} + 4w_{21}^{*2}} + \|b_1(0)\|_2^2\right)}{2w_{i1}^{*2}}, \quad \text{for } i = 1, 2.$$

The theorem shows that small initial norms for $A(0)$ lead to the alignment of $a_1(\infty)$ and $a_2(\infty)$ with $b_1(\infty)$, implying a near rank-1 product matrix $W_{A,B}(\infty)$. This suggests that for depth-2 networks, coupled training dynamics (resulting from connected observations) facilitate the emergence of low-rank solutions under such small initialization, in contrast to the decoupled dynamics of disconnected observations, where no such bias exists regardless of initialization scale. This connection between observation connectivity and the coupling of training dynamics in depth-2 models motivates our investigation into how coupled dynamics manifest and induce low-rank bias in deeper networks, irrespective of connectivity patterns, as explored in the subsequent sections.

**Remark.** Analyzing these dynamics is challenging because the time evolutions of $a_1, a_2$, and $b_1$ are mutually dependent. We note that Theorem 3.1 is not a direct corollary of Theorem 3 in Bai et al. (2024). We explicitly characterize the degree of misalignment as a function of the initialization scale, unlike their assumption of an infinitesimal initialization scale with additional conditions.

## 3.2 Coupled Dynamics in Deep Networks Induce Implicit Bias Towards Low Rank

Section 3.1 illustrated the importance of coupled training dynamics, driven by data connectivity, for achieving low-rank solutions in simple two-layer factorizations ($L = 2$). Building on this understanding, we now extend our analysis to deep networks ($L \geq 3$). For illustrative purposes, consider a depth-3 network $\boldsymbol{W}_{3:1}$. An arbitrary observed entry $w_{ij}$ from this matrix is given by:

$$w_{ij} = \sum_{k=1}^{d_2} \sum_{l=1}^{d_1} (\boldsymbol{W}_3)_{ik}(\boldsymbol{W}_2)_{kl}(\boldsymbol{W}_1)_{lj}. \tag{5}$$

Crucially, because all elements of the intermediate matrix $\boldsymbol{W}_2$ contribute to the computation of $w_{ij}$ regardless of $(i, j)$, gradients of different observed entries will propagate through and update these shared elements in $\boldsymbol{W}_2$. This inherently couples their training dynamics, a structural feature distinct from the depth-2 case, where coupling is primarily determined by the observation pattern. Such inherent coupling, in turn, implies a potential intrinsic bias towards low-rank solutions for deep models. To formalize this notion, we introduce the following definition of coupled dynamics.

**Definition 2** (Coupled/Decoupled Dynamics). *Consider the matrix completion setup with the model $\boldsymbol{W}_{L:1}(t) = \boldsymbol{W}_L(t) \cdots \boldsymbol{W}_1(t) \in \mathbb{R}^{d \times d}$. Let $\boldsymbol{\theta}(t)$ be the vector of all trainable parameters evolving according to the gradient flow dynamics (defined in (3)). The gradient flow dynamics are **decoupled** if there exists a partition of $\Omega$ into non-empty, disjoint subsets $\Omega_1, \ldots, \Omega_K$ ($K \geq 2$) such that $\bigcup_{k=1}^K \Omega_k = \Omega$ and the following condition holds for any $(i, j) \in \Omega_k$ and $(p, q) \in \Omega_l$ with $k \neq l$:*

$$\langle \nabla_{\boldsymbol{\theta}} w_{ij}(t), \nabla_{\boldsymbol{\theta}} w_{pq}(t) \rangle = 0, \quad \forall t \geq 0. \tag{6}$$

*The gradient flow dynamics are **coupled** if they are not decoupled.*

While Bai et al. (2024) introduce similar terminology in Definition A.5, their definition is restricted to depth-2 networks. We extend this notion to networks of arbitrary depth. For depth-2 matrices, it is straightforward to verify that coupled and decoupled dynamics typically correspond to connected and disconnected graphs, respectively, based on Definitions 1 and 2. For depth $\geq 3$ matrices, any initialization with an absolutely continuous distribution (e.g., Gaussian, uniform) yields gradient flow dynamics that are coupled with probability one, irrespective of the observation pattern (see Proposition B.1 in Appendix B). However, special cases exist where training dynamics are decoupled even for $L \geq 3$. Refer to Appendix B for further discussion.

### 3.2.1 Implicit Bias of Depth Under Block-Diagonal Observations

To gain deeper theoretical insight into *how coupled dynamics induce low-rank bias as depth increases*, we study the block-diagonal observation setting. Specifically, we consider a ground-truth matrix $\boldsymbol{W}^* \in \mathbb{R}^{d \times d}$ with the block-diagonal observation set

$$\Omega_{\text{block}}^{(s,n)} \triangleq \bigcup_{b \in [n]} \{(i, j) \mid i, j \in \{(b-1)s + 1, \ldots, bs\}\},$$

where $s, n \in \mathbb{N}$ satisfy $d = sn$. Here, $s$ denotes the block size and $n$ the number of blocks. We consider $\boldsymbol{W}^*$ with positive and identical observations $w^* \triangleq w_{ij}^* > 0$ for $(i, j) \in \Omega_{\text{block}}^{(s,n)}$. In this setting, the observed entries are confined to disjoint square blocks along the diagonal, forming a disconnected observation pattern. Note that this formulation recovers the diagonal observation setting as the special case $s = 1$, and therefore strictly generalizes the diagonal case. As highlighted in the $2 \times 2$ example with $s = 1$ (*cf.* Figure 1b), this setting reveals a stark difference between shallow and deep networks despite the lack of connectivity.

We consider a depth-$L$ factorization of the model, $\boldsymbol{W}_{L:1}(t) = \boldsymbol{W}_L(t)\boldsymbol{W}_{L-1}(t) \cdots \boldsymbol{W}_1(t)$ where $\boldsymbol{W}_l \in \mathbb{R}^{d \times d}$ for all $l \in [L]$. To investigate how dynamic coupling affects the low-rank bias, we consider a family of initializations where, for parameters $\alpha > 0$ and $m > 1$, each factor matrix $\boldsymbol{W}_l(0)$ is initialized as follows:

$$\boldsymbol{W}_l(0) = \begin{pmatrix} \alpha & \alpha/m & \cdots & \alpha/m \\ \alpha/m & \alpha & \cdots & \alpha/m \\ \vdots & \vdots & \ddots & \vdots \\ \alpha/m & \alpha/m & \cdots & \alpha \end{pmatrix} \in \mathbb{R}^{d \times d}, \quad \forall l \in [L]. \tag{7}$$

**Remark.** Random Gaussian initialization allows coupling but introduces $Ld^2$ degrees of freedom, making it almost impossible to analyze individual training trajectories. For this reason, prior work often adopts deterministic initializations such as $\alpha \boldsymbol{I}_d$ (Gunasekar et al., 2017; Arora et al., 2019). We follow this approach but adopt a more general deterministic family that is adequate for establishing our theoretical claims. Our initialization interpolates between $\alpha \boldsymbol{J}_d$ (as $m \to 1$) and $\alpha \boldsymbol{I}_d$ (as $m \to \infty$), and the parameter $m$ allows direct control over the initial numerical rank.

Using this initialization scheme with diagonal observations, the following proposition specifies how parameters $m$ and network depth $L$ determine if training dynamics are coupled or decoupled:

**Proposition 3.2.** *Consider a depth-$L$ model, where each factor $\boldsymbol{W}_l(0) \in \mathbb{R}^{d \times d}$ is initialized with (7) trained with $\Omega_{\text{block}}^{(s,n)}$. Then, by Definition 2, the following hold for any $s \geq 1$ and $n \geq 2$:*

- *For depth $L = 2$, the training dynamics are **decoupled** for all $m > 1$.*

- *For depth $L \geq 3$:*
    - *The training dynamics are **coupled** if $1 < m < \infty$.*
    - *The training dynamics are **decoupled** if $m = \infty$ (i.e., initialization with $\alpha \boldsymbol{I}_d$).*

By Proposition D.1 in Appendix D.4, the loss converges to zero under the gradient flow dynamics (3). Building on this result, our objective is to determine the rank of solutions found by gradient flow depending on the coupling of dynamics. The theorem below presents an equation of each singular value of the converged matrix $\boldsymbol{W}_{L:1}(\infty)$, for all $L \geq 2$.

**Theorem 3.3.** *Consider the product matrix $\boldsymbol{W}_{L:1}$ whose factor matrices $\boldsymbol{W}_l \in \mathbb{R}^{d \times d}$ are initialized according to (7). Under the gradient flow dynamics (3), we have $\ell(\boldsymbol{W}_{L:1}(\infty); \Omega_{\text{block}}^{(s,n)}) = 0$ (Proposition D.1, Appendix D.4). For all parameters $\alpha > 0, m > 1, n \geq 2, s \geq 1$, and $L \geq 2$, the singular values $\sigma_1 \geq \sigma_2 \geq \cdots \geq \sigma_d$ of $\boldsymbol{W}_{L:1}(\infty)$ satisfy $\sigma_j = 0$ for all $j > n$. The principal singular value $\sigma_1$ and the secondary singular values $\sigma_i$ (for $i \in \{2, \ldots, n\}$) are determined as follows:*

*- If $L = 2$ (**decoupled dynamics**): The singular values are given in closed form by*

$$\sigma_1 = \frac{w^* d(m+d-1)^2}{(m+d-1)^2 + (n-1)(m-1)^2}, \quad \sigma_i = \frac{w^* d(m-1)^2}{(m+d-1)^2 + (n-1)(m-1)^2}.$$

*- If $L \geq 3$ and $1 < m < \infty$ (**coupled dynamics**): The singular values satisfy the implicit equations:*

$$\sigma_1^{\frac{2-L}{L}} - \left(\frac{w^* d - \sigma_1}{n-1}\right)^{\frac{2-L}{L}} = C_{\alpha, m, L, d}, \tag{8}$$

$$(w^* d - (n-1)\sigma_i)^{\frac{2-L}{L}} - \sigma_i^{\frac{2-L}{L}} = C_{\alpha, m, L, d}, \tag{9}$$

*where $C_{\alpha, m, L, d} \triangleq \left(\frac{\alpha}{m}\right)^{2-L}\left((m+d-1)^{2-L} - (m-1)^{2-L}\right)$.*

*- If $L \geq 3$ and $m = \infty$ (**decoupled dynamics**): The singular values converge to:*

$$\sigma_1 = \sigma_i = sw^*.$$

The proof of the theorem is provided in Appendix D.3. The theorem details the converged singular values of $\boldsymbol{W}_{L:1}(\infty)$ for our initialization scheme (7). Crucially, it reveals distinct outcomes based on the nature of the training dynamics. For decoupled dynamics—specifically, when $L = 2$ (for sufficiently large $m > 1$), or when $L \geq 3$ and $m = \infty$—singular values from $\sigma_1$ to $\sigma_n$ approach $sw^*$ and are independent of the scale $\alpha$. This implies convergence to a rank-$n$ solution. In contrast, for coupled dynamics ($L \geq 3$ with finite $m$), the outcome becomes $\alpha$-dependent. To illustrate the implications of these implicit equations, we present the following corollary.

**Corollary 3.4.** *Let $1 < m < \infty$, $n \geq 2$, $s \geq 1$, $w^* > 0$, and $L \geq 3$ be fixed. Then, as $\alpha \to 0$, the stable rank of the limit product matrix $\boldsymbol{W}_{L:1}(\infty)$ converges to one; that is,*

$$\text{srank}\big(\boldsymbol{W}_{L:1}(\infty)\big) \to 1.$$

The proof of the corollary is provided in Appendix D.6. Note that, according to Theorem 3.3, the stable rank of the depth-2 network satisfies $\text{srank}\big(\boldsymbol{W}_{2:1}(\infty)\big) = \frac{(m+d-1)^4 + (m-1)^4(n-1)}{(m+d-1)^4}$, which is

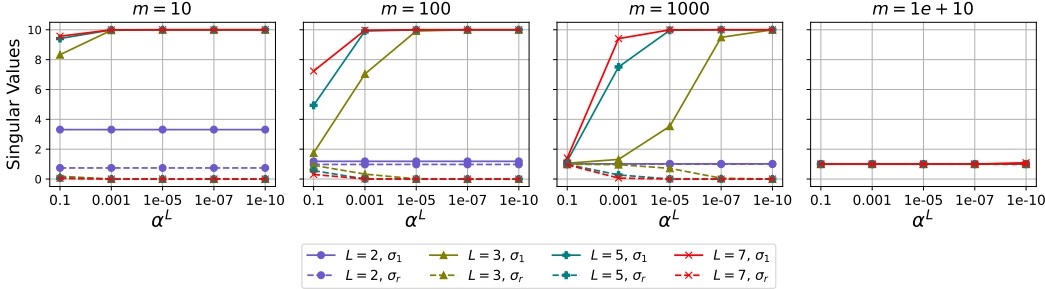

Figure 2: Singular values of $\boldsymbol{W}_{L:1}(\infty)$ (numerically obtained from Theorem 3.3) against initialization scale $\alpha^L$, for the diagonal observation task where $s = 1$. Solid lines represent the largest singular value $\sigma_1$; dashed lines denote the other (identical) singular values $\sigma_r$ for $r \geq 2$. For finite $m$, these results illustrate that both greater depth $L$ and a smaller initial scale $\alpha$ strengthen the low-rank bias, in contrast to the $L = 2$ case. Conversely, a very large $m$ ($m = 10^{10}$), approximating an $\alpha \boldsymbol{I}_d$ (rank-$d$) initialization, leads to decoupled dynamics and a full-rank solution, independent of both $L$ and $\alpha$.

independent of the initialization scale $\alpha$, and is approximately $n$ when $m$ is large. In contrast, for any depth $L \geq 3$ with finite $m$, Corollary 3.4 implies that as $\alpha \to 0$, then $\mathrm{srank}\big(\boldsymbol{W}_{L:1}(\infty)\big) \to 1$, so the depth-$L$ network converges to a nearly rank-1 solution.

**Remark.** Readers may wonder why, even under decoupled dynamics, the solution converges to a rank-$n$ rather than a rank-$d$ matrix. Although the dynamics are decoupled at the level of the full matrix (Definition 2), applying the same notion of coupling to each diagonal block separately reveals that the dynamics are coupled within each block. Since all rows (or columns) inside a block share the same observation pattern, the row (or column) space is spanned by at most $n$ block-wise patterns, yielding a rank at most $n$. In particular, when $s = 1$ (diagonal observations) and hence $n = d$, the solution becomes full rank (see Figure 2). This block-diagonal example thus highlights how coupled versus decoupled dynamics govern the strength of the low-rank bias.

We consider diagonal observations ($s = 1$) with $w^* = 1$ and $d = 10$, and numerically solve the implicit equations (8) and (9) to examine how the depth $L$ and initialization parameters $(\alpha, m)$ influence the singular value distribution. Note that both equations admit unique solutions (Proposition D.2 in Appendix D.5). To ensure a fair comparison across depths, we set the initialization scale so that the scale of the $\mathbf{W}_{L:1}(0)$ is comparable across depths; concretely, we match the scale of $\alpha^L$ across different values of $L$. The results in Figure 2 confirm that these coupled dynamics in models with $L \geq 3$ and finite $m$ indeed induce a low-rank bias, contrasting with the full-rank outcomes of the decoupled cases. Moreover, this bias becomes more pronounced as $L$ increases, evidenced by a wider gap between $\sigma_1$ and $\sigma_r$ for $r \geq 2$.

Additional numerical evidences are provided in Figures 5–8 (Appendix C.1). Moreover, Figure 9 in Appendix C.1 shows that these numerical results agree with the outcomes of a gradient descent with a sufficiently small learning rate. Moreover, although we do not provide a theoretical proof, we conduct experiments under noisy diagonal observations (Figure 11), non-equal diagonal observations (Figure 12), and with various optimizers including SGD, GD with momentum, Adam, RMSProp, and Adagrad (Figures 13–17). Across all settings, we consistently observe the emergence of a depth-induced low-rank bias. We further train practical neural networks to examine whether increased depth indeed leads to a low-rank bias. The results shown in Figures 18–21 (SGD with momentum), 22–25 (Adam), and 26–29 (RMSProp) in Appendix C.1.1 indicate that as depth increases (e.g., ResNet-18 to 101 and VGG-11 to 19), the average effective rank decreases, highlighting the emergence of low-rank bias in practical neural networks across these optimizers.

**Remark.** Our analysis of low-rank bias for a specific family of deterministic initializations resolves the challenging open problem (1) highlighted in Section 14.1 of Menon (2024). Figure 10 in Appendix C.1 further demonstrates that our proposed deterministic initialization exhibits qualitative trends similar to Gaussian initialization. We therefore argue that our results provide foundational insights into low-rank bias applicable to more general random initializations.

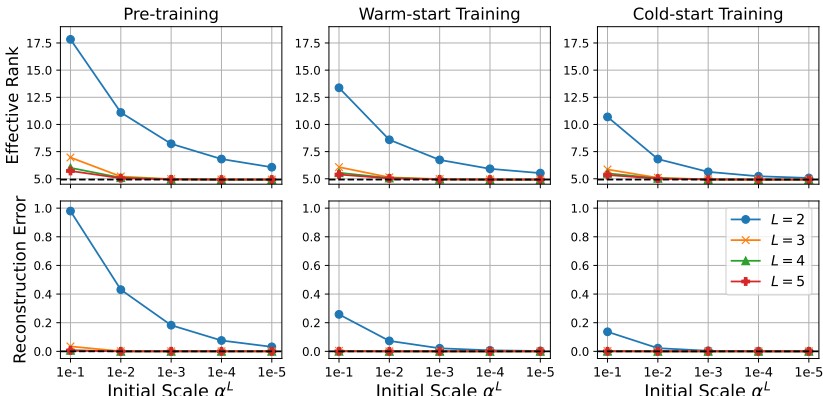

Figure 3: Experiments use a $100 \times 100$ rank-5 ground-truth matrix. Pre-training utilizes 2000 randomly sampled entries ($\Omega_{\mathrm{pre}}$; $|\Omega_{\mathrm{pre}}| = 2000$), while post-training adds 1000 more, forming $\Omega_{\mathrm{post}}$ ($\Omega_{\mathrm{pre}} \subset \Omega_{\mathrm{post}}$; $|\Omega_{\mathrm{post}}| = 3000$). The top row of panels displays effective rank, and the bottom row shows reconstruction error, both measured at convergence. The leftmost panels depict training on $\Omega_{\mathrm{pre}}$, and the rightmost on $\Omega_{\mathrm{post}}$, both starting from random Gaussian initialization. The middle panels show warm-start training on $\Omega_{\mathrm{post}}$, initialized from converged pre-trained models with $\Omega_{\mathrm{pre}}$.

## 4 UNDERSTANDING LOSS OF PLASTICITY IN DEPTH-2 MATRIX COMPLETION

Studying the inherent tendency towards low-rank solutions in matrix completion can offer further insights into the loss of plasticity phenomenon. Kleinman et al. (2024) report the emergence of this phenomenon in matrix completion: models pre-trained on limited observations struggle to adapt when training continues on augmented observations.

We conduct an experiment using a rank-5 ground-truth matrix $\boldsymbol{W}^* \in \mathbb{R}^{100 \times 100}$, where pre-training is performed on a sparse observation set and post-training continues with additional observations. In Figure 3, we compare two post-training strategies: *warm-start* training, initialized from the pre-trained model, and *cold-start* training, initialized from scratch on the augmented observations, across different depths in terms of effective rank and reconstruction error. We observe that, even when pre-trained on sparse observations, deeper models increasingly favor low-rank solutions as depth increases. This supports our argument (Section 3.2) that deeper networks inherently converge toward low-rank solutions even from limited and disconnected initial data. Consequently, further training on augmented data does not substantially increase the rank compared to training from scratch on the augmented observations. Based on our observations, we conclude that the low-rank bias of deep models helps them mitigate the loss of plasticity, while the phenomenon is more pronounced in depth-2 models. To theoretically understand the underlying cause of this phenomenon itself, we henceforth focus our analysis on depth-2 models.

In Section 4.1, we study pre-training on diagonal-only observations. We then consider post-training on $2 \times 2$ (Section 4.2) and $d \times d$ (Section 4.3) matrices. For the $2 \times 2$ case, we set $\Omega_{\mathrm{pre}}^{(2)} \triangleq \{(1,1), (2,2)\}$ and obtain the post-training set $\Omega_{\mathrm{post}}^{(2)}$ by adding a single off-diagonal entry to ensure connectivity, i.e. $\Omega_{\mathrm{post}}^{(2)} \triangleq \{(1,1), (1,2), (2,2)\}$. Likewise, for the $d \times d$ case, $\Omega_{\mathrm{pre}}^{(d)} \triangleq \{(i,i)\}_{i \in [d]}$, and $\Omega_{\mathrm{post}}^{(d)}$ is formed by adding additional (off-diagonal) observations; see Section 4.3 for details.

**Remark.** Kleinman et al. (2024) observe that loss of plasticity is further intensified with increasing network depth, a conclusion they reached by measuring a "relative reconstruction loss" when compared to models trained from scratch on the augmented dataset. In their setup, training is run for a fixed number of iterations without waiting for convergence, whereas in our experiments we terminate each training phase once the loss falls below a fixed threshold.

### 4.1 PRE-TRAINING WITH DIAGONAL OBSERVATIONS

To clearly observe loss of plasticity in a setting consistent with Section 3.2, we pre-train using only diagonal entries, yielding a disconnected pattern. We consider decoupled-to-coupled scenarios,

where additional data is introduced to induce coupled training dynamics. For depth-2 models, they correspond to a disconnected-to-connected observation pattern. For the pre-training, closed-form solutions that depend *solely* on the network's initialization can be found in the following proposition:

**Proposition 4.1.** *Consider a ground truth matrix $\boldsymbol{W}^* \in \mathbb{R}^{d \times d}$ with diagonal observations $\Omega_{\mathrm{pre}}^{(d)}$. The model is factorized as $\boldsymbol{W}_{\boldsymbol{A},\boldsymbol{B}}(t) = \boldsymbol{A}(t)\boldsymbol{B}(t)$, where $\boldsymbol{A}(t), \boldsymbol{B}(t) \in \mathbb{R}^{d \times d}$. For each observation $(i,i) \in \Omega_{\mathrm{pre}}^{(d)}$, define the constants $P_i$ and $Q_i$ based on the initial values:*

$$P_i \triangleq \sum_{k=1}^{d} a_{ik}(0)b_{ki}(0) \quad and \quad Q_i \triangleq \sum_{k=1}^{d} \left(a_{ik}(0)^2 + b_{ki}(0)^2\right).$$

*Furthermore, for each diagonal observation, let the parameter $\bar{r}_i$ be determined from the ground truth entry $w_{ii}^*$ and the constants defined above, $\bar{r}_i \triangleq \frac{1}{2} \log \left( \frac{P_i + \frac{Q_i}{2}}{w_{ii}^* + \sqrt{w_{ii}^{*2} - P_i^2 + \left(\frac{Q_i}{2}\right)^2}} \right)$. Then, assuming convergence to a zero-loss solution of the loss $\ell(\boldsymbol{W}_{\boldsymbol{A},\boldsymbol{B}}; \Omega_{\mathrm{pre}}^{(d)})$, any entry $a_{pq}(\infty)$ of the converged matrix $\boldsymbol{A}(\infty)$ and any entry $b_{pq}(\infty)$ of the converged matrix $\boldsymbol{B}(\infty)$ (for any $p, q \in [d]$) are given by:*

$$a_{pq}(\infty) = a_{pq}(0) \cosh(\bar{r}_p) - b_{qp}(0) \sinh(\bar{r}_p),$$
$$b_{pq}(\infty) = b_{pq}(0) \cosh(\bar{r}_q) - a_{qp}(0) \sinh(\bar{r}_q).$$

**Remark.** The proposition covers *arbitrary* initializations with *distinct* $w_{ii}^*$, which goes beyond Theorem 3.3 in the $s = 1, L = 2$ setting. While the above analysis focuses on (block-)diagonal observation cases, it can be generalized to any fully disconnected case (i.e., a single observation per row and column). This yields distinct solutions for various types of observation sets, as detailed in Appendix E.1.

We analyze the scenario where training resumes from a state obtained through pre-training. Let the pre-training phase conclude at a sufficiently large timestep $T_1$. For simplicity, we assume that the solution $\boldsymbol{W}_{\boldsymbol{A},\boldsymbol{B}}(T_1)$ has perfectly converged with respect to the pre-training objective, neglecting any residual error due to the finite duration of this phase. Our subsequent analysis demonstrates that, starting from $\boldsymbol{W}_{\boldsymbol{A},\boldsymbol{B}}(T_1)$, the model $\boldsymbol{W}_{\boldsymbol{A},\boldsymbol{B}}(t)$ cannot converge to a low-rank solution.

## 4.2 POST-TRAINING: $2 \times 2$ MATRIX EXAMPLE

We aim to analyze scenarios where training is resumed under coupled dynamics, building upon solutions obtained from an initial decoupled pre-training phase (Proposition 4.1). To this end, we first define the specific pre-training setup for an illustrative $2 \times 2$ case: We observe diagonal entries ($\Omega_{\mathrm{pre}}^{(2)}$), which are identical and positive, i.e., $w^* \triangleq w_{11}^* = w_{22}^* > 0$. To make loss of plasticity particularly pronounced during the pre-training, we initialize the model with $\alpha \boldsymbol{I}_2$ (for $\alpha > 0$), which is the $m = \infty$ setting of our initialization scheme in (7). Then, from Proposition 4.1, it follows that:

$$\boldsymbol{A}(T_1) = \boldsymbol{B}(T_1) = \begin{pmatrix} \sqrt{w^*} & 0 \\ 0 & \sqrt{w^*} \end{pmatrix}. \tag{10}$$

For the subsequent post-training phase, an additional off-diagonal observation is introduced to establish connectivity. Without loss of generality, we assume $w_{12}^* > 0$ is revealed, while the diagonal entries $w_{11}^*$ and $w_{22}^*$ from the pre-training phase remain observed. Thus, the updated set of observed entries becomes $\Omega_{\mathrm{post}}^{(2)} = \{(1,1), (1,2), (2,2)\}$. The ground-truth matrix is assumed to be rank-1, ensuring the setting is non-trivial, and the task is thus to predict the remaining entry $w_{21}^* = w^{*2}/w_{12}^* > 0$. The following theorem, however, reveals a contrasting outcome for this entry.

**Theorem 4.2.** *Let $\boldsymbol{A}(T_1), \boldsymbol{B}(T_1)$ be the factor matrices obtained from the pre-training phase, as specified by (10). Then, running gradient flow during the subsequent post-training phase (for $t \geq T_1$), starting from $\boldsymbol{A}(T_1)$ and $\boldsymbol{B}(T_1)$, results in exponential decay of the loss:*

$$\ell(\boldsymbol{W}_{\boldsymbol{A},\boldsymbol{B}}(t); \Omega_{\mathrm{post}}^{(2)}) \leq \frac{1}{2} w_{12}^{*2} e^{-2w^*(t-T_1)}.$$

*Consequently, a lower bound for the stable rank of the converged matrix $\boldsymbol{W}_{\boldsymbol{A},\boldsymbol{B}}(\infty)$ is given by:*

$$\mathrm{srank}\left(\boldsymbol{W}_{\boldsymbol{A},\boldsymbol{B}}(\infty)\right) \geq 1 + \exp\left(-8\frac{w_{12}^*}{w^*}\right).$$

*Furthermore, for all $t > T_1$, $w_{21}(t)$ of the evolving matrix $\boldsymbol{W}_{\boldsymbol{A},\boldsymbol{B}}(t)$ satisfies $w_{21}(t) < 0$.*

The theorem indicates that the loss decreases exponentially fast, particularly when starting from large-norm solutions (at a rate governed by $w^*$). Therefore, since the model converged to high-rank solutions during pre-training, its singular values remain largely unchanged from this initial state, as long as $w_{12}^*$ has a small magnitude compared to $w^*$. Furthermore, the unobserved entry $w_{21}(t)$ converges to a negative value, which contradicts the positive $w_{21}^*$ expected for the true rank-1 solution.

### 4.3 POST-TRAINING: $d \times d$ MATRIX UNDER LAZY TRAINING REGIME

We attribute Theorem 4.2 primarily to the model's "*lazy training*" (Chizat et al., 2019) as large-norm initializations lead to faster loss decay, causing the model to converge to a nearby global minimum that may not be a low-rank solution. Drawing on this concept, we extend the preceding analysis of loss of plasticity to the more general case of $d \times d$ ground-truth matrices. The following theorem states that when the model is initialized with a sufficiently small loss, resulting from warm-starting that perfectly fits all previously observed data, the model exhibits lazy training. This, in turn, prevents further learning that would reduce the rank and instead steers the model towards a nearby minimum.

**Theorem 4.3.** *For factor matrices $\boldsymbol{A}, \boldsymbol{B} \in \mathbb{R}^{d \times d}$, suppose $\boldsymbol{A}$ and $\boldsymbol{B}$ are balanced at $t = 0$, i.e., $\boldsymbol{A}(0)^\top \boldsymbol{A}(0) = \boldsymbol{B}(0)\boldsymbol{B}(0)^\top$. Let $f(\boldsymbol{A}, \boldsymbol{B})$ be the function that maps $(\boldsymbol{A}, \boldsymbol{B})$ to the vector of model predictions for a given set of observed entries $\Omega_{\text{post}}^{(d)}$. We then define $\sigma_{\max}$ and $\sigma_{\min}$ as the maximum and minimum singular values, respectively, of the Jacobian of the function $f$ evaluated at the pre-trained state (at $t = T_1$). If the loss at time $T_1$ satisfies $\ell\left(\boldsymbol{W}_{\boldsymbol{A},\boldsymbol{B}}(T_1); \Omega_{\text{post}}^{(d)}\right) \leq \frac{\sigma_{\min}^6}{1152d\sigma_{\max}^2}$, this results in exponential decay of the loss:*

$$\ell\left(\boldsymbol{W}_{\boldsymbol{A},\boldsymbol{B}}(t); \Omega_{\text{post}}^{(d)}\right) \leq \ell\left(\boldsymbol{W}_{\boldsymbol{A},\boldsymbol{B}}(T_1); \Omega_{\text{post}}^{(d)}\right) \exp\left(-\frac{1}{2}\sigma_{\min}^2(t - T_1)\right).$$

*Consequently, the stable rank of $\boldsymbol{A}(t)$ (which is equal to that of $\boldsymbol{B}(t)$) remains bounded below by*

$$\text{srank}\big(\boldsymbol{A}(t)\big) \geq \left(\frac{\|\boldsymbol{A}(T_1)\|_F - \frac{\sigma_{\min}}{4\sqrt{2d}}}{\|\boldsymbol{A}(T_1)\|_2 + \frac{\sigma_{\min}}{4\sqrt{2d}}}\right)^2.$$

The theorem states that if a model has little remaining to learn (achieved via pre-training), it undergoes lazy training regime. In this regime, the loss converges rapidly, while its stable rank remains largely unchanged from the initial state. Thus, once a model has converged to a high-rank state, it struggles to recover a low-rank structure even when new observations are introduced to form connectivity. The proof of Theorem 4.3 is provided in Appendix E.3.

**Example.** As an illustrative example, consider a rank-1 ground-truth matrix $\boldsymbol{W}^* \in \mathbb{R}^{d \times d}$,

$$\boldsymbol{W}^* = \begin{pmatrix} w^* & cw^* & \cdots & c^{d-1}w^* \\ c^{-1}w^* & w^* & \cdots & c^{d-2}w^* \\ \vdots & \vdots & \ddots & \vdots \\ c^{1-d}w^* & c^{2-d}w^* & \cdots & w^* \end{pmatrix}, \quad c = O\left(\frac{1}{d}\right).$$

We pre-train only on the identical diagonal observations $w^*$ using $\Omega_{\text{pre}}^{(d)}$, with initialization $\boldsymbol{A}(0) = \boldsymbol{B}(0) = \alpha\boldsymbol{I}_d$ up to time $T_1$ (see Proposition 4.1 for the pre-training solution). We then reveal the full upper-triangular set $\Omega_{\text{post}}^{(d)} = \{(i,j) : 1 \leq i \leq j \leq d\}$ to form connectivity and continue training. By Theorem 4.3, for every $t \geq T_1$, the stable rank of $\boldsymbol{A}(t)$ is uniformly lower-bounded by $\Omega(d)$:

$$\text{srank}\big(\boldsymbol{A}(t)\big) \geq \left(\frac{4d-1}{4\sqrt{d}+1}\right)^2.$$

## 5 CONCLUSION

We demonstrate that in matrix completion, deeper networks ($L \geq 3$) inherently exhibit a stronger low-rank bias than shallow networks, primarily due to their coupled training dynamics that manifest regardless of observation patterns. For tractable analysis, we consider gradient flow starting at a family of deterministic initializations, showing in the block-diagonal observation setting that depth amplifies the low-rank bias. Furthermore, our theoretical analysis of warm-starting scenarios details the loss of plasticity phenomenon, revealing how large-norm, high-rank initial states hinder convergence to low-rank solutions. We believe the theoretical results from matrix completion provide broader insight into how depth shapes implicit bias and explains the loss of plasticity in practical deep networks.

ETHICS STATEMENT

This work is purely theoretical and involves no human subjects, personal data, or new dataset collection. We foresee no safety, fairness, or privacy risks and confirm that we are in accordance with the ICLR Code of Ethics.

REPRODUCIBILITY STATEMENT

The proofs of all theorems and propositions in the main text appear in the corresponding appendices: Theorem 3.1 in Appendix D.1, Proposition 3.2 in Appendix D.2, Theorem 3.3 in Appendix D.3, Proposition 4.1 in Appendix E.1, and Theorems 4.2 and 4.3 in Appendices E.2 and E.3, respectively.

ACKNOWLEDGEMENT

This work was supported by two Institute of Information & communications Technology Planning & Evaluation (IITP) grants (No. RS-2022-II220184, Development and Study of AI Technologies to Inexpensively Conform to Evolving Policy on Ethics; No. RS-2019-II190075, Artificial Intelligence Graduate School Program (KAIST)) funded by the Korean government (MSIT). The work was also partly supported by a National Research Foundation of Korea (NRF) grant (No. RS-2024-00421203) funded by the Korean government (MSIT).

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

CONTENTS

## A  FURTHER RELATED WORKS

### A.1  IMPLICIT REGULARIZATION IN NEURAL NETWORKS

A substantial body of work investigates the *implicit regularization* of gradient-based training in overparameterized models (Gunasekar et al., 2017; Ji & Telgarsky, 2019a;b; Woodworth et al., 2020; Yun et al., 2021; Razin et al., 2021; Andriushchenko et al., 2023; Frei et al., 2023; Jung et al., 2025; Hui et al., 2025; Moon et al., 2026). For linearly separable classification trained with (S)GD, Soudry et al. (2018) show that gradient descent on the logistic loss converges in direction to the $\ell_2$ max-margin classifier. Building on this result, Nacson et al. (2019b) establish analogous directional convergence guarantees for SGD, and Nacson et al. (2019a) extend the theory to a broader family of loss functions. For homogeneous neural networks, gradient descent likewise exhibits directional convergence, and the limit direction coincides with a KKT point of an appropriate margin-maximization problem (Ji & Telgarsky, 2020; Lyu & Li, 2020).

For adaptive methods in linearly separable classification, Wang et al. (2022) analyze (S)GD with momentum and deterministic Adam and show that these methods also converge in direction to the max-margin solution. This analysis is further extended to homogeneous models by Wang et al. (2021). More recently, Zhang et al. (2024) demonstrate that when the stability constant is negligible, Adam exhibits a qualitatively different implicit bias and converges to the maximum $\ell_\infty$ margin rather than the $\ell_2$ max-margin direction selected by (S)GD. Along a related line, Cattaneo et al. (2024) use backward error analysis to study RMSProp and Adam and show that their implicit regularization depends sensitively on hyperparameters and the training stage. Closely related to our setting, Zhao (2022) examine matrix completion and show that Adam, when combined with an explicit spectral ratio penalty, induces a strong low-rank bias even in depth-1 linear networks. However, their analysis focuses on deriving the flow of Adam and does not characterize the limiting solution.

Several works investigate how depth promotes low-rank solutions (Arora et al., 2019; Gissin et al., 2020; Li et al., 2021; Huh et al., 2021; Jacot, 2022; Timor et al., 2023). Huh et al. (2021) provide empirical evidence that deeper networks (both linear and nonlinear) tend to find solutions with lower effective-rank embeddings. Complementing this, Timor et al. (2023) show theoretically that ReLU networks trained with squared loss exhibit a bias toward low-rank solutions under the assumption that gradient flow converges to the solution minimizing the $\ell_2$ norm.

Turning to deep linear networks, Gissin et al. (2020) and Li et al. (2021) study depth-induced bias as a function of initialization scale. They report that, as depth increases, the dependence on initialization can become weaker, and incremental learning can emerge. However, their analyses consider a matrix factorization task, which they frame as matrix completion with full observations. Therefore, in their setting, convergence to a low-rank solution is guaranteed if the model converges to zero-loss, which does not hold in our matrix completion task settings.

While Arora et al. (2019) investigate the matrix completion task in deep linear networks, offering insights from derived singular value dynamics, they cannot fully track these dynamics to prove low-rank convergence as network depth increases. Their analysis is primarily restricted to the regime where $t \geq t_0$, after which singular vectors are assumed to have stabilized. For $t \geq t_0$, they find that one singular value can be expressed as a function of another, involving a constant term that emerges from the state at $t_0$ (which can be the dominant component). Based on this derivation, they demonstrate that the gap between these singular values widens with increasing depth. In contrast, our Theorem 3.3, by precisely tracking the converged values of singular values, rigorously establishes their ultimate behavior and the resulting low-rank bias.

Closely related to our setting, Razin & Cohen (2020) study a depth $L \geq 2$ matrix completion problem in a $2 \times 2$ example with three observations (one diagonal and two off diagonal entries). Their Theorems 1 and 2 show that, as the loss converges, the effective rank converges to its infimum. However, their analysis does not distinguish between the depth $L = 2$ and $L \geq 3$ regimes, and therefore does not identify a depth dependent low-rank bias or an underlying mechanism that explains it. In addition, their guarantees are independent of the initialization scale, so they do not capture the empirically observed phenomenon that low-rank bias becomes stronger as the initialization scale decreases. In contrast, our results explicitly separate the $L = 2$ and $L \geq 3$ cases, characterize the limiting singular values, and show how depth and initialization scale jointly control the emergence of low rank solutions in matrix completion.

For depth-2 matrix completion tasks, Bai et al. (2024) introduce the connectivity argument. They prove that if the observations construct a connected bipartite graph, the model can converge to a low-rank solution when the initialization scale is infinitesimally small, subject to certain technical assumptions. Conversely, if the observations form a disconnected graph, the model generally cannot converge to a low-rank solution. However, a special case occurs if this disconnected graph is composed of complete bipartite components: here, the model converges to the minimum nuclear norm solution, again under specific technical assumptions. This characterization of implicit bias does not readily generalize to matrices with deeper matrices, as depicted in Figure 1.

## A.2 Loss of Plasticity

Loss of plasticity describes a widely observed phenomenon where a model's ability to adapt to new information diminishes over time (Achille et al., 2018; Ash & Adams, 2020; Dohare et al., 2021; Nikishin et al., 2022; Lee et al., 2024; Shin et al., 2024; Lee et al., 2025; Lyle et al., 2025; Springer et al., 2025; Kim et al., 2025; Han et al., 2026). The phenomenon is frequently observed in scenarios with gradually changing datasets, such as those encountered in reinforcement learning (Igl et al., 2020; Nikishin et al., 2022; Lyle et al., 2023) or continual learning (Dohare et al., 2021; Kumar et al., 2025; Chen et al., 2023; Park et al., 2025; Hernandez-Garcia et al., 2025; Rohani et al., 2025), where the model may struggle to adapt to new environments.

Although loss of plasticity is typically studied in non-stationary settings, a similar effect arises in stationary regimes where the dataset grows incrementally while the underlying distribution remains fixed (Ash & Adams, 2020; Berariu et al., 2021; Shin et al., 2024). In such cases, a model is first trained to convergence on an initial i.i.d. subset (e.g., a subset of CIFAR-10/100) and then warm-started for continued training on an expanded sample from the same distribution (e.g., the full CIFAR-10/100). Perhaps counterintuitively, these warm-started models often generalize worse, yielding lower test accuracy than models trained from scratch on the combined dataset.

While this phenomenon is problematic in many real-world applications where new data is continuously added, theoretical studies on it remain scarce. Shin et al. (2024), for instance, offer a theoretical explanation using an artificial framework. Within this framework, they demonstrate that such behavior occurs because warm-started models often complete training by memorizing data-dependent noise, which is not useful for generalization. However, the analytical framework they employ is considered artificial and limited in its ability to accurately characterize the optimization processes of typical deep learning models.

Recently, Kleinman et al. (2024) observed loss of plasticity in deep linear networks, identifying "critical learning periods": an initial phase of effective learning followed by a significantly reduced capacity to learn later (Achille et al., 2018; Vock & Meisel, 2025). They employ a matrix completion framework to further observe this behavior. When observations from matrix completion tasks are treated as training samples in neural network training, they observed that a model initially trained on a sparse set of observations and subsequently retrained (i.e., warm-started) on an expanded dataset typically exhibits a larger performance gap (in terms of reconstruction error) compared to a model trained from scratch on the entire expanded dataset. However, their work does not offer theoretical guarantees to account for these observations. Motivated by this, in Section 4, we attempt to explain this behavior within the specific context of depth-2 matrix completion settings.

## B   COUPLED AND DECOUPLED TRAINING DYNAMICS

This section introduces coupled and decoupled training dynamics (Definition 2) and illustrates them with concrete examples. Before that, we present Proposition B.1, which shows that for deep models ($L \geq 3$), generic (absolutely continuous) initialization yields coupled dynamics almost surely.

**Lemma B.1.** *Define $\boldsymbol{W}_{b:a} \triangleq \boldsymbol{W}_b \boldsymbol{W}_{b-1} \cdots \boldsymbol{W}_a$, and $\boldsymbol{W}_{a:b} \triangleq \boldsymbol{I}_d$ where $b \geq a$. For $w_{ij}(t) \triangleq \boldsymbol{e}_i^\top \boldsymbol{W}_{L:1}(t) \boldsymbol{e}_j$,*

$$\nabla_{\boldsymbol{W}_l} w_{ij}(t) = \left( \boldsymbol{W}_{L:l+1}(t)^\top \boldsymbol{e}_i \right) \left( \boldsymbol{W}_{l-1:1}(t) \boldsymbol{e}_j \right)^\top \in \mathbb{R}^{d \times d}.$$

*Hence, for any $(i,j)$ and $(p,q)$,*

$$\langle \nabla_{\boldsymbol{\theta}} w_{ij}(t), \nabla_{\boldsymbol{\theta}} w_{pq}(t) \rangle = \sum_{l=1}^{L} \left( \boldsymbol{e}_i^\top \boldsymbol{T}_l(t) \boldsymbol{e}_p \right) \left( \boldsymbol{e}_j^\top \boldsymbol{S}_l(t) \boldsymbol{e}_q \right),$$

*where $\boldsymbol{T}_l(t) \triangleq \boldsymbol{W}_{L:l+1}(t) \boldsymbol{W}_{L:l+1}(t)^\top$ and $\boldsymbol{S}_l(t) \triangleq \boldsymbol{W}_{l-1:1}(t)^\top \boldsymbol{W}_{l-1:1}(t)$ are symmetric positive semidefinite matrix.*

*Proof.* Define $\boldsymbol{a}_l^{(i)}(t) \triangleq \boldsymbol{W}_{L:l+1}(t)^\top \boldsymbol{e}_i$ and $\boldsymbol{b}_l^{(j)}(t) \triangleq \boldsymbol{W}_{l-1:1}(t) \boldsymbol{e}_j$. By

$$w_{ij}(t) = \boldsymbol{e}_i^\top \boldsymbol{W}_{L:l+1}(t) \boldsymbol{W}_l(t) \boldsymbol{W}_{l-1:1}(t) \boldsymbol{e}_j = \boldsymbol{a}_l^{(i)}(t)^\top \boldsymbol{W}_l(t) \boldsymbol{b}_l^{(j)}(t),$$

we have $\nabla_{\boldsymbol{W}_l} w_{ij}(t) = \boldsymbol{a}_l^{(i)}(t) \boldsymbol{b}_l^{(j)}(t)^\top$. Furthermore,

$$\begin{aligned}
\langle \nabla_{\boldsymbol{\theta}} w_{ij}(t), \nabla_{\boldsymbol{\theta}} w_{pq}(t) \rangle &= \sum_{l=1}^{L} \langle \nabla_{\boldsymbol{W}_l} w_{ij}(t), \nabla_{\boldsymbol{W}_l} w_{pq}(t) \rangle_F \\
&= \sum_{l=1}^{L} \left\langle \boldsymbol{a}_l^{(i)}(t) \boldsymbol{b}_l^{(j)}(t)^\top, \boldsymbol{a}_l^{(p)}(t) \boldsymbol{b}_l^{(q)}(t)^\top \right\rangle_F \\
&= \sum_{i=1}^{L} \left( \boldsymbol{a}_l^{(i)}(t)^\top \boldsymbol{a}_l^{(p)}(t) \right) \left( \boldsymbol{b}_l^{(j)}(t)^\top \boldsymbol{b}_l^{(q)}(t) \right) \\
&= \sum_{i=1}^{L} \left( \boldsymbol{e}_i^\top \boldsymbol{T}_l(t) \boldsymbol{e}_p \right) \left( \boldsymbol{e}_j^\top \boldsymbol{S}_l(t) \boldsymbol{e}_q \right),
\end{aligned}$$

which concludes the proof. □

**Proposition B.1.** *Let $L \geq 3$ and initialize $\{\boldsymbol{W}_l(0)\}_{l=1}^L$ with i.i.d. entries from any absolutely continuous distribution. For any observation set $\Omega \subseteq [d] \times [d]$ where $|\Omega| \geq 2$, with probability 1,*

$$\langle \nabla_{\boldsymbol{\theta}} w_{ij}(0), \nabla_{\boldsymbol{\theta}} w_{pq}(0) \rangle \neq 0$$

*holds for all distinct $(i,j), (p,q) \in \Omega$. Consequently, no nontrivial partition $\Omega = \bigcup_{k=1}^K \Omega_k$ with $K \geq 2$ can satisfy the decoupling condition (6) at $t = 0$. Hence, by Definition 2, the gradient flow dynamics are coupled with probability 1 irrespective of the observation pattern.*

*Proof.* By Lemma B.1, at $t = 0$ we have

$$\varphi_{ij,pq}(\boldsymbol{W}_1, \ldots, \boldsymbol{W}_L) \triangleq \langle \nabla_{\boldsymbol{\theta}} w_{ij}, \nabla_{\boldsymbol{\theta}} w_{pq} \rangle = \sum_{l=1}^{L} \left( \boldsymbol{e}_i^\top \boldsymbol{T}_l \boldsymbol{e}_p \right) \left( \boldsymbol{e}_j^\top \boldsymbol{S}_l \boldsymbol{e}_q \right),$$

which is a polynomial in the entries of $\{\boldsymbol{W}_l\}_{l=1}^L$. For any $(i,j) \neq (p,q)$, we now show that $\varphi_{ij,pq}$ is not the zero polynomial.

If $i = p$, the $l = L$ term reduces to $\boldsymbol{e}_j^\top \boldsymbol{S}_L \boldsymbol{e}_q$. By choosing $\boldsymbol{W}_{1:L}$ so that $\boldsymbol{S}_L$ has a nonzero $(j,q)$ entry, this term evaluates to a nonzero value; hence $\varphi_{ij,pq}$ is not identically zero. By symmetry, the same argument applies when $j = q$.

If $i \neq p$ and $j \neq q$, consider $l = 2$. Setting all other layers to $\boldsymbol{I}_d$, choose $\boldsymbol{W}_3$ so that $(\boldsymbol{e}_i^\top \boldsymbol{T}_2 \boldsymbol{e}_p) \neq 0$ and choose $\boldsymbol{W}_1$ so that $(\boldsymbol{e}_j^\top \boldsymbol{S}_2 \boldsymbol{e}_q) \neq 0$. Then $\varphi_{ij,pq} = (\boldsymbol{e}_i^\top \boldsymbol{T}_2 \boldsymbol{e}_p)(\boldsymbol{e}_j^\top \boldsymbol{S}_2 \boldsymbol{e}_q) \neq 0$. Consequently, in all cases $\varphi_{ij,pq}$ is not identically zero.

Since $\varphi_{ij,pq}$ is a nonzero polynomial in the entries of $\{\boldsymbol{W}_l\}_{l=1}^L$, its zero set $Z_{ij,pq} \triangleq \{(\boldsymbol{W}_1, \ldots, \boldsymbol{W}_L) : \varphi_{ij,pq}(\boldsymbol{W}_1, \ldots, \boldsymbol{W}_L) = 0\}$ is a proper algebraic set in $\mathbb{R}^{Ld^2}$ and hence has Lebesgue measure zero.

Let the initialization distribution of $(\boldsymbol{W}_1(0), \ldots, \boldsymbol{W}_L(0))$ be absolutely continuous with respect to Lebesgue measure. Then
$$\Pr\big[(\boldsymbol{W}_1(0), \ldots, \boldsymbol{W}_L(0)) \in Z_{ij,pq}\big] = 0,$$
so for this fixed pair $(i, j) \neq (p, q)$ we have $\varphi_{ij,pq}(\boldsymbol{W}_1(0), \ldots, \boldsymbol{W}_L(0)) \neq 0$ almost surely. There are only finitely many distinct pairs in $\Omega$. A finite union of measure-zero sets still has measure zero; hence, with probability one,
$$\varphi_{ij,pq} \neq 0 \quad \text{for all distinct } (i, j), (p, q) \in \Omega. \tag{11}$$

By Definition 2, a decomposition $\Omega = \bigcup_{k=1}^K \Omega_k$ ($K \geq 2$) yields decoupled dynamics only if
$$\langle \nabla_{\boldsymbol{\theta}} w_{ij}(t), \nabla_{\boldsymbol{\theta}} w_{pq}(t) \rangle = 0$$
for all $(i, j) \in \Omega_k$, $(p, q) \in \Omega_l$ with $k \neq l$ and for all $t \geq 0$.

However, this already fails at $t = 0$, since every cross-pair inner product is nonzero by (11). Thus, no such partition exists. Consequently, for $L \geq 3$ and any observation set $\Omega$, the gradient flow dynamics are *coupled almost surely* under any absolutely continuous initialization. $\square$

## B.1 COUPLED DYNAMICS EXAMPLE

### B.1.1 DEPTH-2 MODEL

For shallow ($L = 2$) matrices, coupled dynamics typically correspond to connected observations under generic initialization, in accordance with Definitions 1 and 2 (the specific case of initialization, such as zero matrices, which leads to decoupled dynamics, will be further detailed in a later subsection). We illustrate this principle with an example where the observed entries form the first column of a $2 \times 2$ matrix.

Consider a $2 \times 2$ matrix, denoted $\boldsymbol{M}_{\mathrm{C}}$, which is to be completed using its first column as observations:
$$\boldsymbol{M}_{\mathrm{C}} \triangleq \begin{bmatrix} w_{11}^* & ? \\ w_{21}^* & ? \end{bmatrix}.$$
The corresponding observation pattern matrix $\boldsymbol{P}_{\mathrm{C}}$ is:
$$\boldsymbol{P}_{\mathrm{C}} = \begin{bmatrix} 1 & 0 \\ 1 & 0 \end{bmatrix}.$$
The associated adjacency matrix $\mathcal{A}_{\mathrm{C}}$ for the bipartite graph is constructed as:
$$\mathcal{A}_{\mathrm{C}} = \begin{bmatrix} \boldsymbol{0}_{2,2} & \boldsymbol{P}_{\mathrm{C}}^\top \\ \boldsymbol{P}_{\mathrm{C}} & \boldsymbol{0}_{2,2} \end{bmatrix} = \begin{bmatrix} 0 & 0 & 1 & 1 \\ 0 & 0 & 0 & 0 \\ 1 & 0 & 0 & 0 \\ 1 & 0 & 0 & 0 \end{bmatrix},$$
which forms a connected graph as illustrated in Figure 1a. This setup leads to coupled training dynamics under non-zero initialization. The coupling arises because parameters used to construct $w_{11}$ and $w_{21}$ overlap. Specifically, elements from the first column of matrix $\boldsymbol{B}$ (i.e., $b_{11}, b_{21}$) are common to the computation of both $w_{11}$ and $w_{21}$. This shared dependency links the dynamics. The below illustration highlights these shared (teal) and distinct (red/blue) parameters involved in forming the observed entries $w_{11}$ and $w_{21}$:
$$\begin{bmatrix} w_{11} & w_{12} \\ w_{21} & w_{22} \end{bmatrix} = \begin{bmatrix} a_{11} & a_{12} \\ a_{21} & a_{22} \end{bmatrix} \begin{bmatrix} b_{11} & b_{12} \\ b_{21} & b_{22} \end{bmatrix}$$
$$w_{11} = a_{11}b_{11} + a_{12}b_{21}$$
$$w_{21} = a_{21}b_{11} + a_{22}b_{21}$$
The shared use of $b_{11}$ and $b_{21}$ in reconstructing both observed entries is what couples their learning dynamics.

### B.1.2 DEPTH ≥ 3 MODEL

For deeper matrices ($L \geq 3$), training dynamics are typically coupled, irrespective of the observation pattern (See Proposition B.1). Consider, for instance, predicting entries from the disconnected matrix $\boldsymbol{M}_{\mathrm{D}}$ where only diagonal elements are observed:

$$\boldsymbol{M}_{\mathrm{D}} \triangleq \begin{bmatrix} w_{11}^* & ? \\ ? & w_{22}^* \end{bmatrix}.$$

Even with such observations, for $L \geq 3$, coupling arises because parameters in intermediate layers are involved in computing multiple observed entries. This is illustrated in the following depth-3 example ($\boldsymbol{W}_{3:1} = \boldsymbol{W}_1 \boldsymbol{W}_2 \boldsymbol{W}_3$). Elements of the intermediate matrix $\boldsymbol{W}_2$ (colored teal) contribute to both the computation of $w_{11}$ and $w_{22}$:

$$\begin{bmatrix} w_{11} & w_{12} \\ w_{21} & w_{22} \end{bmatrix} = \begin{bmatrix} (w_1)_{11} & (w_1)_{12} \\ (w_1)_{21} & (w_1)_{22} \end{bmatrix} \begin{bmatrix} (w_2)_{11} & (w_2)_{12} \\ (w_2)_{21} & (w_2)_{22} \end{bmatrix} \begin{bmatrix} (w_3)_{11} & (w_3)_{12} \\ (w_3)_{21} & (w_3)_{22} \end{bmatrix}.$$

Specifically, the observed entries are formed as:

$$\begin{aligned} w_{11} = & \left( (w_1)_{11}(w_2)_{11} + (w_1)_{12}(w_2)_{21} \right)(w_3)_{11} \\ & + \left( (w_1)_{11}(w_2)_{12} + (w_1)_{12}(w_2)_{22} \right)(w_3)_{21}, \\ w_{22} = & \left( (w_1)_{21}(w_2)_{11} + (w_1)_{22}(w_2)_{21} \right)(w_3)_{12} \\ & + \left( (w_1)_{21}(w_2)_{12} + (w_1)_{22}(w_2)_{22} \right)(w_3)_{22}. \end{aligned}$$

The shared involvement of all elements from $\boldsymbol{W}_2$ (the teal matrix) in forming both $w_{11}$ and $w_{22}$ leads to coupled dynamics, provided these elements are non-zero. (Conversely, if some elements were to become zero, this could potentially lead to decoupled dynamics, as illustrated in the subsequent subsection.)

### B.2 DECOUPLED DYNAMICS EXAMPLE

### B.2.1 DEPTH-2 MODEL

For depth-2 models, decoupled dynamics coincide with disconnected observation patterns. Indeed, by Lemma B.1,

$$\begin{aligned} \langle \nabla_{\boldsymbol{\theta}} w_{ij}, \nabla_{\boldsymbol{\theta}} w_{pq} \rangle &= \sum_{l=1}^{2} \left( \boldsymbol{e}_i^{\top} \boldsymbol{T}_l \boldsymbol{e}_p \right) \left( \boldsymbol{e}_j^{\top} \boldsymbol{S}_l \boldsymbol{e}_q \right) \\ &= \left( \boldsymbol{e}_i^{\top} \boldsymbol{W}_2 \boldsymbol{W}_2^{\top} \boldsymbol{e}_p \right) \delta_{jq} + \delta_{ip} \left( \boldsymbol{e}_j^{\top} \boldsymbol{W}_1^{\top} \boldsymbol{W}_1 \boldsymbol{e}_q \right), \end{aligned}$$

where $\delta_{ab} = 1$ if $a = b$ and 0 otherwise. Hence, if $i \neq p$ and $j \neq p$, the inner product is identically zero for all weights, which explains the decoupling for the depth-2 matrix when the observations are disconnected.

To illustrate the disconnected case, consider the $2 \times 2$ incomplete matrix example $\boldsymbol{M}_{\mathrm{D}}$, to be completed from diagonal-only observations.

$$\boldsymbol{M}_{\mathrm{D}} \triangleq \begin{bmatrix} w_{11}^* & ? \\ ? & w_{22}^* \end{bmatrix}.$$

Then the observation matrix $\boldsymbol{P}_{\mathrm{D}}$ can be constructed as:

$$\boldsymbol{P}_{\mathrm{D}} = \begin{bmatrix} 1 & 0 \\ 0 & 1 \end{bmatrix},$$

and the adjacency matrix $\mathcal{A}_{\mathrm{D}}$ can be constructed as:

$$\mathcal{A}_{\mathrm{D}} = \begin{bmatrix} \boldsymbol{0}_{2,2} & \boldsymbol{P}_{\mathrm{D}}^{\top} \\ \boldsymbol{P}_{\mathrm{D}} & \boldsymbol{0}_{2,2} \end{bmatrix} = \begin{bmatrix} 0 & 0 & 1 & 0 \\ 0 & 0 & 0 & 1 \\ 1 & 0 & 0 & 0 \\ 0 & 1 & 0 & 0 \end{bmatrix},$$

which forms the disconnected graph as illustrated in Figure 1a. This setup inherently leads to decoupled training dynamics. The decoupling can be visually understood by examining how distinct sets of elements in the factor matrices $\boldsymbol{A}$ and $\boldsymbol{B}$ contribute to the observed entries $w_{11}$ and $w_{22}$. Specifically, as illustrated below, red-colored entries are exclusively involved in predicting $w_{11}$, while blue-colored entries are exclusively involved in predicting $w_{22}$. These two sets of entries are disjoint, confirming the decoupled nature of the dynamics:

$$\begin{bmatrix} w_{11} & w_{12} \\ w_{21} & w_{22} \end{bmatrix} = \begin{bmatrix} a_{11} & a_{12} \\ a_{21} & a_{22} \end{bmatrix} \begin{bmatrix} b_{11} & b_{12} \\ b_{21} & b_{22} \end{bmatrix},$$

$$w_{11} = a_{11}b_{11} + a_{12}b_{21},$$

$$w_{22} = a_{21}b_{12} + a_{22}b_{22}.$$

### B.2.2 DEPTH$\geq 3$ MODEL

For deep ($L \geq 3$) matrices, decoupled training dynamics are observed in at least two key scenarios. First, as detailed in Appendix D.2.3, an $\alpha \boldsymbol{I}_d$ initialization combined with block-diagonal observations leads to decoupled dynamics for any depth-factorized matrix.

To illustrate this for a deeper case, we revisit the $\boldsymbol{M}_{\mathrm{D}}$ observation pattern in a depth-3 context. Appendix D.2.3 states that with such an initialization and observing only diagonal entries (which corresponds to $s = 1$ case), all off-diagonal elements of the factor matrices $\boldsymbol{W}_l(t)$ remain zero throughout training. Consequently, the factor matrices $\boldsymbol{W}_1, \boldsymbol{W}_2, \boldsymbol{W}_3$ are diagonal. The product matrix $\boldsymbol{W}_{L:1}(t)$ is thus formed as:

$$\begin{bmatrix} w_{11} & w_{12} \\ w_{21} & w_{22} \end{bmatrix} = \begin{bmatrix} (w_1)_{11} & 0 \\ 0 & (w_1)_{22} \end{bmatrix} \begin{bmatrix} (w_2)_{11} & 0 \\ 0 & (w_2)_{22} \end{bmatrix} \begin{bmatrix} (w_3)_{11} & 0 \\ 0 & (w_3)_{22} \end{bmatrix}.$$

The observed entries are therefore computed as products of the respective diagonal elements:

$$w_{11} = (w_1)_{11}(w_2)_{11}(w_3)_{11},$$

$$w_{22} = (w_1)_{22}(w_2)_{22}(w_3)_{22}.$$

Since $w_{11}$ depends only on the set of parameters $\{(\boldsymbol{W}_k)_{11}\}_{k=1}^3$ and $w_{22}$ depends only on the entirely disjoint set of parameters $\{(\boldsymbol{W}_k)_{22}\}_{k=1}^3$, their training dynamics are decoupled.

Second, the training dynamics are also decoupled when all factor matrices are initialized as $d \times d$ zero matrices, $\boldsymbol{0}_{d \times d}$. To see this, note that by the chain rule, we have

$$\frac{\partial w_{pq}(t)}{\partial (w_l(t))_{ij}} = (\boldsymbol{W}_L(t)\boldsymbol{W}_{L-1}(t)\cdots\boldsymbol{W}_{l+1}(t))_{pi} (\boldsymbol{W}_{l-1}(t)\boldsymbol{W}_{l-2}(t)\cdots\boldsymbol{W}_1(t))_{jq}, \quad (12)$$

where we define the $(i,j)$-th entry of the factor matrix $\boldsymbol{W}_l(t)$ as $(w_l(t))_{ij}$. If at some time $t$ all factor matrices satisfy $\boldsymbol{W}_k(t) = \boldsymbol{0}$, then the right-hand side of (12) is the zero matrix, and thus

$$\frac{\partial w_{pq}(t)}{\partial (w_l(t))_{ij}} = 0 \quad \text{for all } p, q.$$

Therefore,

$$\frac{\partial \phi}{\partial (w_l(t))_{ij}} = \sum_{(p,q)\in\Omega} \left(w_{pq}(t) - w_{pq}^*\right) \frac{\partial w_{pq}(t)}{\partial (W_l(t))_{ij}} = 0,$$

which implies

$$(\dot{w_l}(t))_{ij} = -\frac{\partial \phi}{\partial (w_l(t))_{ij}} = 0.$$

Since the initial condition is $(w_l(0))_{ij} = 0$, uniqueness of ODE solutions guarantees that $(w_l(t))_{ij} \equiv 0$ for all $t \geq 0$. As this holds for arbitrary $l, i, j$, we conclude that $\boldsymbol{W}_l(t) \equiv \boldsymbol{0}$ for all $l$ and all $t \geq 0$.

Finally, because $\nabla_{\boldsymbol{\theta}(t)} w_{pq}(t) = \boldsymbol{0}$ for all $p, q$ and $t \geq 0$, the inner product condition

$$\langle \nabla_{\boldsymbol{\theta}(t)} w_{ij}(t), \nabla_{\boldsymbol{\theta}(t)} w_{pq}(t) \rangle = 0$$

is satisfied for all $(i,j), (p,q) \in \Omega$ and for all $t \geq 0$. Hence, the dynamics are (trivially) decoupled.

## C    ADDITIONAL EXPERIMENTS

This section provides additional experiments omitted from the main text.

### C.1    IMPLICIT BIAS EXPERIMENTS

**Connected vs Disconnected Observation Patterns.**    In Figure 1, we present experiments with specific choices of $M_C$ and $M_D$, which are $2 \times 2$ rank-1 ground-truth matrices illustrating connected and disconnected examples, respectively. To generalize these observations, we extended our experiments to a $3 \times 3$ rank-1 ground truth matrix, considering all possible connected and disconnected observation patterns. After accounting for symmetries to eliminate duplicates, this results in a total of 23 unique observation patterns, which are categorized into 17 connected and 6 disconnected cases.

For each of these 23 observation patterns, the $3 \times 3$ rank-1 ground truth matrix was generated using constituent vectors whose entries were sampled from a standard normal distribution. Each factor matrix was then initialized by sampling its entries from a Gaussian distribution with a mean of zero and a standard deviation of $\alpha$. We performed 10 independent trials for each pattern.

Figure 4 illustrates that, consistent with the findings in Figure 1, a significant discrepancy exists between the behavior of depth-2 matrices and that of deeper matrices. This discrepancy becomes notably more pronounced for the disconnected observation patterns.

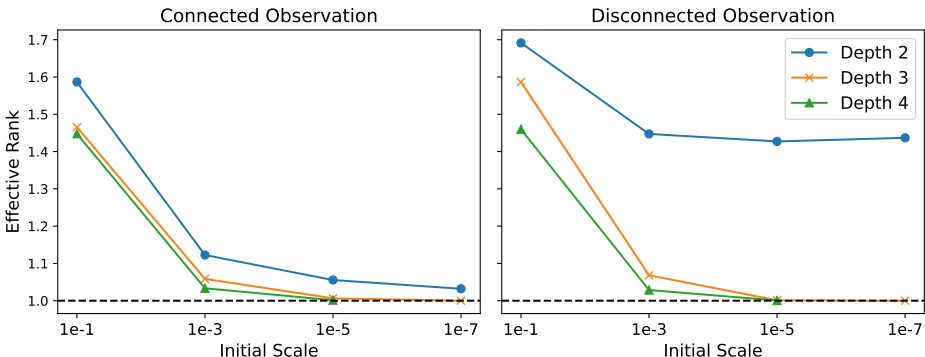

Figure 4: The left panel shows the averaged effective rank of all possible connected patterns as a function of the initial scale $\alpha^L$. The right panel displays the averaged effective rank of all possible disconnected patterns.

**Numerical Solutions of the Implicit Equations.**    We next provide a theoretical validation of our main claim: coupled dynamics induce a low-rank bias, whereas decoupled dynamics do not. This validation builds on Theorem 3.3, under various conditions, by numerically solving the equations while varying the ground truth value $w^*$, the number of blocks $n$, and the block size $s$. The results shown in Figure 5 (for $w^* = 1$, $n = 5$, $s = 2$), Figure 6 (for $w^* = 10, n = 10, s = 1$), Figure 7 (for $w^* = 0.1, n = 10, s = 1$), and Figure 8 (for $w^* = 1, n = 3, s = 1$) provide strong supporting evidence for the claim.

**Gradient Descent Validation.**    Furthermore, we ran gradient descent with a sufficiently small step size to validate our derived equations. For the results shown in Figure 9, we replicated the setup of Figure 8 ($w^* = 1, n = 3, s = 1$), excluding the $\alpha = 10^{-10}$ case due to prohibitive computation time. The observed values closely match the theoretical predictions from Theorem 3.3, as illustrated in Figure 8.

**Comparison with Gaussian Initialization.**    To validate that our initialization scheme (7) can achieve comparable outcomes to Gaussian initialization while offering more control, we conducted experiments on a $3 \times 3$ matrix completion task with diagonal observations (i.e., $w_{11}^* = w_{22}^* = w_{33}^* = 1$). While our scheme allows initial rank properties to be adjusted via the parameter $m$, Gaussian initialization's inherent randomness precludes such direct control. Therefore, for comparison with

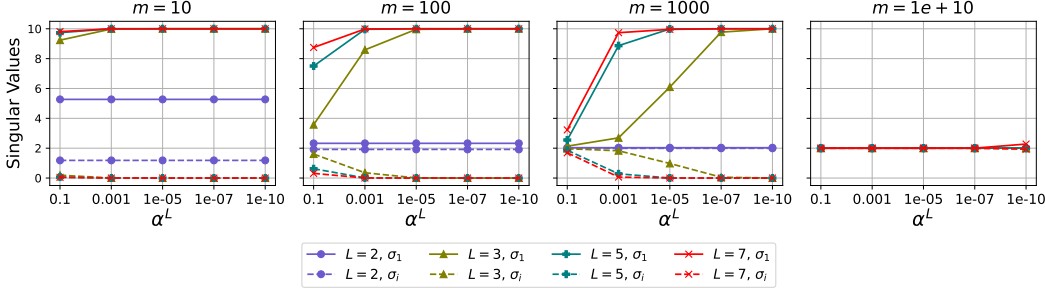

Figure 5: Singular values of $\boldsymbol{W}_{L:1}(\infty)$ (numerically obtained from Theorem 3.3) against initialization scale $\alpha^L$ for the block-diagonal observation task. Solid lines represent the largest singular value $\sigma_1$; dashed lines denote the identical singular values $\sigma_i$ for $i \in \{2, \ldots, n\}$. Note that $\sigma_j$ for $j \in \{n+1, \ldots, d\}$ are all zero. For finite $m$, these results show that both greater depth $L$ and a smaller initial scale $\alpha$ strengthen the low-rank bias, in contrast to the $L = 2$ case. Conversely, when $m$ is extremely large (e.g., $m = 10^{10}$), approximating an $\alpha \boldsymbol{I}_d$ rank $d$ initialization, the dynamics decouple and cannot achieve the minimal low-rank solution, regardless of $L$ or $\alpha$.

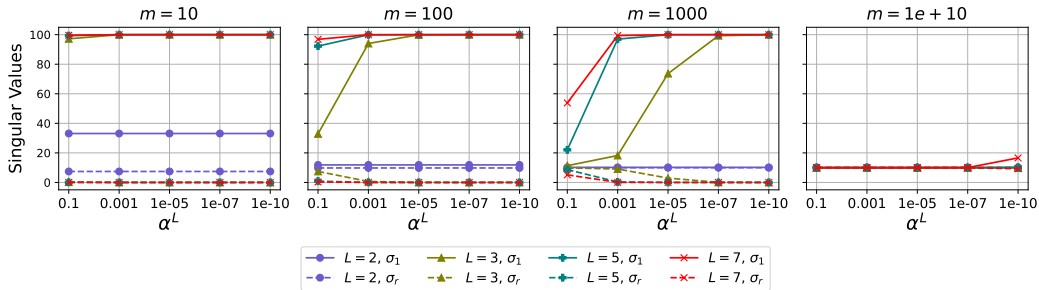

Figure 6: Numerical conditions identical to those in Figure 2, except with ground truth value $w^* = 10$ and dimension $d = 10$ where the block size is $s = 1$.

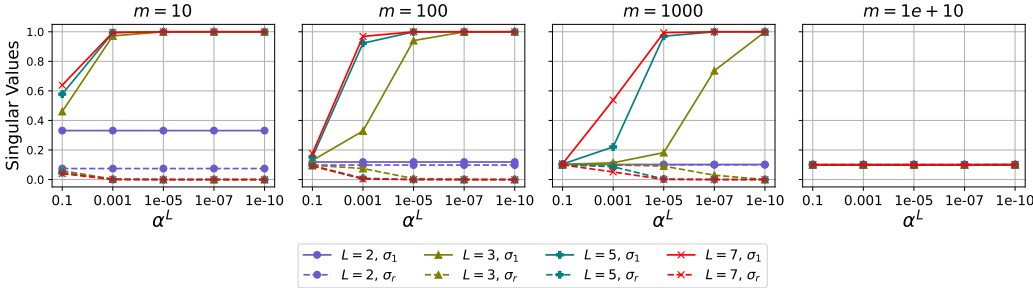

Figure 7: Numerical conditions identical to those in Figure 2, except with ground truth value $w^* = 0.1$ and dimension $d = 10$ where the block size is $s = 1$.

Gaussian initialization, we ran 1000 independent seeds and sorted the converged solutions by their rank. A comparison of the results in Figure 10 suggests that the behavioral trends may appear similar. In the depth-2 case, both initializations tend to converge to high-rank solutions. Moreover, for both initializations, a clear gap emerges between $L = 2$ and $L = 3$, with the depth-3 model exhibiting a stronger low-rank bias. For deeper networks ($L \geq 3$), the tendency to converge toward lower-rank solutions becomes increasingly pronounced as depth increases.

**Noisy Diagonal Experiments.** We also experimented with observing noisy diagonal entries using gradient descent. In particular, instead of fixing all ground truth diagonal entries to be equal, we perturbed them as $(\boldsymbol{W}^*)_{ii} = w^* + \epsilon_i$, where $\epsilon_i \sim \mathcal{N}(0, \sigma^2)$. We set ($w^* = 1$), dimension ($d = 5$),

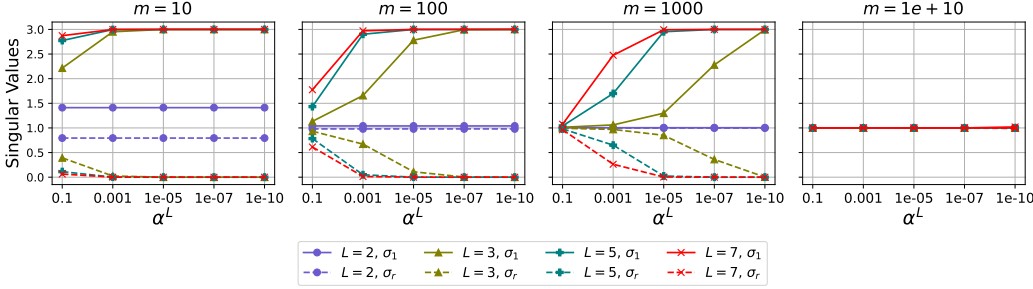

Figure 8: Numerical conditions identical to those in Figure 2, except with ground truth value $w^* = 1$ and dimension $d = 3$ where the block size is $s = 1$.

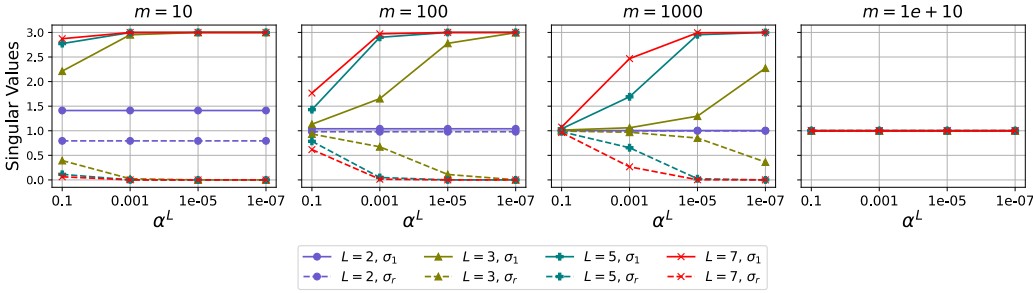

Figure 9: Gradient descent experiments conducted under conditions identical to those in Figure 8.

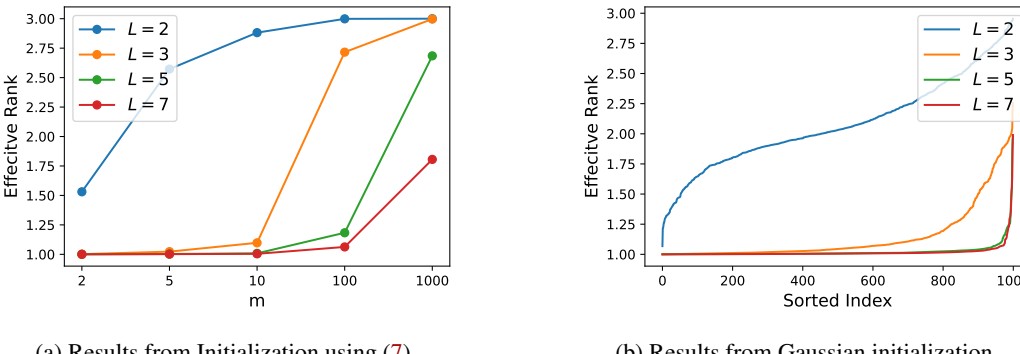

(a) Results from Initialization using (7).

(b) Results from Gaussian initialization.

Figure 10: **(a)** Effective rank for the initialization scheme in (7). The x-axis denotes the parameter $m$, which controls the initial rank characteristics of the model, while the y-axis represents the corresponding effective rank after convergence. **(b)** Effective rank distributions for Gaussian initialization. The results are from 1000 independent trials, sorted by their converged effective rank. The x-axis denotes the sorted trial index (from lowest to highest converged rank), and the y-axis represents the corresponding effective rank after convergence.

and used the initialization scheme (7) with $m = 100$. For each configuration, we independently sampled 10 noise realizations and report the average behavior along with the standard deviations.

As shown in Figure 11, the qualitative trends are consistent with our theory. When $L = 2$, the model converges to a high-rank solution largely independently of the initialization scale, whereas for deeper networks the stable rank decreases as depth increases, indicating a stronger low-rank bias. We also observe that larger noise levels lead to more pronounced low-rank behavior. This is natural, since increasing the noise drives the ground truth further away from the identity. Moreover, the dependence on the noise magnitude appears continuous: in the small noise regime (leftmost panel), the change in stable rank is relatively mild, while in the larger noise regime (rightmost panel), the gap

becomes more substantial. These experiments suggest that our depth-induced low-rank phenomenon is empirically robust to moderate perturbations of the diagonal entries.

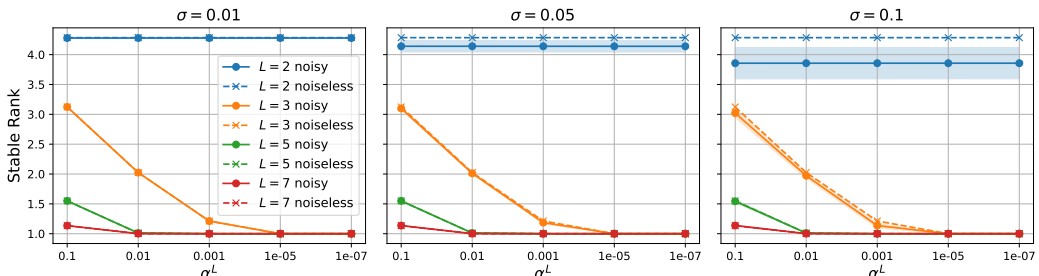

Figure 11: Limiting stable rank (y-axis) as a function of $\alpha^L$ (x-axis) under noisy diagonal observations. Dashed lines indicate the noiseless baseline, and solid lines indicate the noisy case. The noise standard deviation is set to $\sigma = 0.01$ (leftmost), $\sigma = 0.05$ (middle), and $\sigma = 0.1$ (rightmost). The depth dependent low-rank bias persists and follows a trend similar to the noiseless setting.

**Non-equal Diagonal Experiments.** We also experimented with observing non-equal diagonal entries using gradient descent. In particular, instead of fixing all ground truth diagonal entries to be equal, we assigned different values to each diagonal entry. We set the dimension to $d = 5$ and take the diagonal entries of $\boldsymbol{W}^*$ to be $0, 0.5, 1, 1.5, 2$, respectively, and used the initialization scheme (7).

As shown in Figure 12, the qualitative trends are consistent with our theory. When $L = 2$, the model converges to a high rank solution independently of the initialization scale, whereas for deeper networks the stable rank decreases as depth increases. For the case $m = \infty$ (rightmost plot), all models converge to high rank solutions regardless of depth, which is consistent with Theorem 3.3.

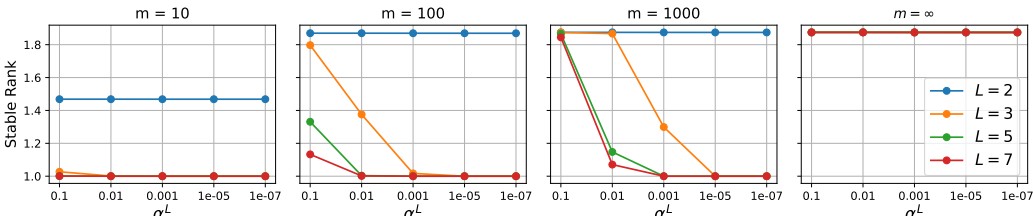

Figure 12: Limiting stable rank (y-axis) as a function of the initialization scale (x-axis) under non-equal diagonal observations. The low-rank bias induced by coupled training dynamics persists and closely matches the behavior in the equal-diagonal setting described in Theorem 3.3 and Figure 2.

**Additional Optimizer Ablations.** We also experimented with other optimizers, including adaptive methods, such as stochastic gradient descent (SGD), gradient descent with momentum, Adam, RMSProp, and Adagrad. In this experiment, we fix the dimension to $d = 5$, use Gaussian initialization with diagonal observations ($s = 1$) with $w^* = 1$, and run gradient based optimization with a sufficiently small step size over 10 random seeds. For each optimizer, we use the default hyperparameters from the PyTorch implementation, and for SGD we update the model using one observed entry per iteration.

The results in Figures 13-17 align well with our theory: for depth-2 (which induces decoupled dynamics), the model converges to high-rank solutions across initialization scales, whereas for depth $L \geq 3$ (which induces coupled dynamics) the solutions become increasingly low-rank as the initialization scale decreases and as depth increases.

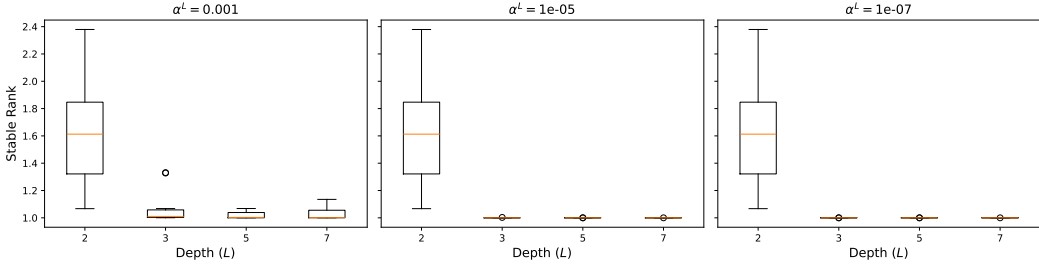

Figure 13: Final stable rank as a function of depth. Each panel corresponds to a different initialization scale. Results are obtained using SGD.

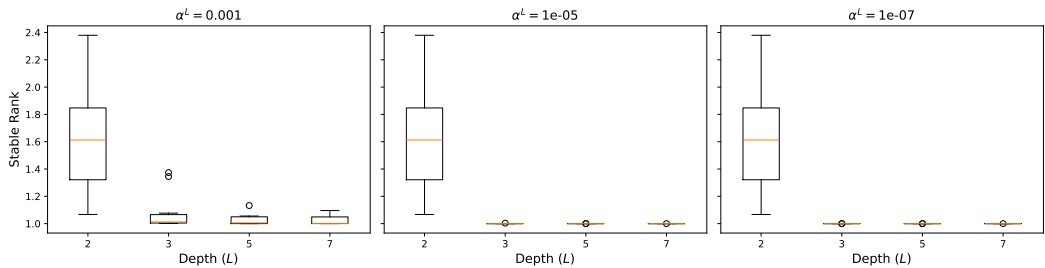

Figure 14: Final stable rank as a function of depth. Each panel corresponds to a different initialization scale. Results are obtained using GD with momentum.

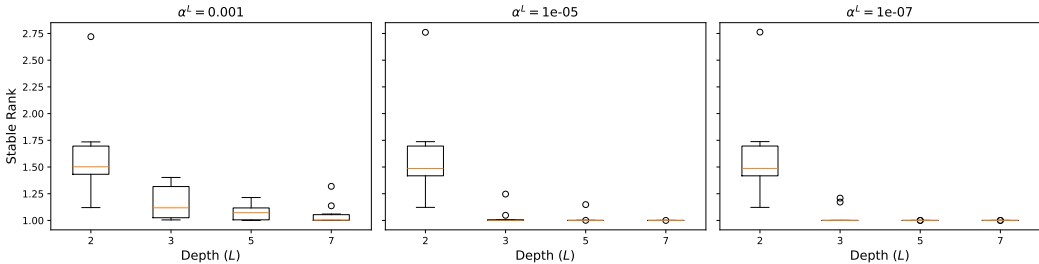

Figure 15: Final stable rank as a function of depth. Each panel corresponds to a different initialization scale. Results are obtained using Adam.

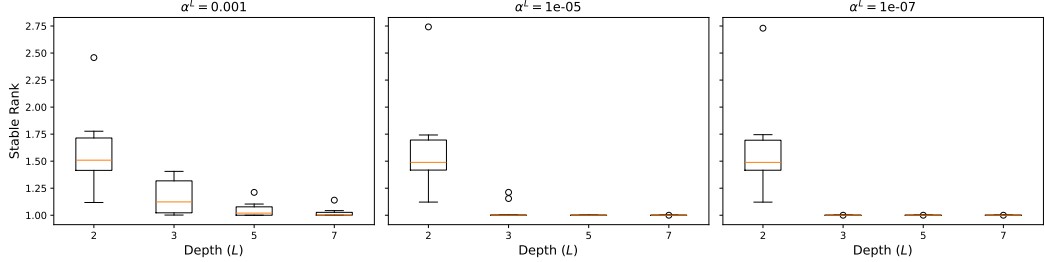

Figure 16: Final stable rank as a function of depth. Each panel corresponds to a different initialization scale. Results are obtained using RMSProp.

### C.1.1 EXPERIMENTS IN NEURAL NETWORKS

To study how depth influences low rank bias in practice, we train ResNet and VGG models across varying depths. While Huh et al. (2021) show that deeper networks yield lower rank embeddings,

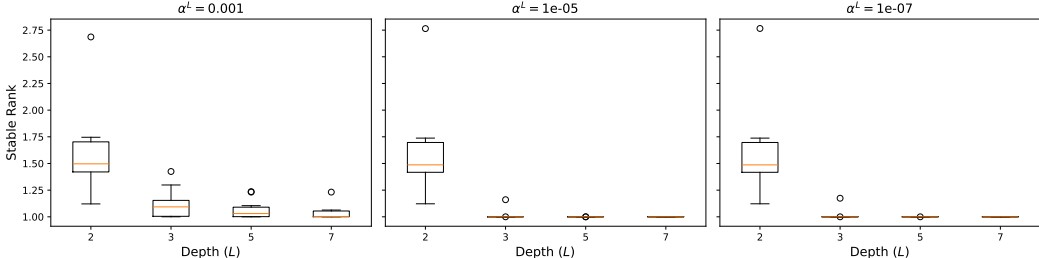

Figure 17: Final stable rank as a function of depth. Each panel corresponds to a different initialization scale. Results are obtained using Adagrad.

their analysis does not address the weight matrices. Following Galanti et al. (2023), we measure the effective rank of the weight matrices directly and find that deeper networks are biased toward low-rank solutions.

To be more specific, we train ResNet–18, 34, 50, and 101, as well as VGG–11, 13, 16, and 19, on CIFAR-10 and CIFAR-100 for 200 epochs with a batch size of 128. Training uses SGD with momentum, Adam, and RMSProp. The initial learning rates are 0.1 for SGD with momentum, and 0.001 for Adam and RMSProp. We apply weight decay of 0.0005 for SGD with momentum and 1e-05 for Adam and RMSProp. A cosine annealing scheduler is used together with standard data augmentation (horizontal flipping and random cropping).

We measure the effective rank across all layers except the final one and average them to obtain a single scalar. Following Galanti et al. (2023), each weight tensor $\mathbf{Z} \in \mathbb{R}^{c_{\text{in}} \times c_{\text{out}} \times k_1 \times k_2}$ of a convolutional layer, where $c_{\text{in}}$ and $c_{\text{out}}$ denote the numbers of input and output channels and $(k_1, k_2)$ is the kernel size, is reshaped into a matrix $\boldsymbol{W} \in \mathbb{R}^{c_{\text{in}} \times (c_{\text{out}} k_1 k_2)}$ to measure the layer's effective rank. We report averages over five runs with 95% confidence intervals.

The results in Figures 18 to 21 for SGD with momentum, Figures 22 to 25 for Adam, and Figures 26 to 29 for RMSProp consistently show that the average effective rank decreases as depth increases. This trend is consistent with Theorem 3.3, which establishes the depth induced low-rank bias in matrix completion settings.

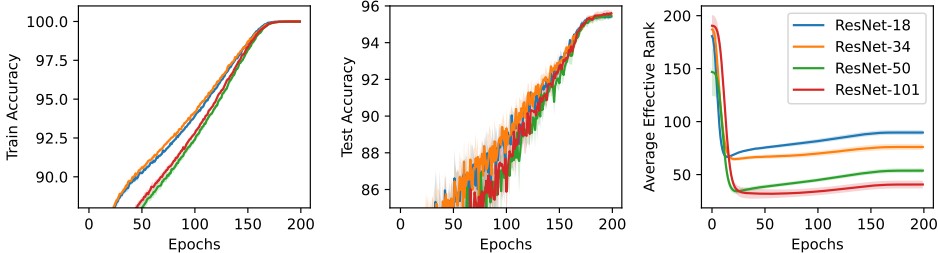

Figure 18: We train CIFAR-10 with ResNet models ranging from 18 to 101 layers using SGD with momentum, averaging results over five independent runs with 95% confidence intervals. The leftmost plot reports the training accuracy, the middle plot the test accuracy, and the rightmost plot the average effective rank. As depth increases, the average effective rank decreases.

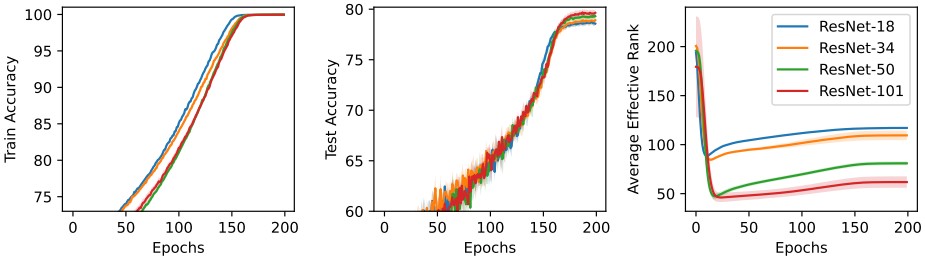

Figure 19: The results for CIFAR-100 with ResNet-18 to 101, under the same conditions as in Figure 18.

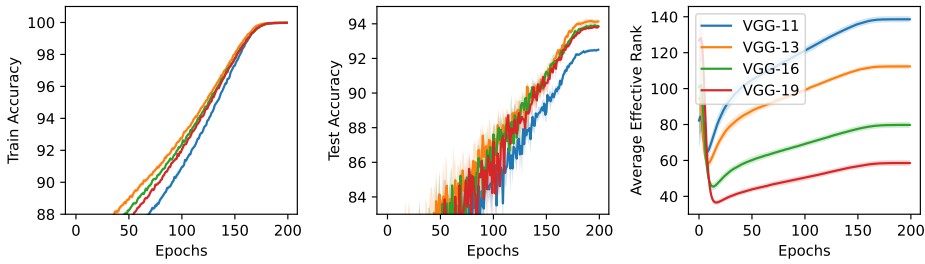

Figure 20: The results for CIFAR-10 with VGG-11 to 19, under the same conditions as in Figure 18.

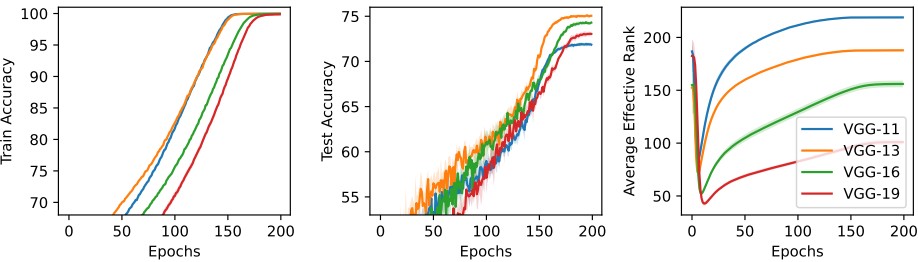

Figure 21: The results for CIFAR-100 with VGG-11 to 19, under the same conditions as in Figure 18.

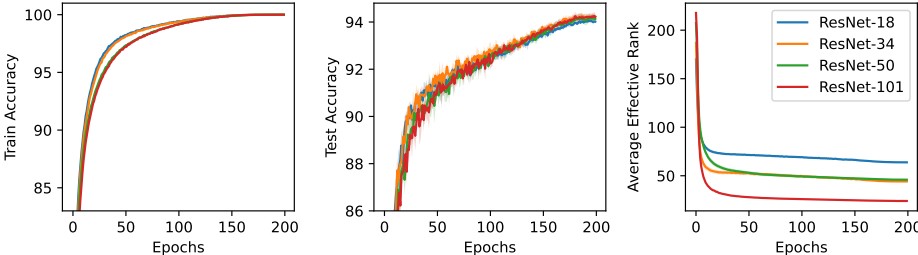

Figure 22: We train CIFAR-10 with ResNet models ranging from 18 to 101 layers using Adam, averaging results over five independent runs with 95% confidence intervals. The leftmost plot reports the training accuracy, the middle plot the test accuracy, and the rightmost plot the average effective rank. As depth increases, the average effective rank decreases.

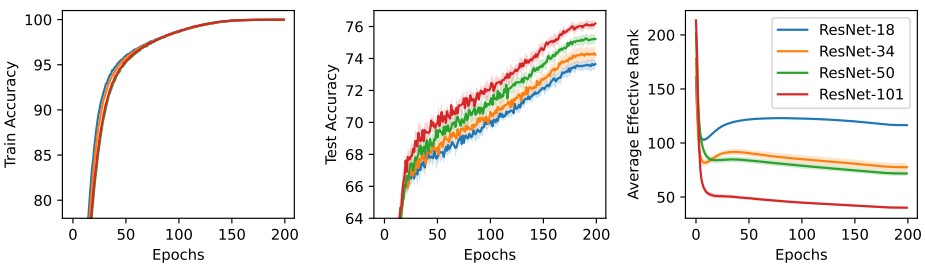

Figure 23: The results for CIFAR-100 with ResNet-18 to 101, under the same conditions as in Figure 22.

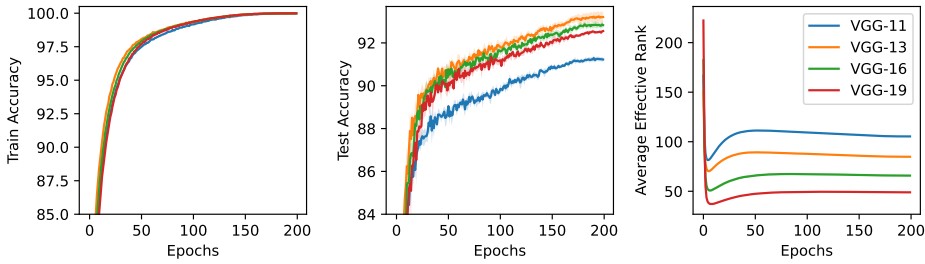

Figure 24: The results for CIFAR-10 with VGG-11 to 19, under the same conditions as in Figure 22.

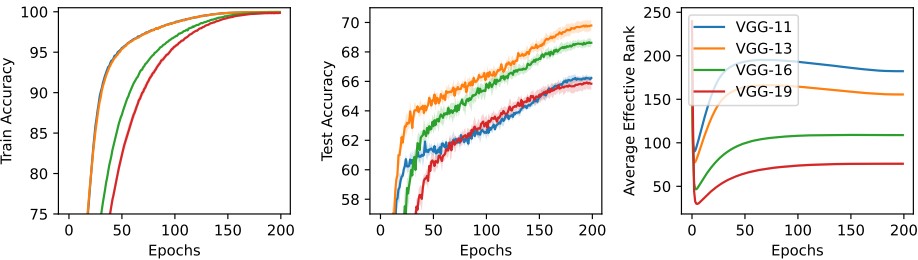

Figure 25: The results for CIFAR-100 with VGG-11 to 19, under the same conditions as in Figure 22.

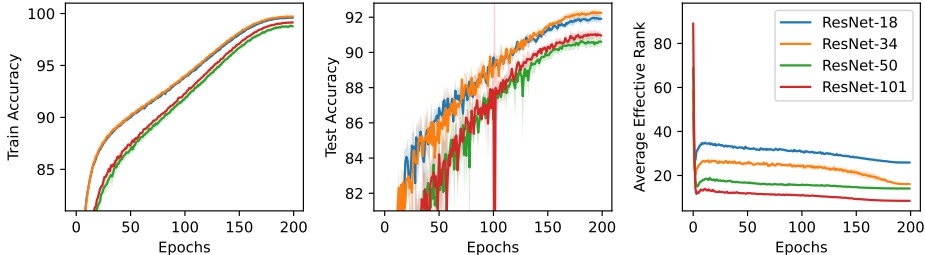

Figure 26: We train CIFAR-10 with ResNet models ranging from 18 to 101 layers using RMSProp, averaging results over five independent runs with 95% confidence intervals. The leftmost plot reports the training accuracy, the middle plot the test accuracy, and the rightmost plot the average effective rank. As depth increases, the average effective rank decreases.

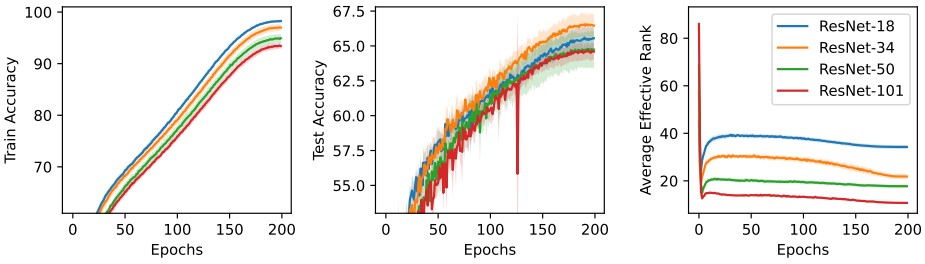

Figure 27: The results for CIFAR-100 with ResNet-18 to 101, under the same conditions as in Figure 26.

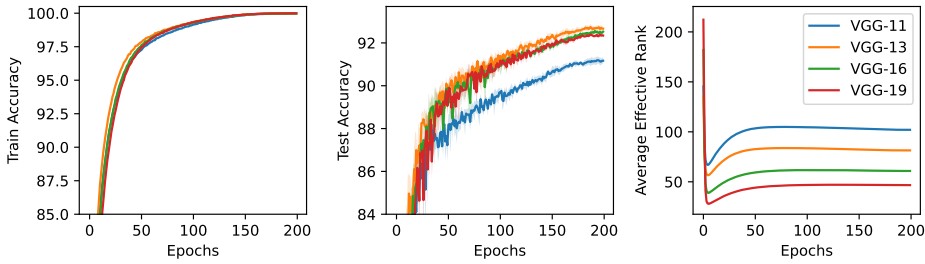

Figure 28: The results for CIFAR-10 with VGG-11 to 19, under the same conditions as in Figure 26.

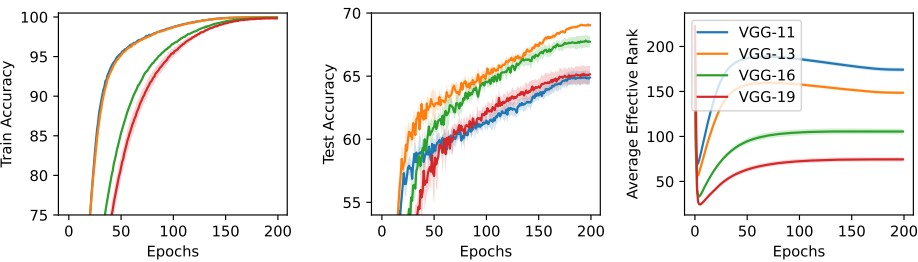

Figure 29: The results for CIFAR-100 with VGG-11 to 19, under the same conditions as in Figure 26.

**Coupled vs. Decoupled Dynamics in NN.** To examine whether coupled and decoupled training dynamics intensify low-rank bias in practical neural networks, we conducted an additional experiment with fully connected networks with ReLU activations, under both Gaussian and identity-based initializations, using the CIFAR-10 dataset. For the Gaussian initialization, all layers are initialized with i.i.d. Gaussian weights. For the identity-based initialization, all hidden layers are initialized as scaled identity matrices, while the first and last layers are initialized with Gaussian weights, since these layers are not square.

We train networks of depth $L \in \{2, 3, 5\}$ with a fixed hidden width of $512$ for $100$ epochs, using SGD with momentum and a constant learning rate of $0.01$. The results show that, even when both initializations successfully achieve low training loss, the low-rank bias is substantially stronger under Gaussian initialization compared to identity initialization, which indicates that low-rank bias is intensified under coupled training dynamics in a way that is consistent with our theoretical findings.

Furthermore, as depth increases, the stable rank of the weight matrices decreases under Gaussian initialization. In contrast, with identity-based initialization, deeper networks tend to converge to higher rank solutions. A plausible explanation is that, as depth grows, a larger fraction of the layers are initialized using identity (recall that the first and last layers are initialized under Gaussian), which makes the overall dynamics closer to a decoupled regime and therefore less biased toward low-rank solutions.

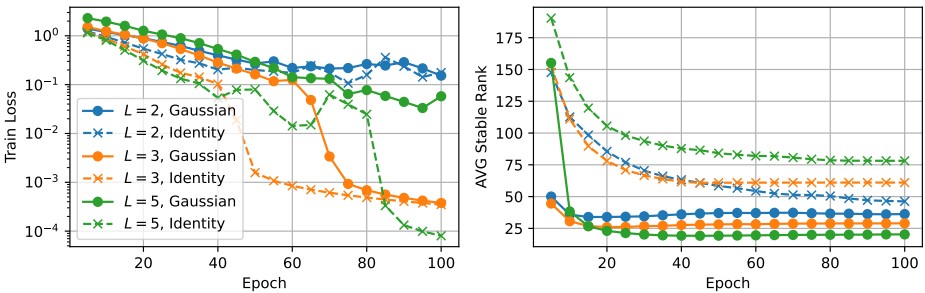

Figure 30: Left: training loss (log scale). Right: average stable rank across all layers except the last. Solid lines correspond to Gaussian initialization and dashed lines to identity-based initialization. Gaussian initialization (corresponding to coupled training dynamics) converges to noticeably lower rank than identity-based initialization (corresponding to more decoupled training dynamics).

## C.2 LOSS OF PLASTICITY EXPERIMENTS

Section 4.2 discusses a scenario where pre-training employs diagonal entries, after which an off-diagonal term (specifically, $w_{12}^*$) is introduced to restore connectivity, leading to coupled dynamics. Theorem 4.2 establishes that, in this situation, the model indeed does not converge to a low-rank solution. To empirically validate this theoretical finding, we conducted experiments using the family of initializations (7) tailored to this specific scenario, with results detailed in Figures 31 and 32. These experiments utilized a depth-2 model to reconstruct the ground-truth matrix, with an initialization scale set to $\alpha = 10^{-35}$. Notably, if the initialization scale $\alpha$ is set significantly lower, as the dynamics are coupled, a cold-started model can converge to solutions exhibiting a more pronounced low-rank structure.

For the case presented in Figure 31, where $w^* = 1, w_{12}^* = 0.1$, following Theorem 4.2, the theoretical lower bound on the stable rank for a warm-started model initialized diagonally ($m = \infty$) is approximately 1.45, while the empirically observed stable rank is approximately 1.8. Even in scenarios where substantial new information must be learned (e.g., by setting $w_{12}^*$ to a large value), loss of plasticity is empirically observed, primarily manifesting as high test error (i.e., a significant gap between the target $w_{21}^*$ and the converged $w_{21}$). While Theorem 4.2's analysis via stable rank does not fully explain an accompanying low-rank bias (a point consistent with Figure 32), the theorem does predict that $w_{21}$ converges to a negative value, which implies a large test loss.

Furthermore, we performed additional experiments with different diagonal entry values to investigate whether this argument extends to other scenarios (results shown in Figure 33), although specific theoretical guarantees have not been established for these broader cases. We observe that even in these varied settings, both the effective rank and the stable rank of a warm-started model substantially exceed one, whereas cold-started models can converge to lower-rank solutions.

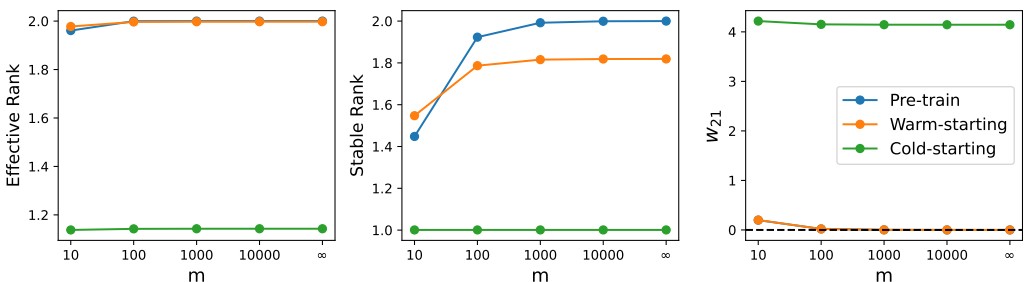

Figure 31: Experimental results for a $2 \times 2$ rank-1 ground-truth matrix $\boldsymbol{W}^*$ with $w_{11}^* = w_{22}^* = 1$ and $w_{12}^* = 0.5$ (implying $w_{21}^* = 2$ for rank-1 structure). Models, initialized according to (7), are first pre-trained on diagonal entries. After achieving zero-loss convergence in pre-training, the off-diagonal element $w_{12}^*$ is introduced, and models are subsequently trained on combined diagonal and off-diagonal observations. The plots display: (Left and Middle) effective rank under different settings; (Right) converged value of $w_{21}(\infty)$. Key observations: (1) Warm-starting with a model that converged to a high-rank solution during pre-training tends to maintain this high rank, even when presented with the same subsequent observations as a cold-started model. (2) In the theoretically analyzed $m = \infty$ case, $w_{21}(\infty) < 0$ is observed, which correlates with the highest effective rank.

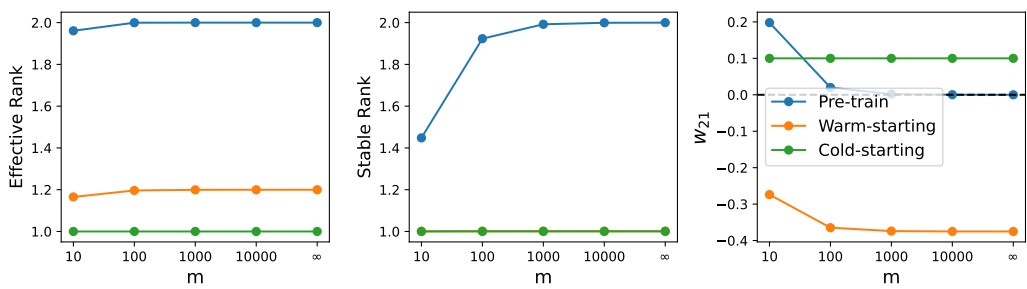

Figure 32: Experimental conditions identical to those in Figure 31, except with ground truth value $w_{12}^* = 10$. The model have to predict $w_{21}^*$ as 0.1

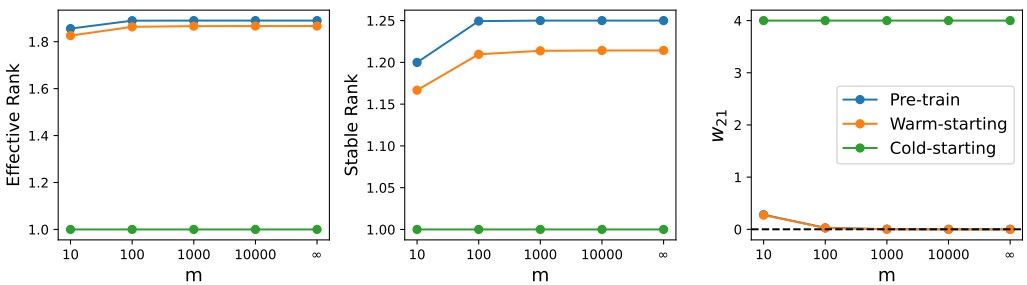

Figure 33: Experimental conditions identical to those in Figure 31, except with ground truth value $w_{11}^* = 1, w_{22}^* = 2$, and $w_{12}^* = 0.5$. The model have to predict $w_{21}^*$ as 4.

# D   PROOF FOR SECTION 3

In this and the following sections, we prove the Propositions and Theorems presented in the main text. We begin with the proof of Theorem 3.1.

## D.1   PROOF FOR THEOREM 3.1

When convergence is guaranteed, we can define the reference vector $\boldsymbol{u}^* \triangleq \frac{\boldsymbol{b}_1(\infty)}{\|\boldsymbol{b}_1(\infty)\|} \in \mathbb{R}^{d_1}$, which is entirely determined by their initial values and the targets. Note that $\boldsymbol{u}^*$ does not change with time, since it is defined at $t = \infty$. We decompose $\boldsymbol{a}_1(t)$, $\boldsymbol{a}_2(t)$, and $\boldsymbol{b}_1(t)$ into two components: one parallel to $\boldsymbol{u}^*$ and one perpendicular to $\boldsymbol{u}^*$:

$$\boldsymbol{a}_1(t) = \boldsymbol{a}_{1\|}(t) + \boldsymbol{a}_{1\perp}(t), \quad \boldsymbol{a}_2(t) = \boldsymbol{a}_{2\|}(t) + \boldsymbol{a}_{2\perp}(t), \quad \boldsymbol{b}_1(t) = \boldsymbol{b}_{1\|}(t) + \boldsymbol{b}_{1\perp}(t).$$

For any vector $\boldsymbol{u} \in \mathbb{R}^{d_1}$, the parallel component is defined as $\boldsymbol{u}_\| = (\boldsymbol{u}^{*\top}\boldsymbol{u})\boldsymbol{u}^*$, and the perpendicular component as $\boldsymbol{u}_\perp = \boldsymbol{u} - \boldsymbol{u}_\|$.

We introduce notation to quantify the alignment of each vector with $\boldsymbol{u}^*$:

$$\alpha_{\boldsymbol{a}_1}(t) = \boldsymbol{u}^{*\top}\boldsymbol{a}_1(t), \quad \alpha_{\boldsymbol{a}_2}(t) = \boldsymbol{u}^{*\top}\boldsymbol{a}_2(t), \quad \alpha_{\boldsymbol{b}_1}(t) = \boldsymbol{u}^{*\top}\boldsymbol{b}_1(t). \tag{13}$$

Additionally, we define notation to measure the magnitude of the perpendicular components:

$$\beta_{\boldsymbol{a}_1}(t) = \|\boldsymbol{a}_{1\perp}(t)\|_2^2, \quad \beta_{\boldsymbol{a}_2}(t) = \|\boldsymbol{a}_{2\perp}(t)\|_2^2, \quad \beta_{\boldsymbol{b}_1}(t) = \|\boldsymbol{b}_{1\perp}(t)\|_2^2. \tag{14}$$

Then, using equation (4), time evolution of each component in equation (13) can be written as:

$$\begin{aligned}
\dot{\alpha}_{\boldsymbol{a}_1}(t) &= \boldsymbol{u}^{*\top}\dot{\boldsymbol{a}}_1(t) \\
&= \underbrace{(w_{11}^* - \boldsymbol{a}_1^\top(t)\boldsymbol{b}_1(t))}_{\triangleq r_1(t)}\boldsymbol{u}^{*\top}\boldsymbol{b}_1(t) \\
&= r_1(t)\alpha_{\boldsymbol{b}_1}(t).
\end{aligned} \tag{15}$$

Likewise, for $\alpha_{\boldsymbol{a}_2}(t)$, we derive:

$$\begin{aligned}
\dot{\alpha}_{\boldsymbol{a}_2}(t) &= \boldsymbol{u}^{*\top}\dot{\boldsymbol{a}}_2(t) \\
&= \underbrace{(w_{21}^* - \boldsymbol{a}_2^\top(t)\boldsymbol{b}_1(t))}_{\triangleq r_2(t)}\boldsymbol{u}^{*\top}\boldsymbol{b}_1(t) \\
&= r_2(t)\alpha_{\boldsymbol{b}_1}(t).
\end{aligned} \tag{16}$$

Finally, for $\alpha_{\boldsymbol{b}_1}(t)$, we have:

$$\begin{aligned}
\dot{\alpha}_{\boldsymbol{b}_1}(t) &= \boldsymbol{u}^{*\top}\dot{\boldsymbol{b}}_1(t) \\
&= (w_{11}^* - \boldsymbol{a}_1^\top(t)\boldsymbol{b}_1(t))\boldsymbol{u}^{*\top}\boldsymbol{a}_1(t) + (w_{21}^* - \boldsymbol{a}_2^\top(t)\boldsymbol{b}_1(t))\boldsymbol{u}^{*\top}\boldsymbol{a}_2(t) \\
&= r_1(t)\alpha_{\boldsymbol{a}_1}(t) + r_2(t)\alpha_{\boldsymbol{a}_2}(t).
\end{aligned} \tag{17}$$

Also, for the perpendicular components, their time evolution can be derived as:

$$\begin{aligned}
\dot{\beta}_{\boldsymbol{a}_1}(t) &= 2\boldsymbol{a}_{1\perp}(t) \cdot \dot{\boldsymbol{a}}_{1\perp}(t) \\
&= 2\boldsymbol{a}_{1\perp}(t) \cdot \frac{d}{dt}\left(\boldsymbol{a}_1(t) - \left(\boldsymbol{u}^{*\top}\boldsymbol{a}_1(t)\right)\boldsymbol{u}^*\right) \\
&= 2\boldsymbol{a}_{1\perp}(t) \cdot \left(r_1(t)\boldsymbol{b}_1(t) - r_1(t)\left(\boldsymbol{u}^{*\top}\boldsymbol{b}_1(t)\right)\boldsymbol{u}^*\right).
\end{aligned}$$

Noting that $\boldsymbol{a}_{1\perp}(t)$ is perpendicular to $\boldsymbol{u}^*$, the second term in the parenthesis is zero. Thus, we have

$$\dot{\beta}_{\boldsymbol{a}_1}(t) = 2r_1(t)\boldsymbol{a}_{1\perp}(t)^\top\boldsymbol{b}_{1\perp}(t).$$

Likewise, for $\beta_{\boldsymbol{a}_2}(t)$ and $\beta_{\boldsymbol{b}_1}(t)$, we can derive their time derivative as:

$$\dot{\beta}_{\boldsymbol{a}_2}(t) = 2r_2(t)\boldsymbol{a}_{2\perp}(t)^\top\boldsymbol{b}_{1\perp}(t), \quad \dot{\beta}_{\boldsymbol{b}_1}(t) = \dot{\beta}_{\boldsymbol{a}_1}(t) + \dot{\beta}_{\boldsymbol{a}_2}(t).$$

Note that by the definition of $\boldsymbol{u}^*$, we have $\beta_{\boldsymbol{b}_1}(\infty) = 0$. Integrating the identity $\dot{\beta}_{\boldsymbol{b}_1}(t) = \dot{\beta}_{\boldsymbol{a}_1}(t) + \dot{\beta}_{\boldsymbol{a}_2}(t)$ from $t = 0$ to $\infty$ gives:

$$\beta_{\boldsymbol{a}_1}(\infty) + \beta_{\boldsymbol{a}_2}(\infty) = \underbrace{\beta_{\boldsymbol{a}_1}(0) + \beta_{\boldsymbol{a}_2}(0) - \beta_{\boldsymbol{b}_1}(0)}_{\triangleq \beta_0 \geq 0}.$$

This equation shows that if the initial value $\beta_0$ is small, it constrains the total perpendicular magnitude at convergence. However, since we do not know $\boldsymbol{u}^*$ in advance, one natural way to ensure small perpendicular components is to initialize the entire norms of $\boldsymbol{a}_1(0)$, $\boldsymbol{a}_2(0)$ to be sufficiently small.

To develop a more rigorous understanding, we analyze the parallel components. Under the assumption of convergence, we have:

$$\boldsymbol{a}_1(\infty)^\top \boldsymbol{b}_1(\infty) = w_{11}^*, \quad \boldsymbol{a}_2(\infty)^\top \boldsymbol{b}_1(\infty) = w_{21}^*.$$

Decomposing $\boldsymbol{a}_1(\infty)$ and $\boldsymbol{a}_2(\infty)$ leads to:

$$\boldsymbol{a}_1(\infty)^\top \boldsymbol{b}_1(\infty) = \left( \boldsymbol{a}_{1\perp}(\infty) + \boldsymbol{u}^{*\top} \boldsymbol{a}_1(\infty) \boldsymbol{u}^* \right)^\top \boldsymbol{b}_1(\infty)$$
$$= \alpha_{\boldsymbol{a}_1}(\infty) \alpha_{\boldsymbol{b}_1}(\infty) = w_{11}^*, \tag{18}$$

$$\boldsymbol{a}_2(\infty)^\top \boldsymbol{b}_1(\infty) = \left( \boldsymbol{a}_{2\perp}(\infty) + \boldsymbol{u}^{*\top} \boldsymbol{a}_2(\infty) \boldsymbol{u}^* \right)^\top \boldsymbol{b}_1(\infty)$$
$$= \alpha_{\boldsymbol{a}_2}(\infty) \alpha_{\boldsymbol{b}_1}(\infty) = w_{21}^*. \tag{19}$$

Using equations (15)–(17), and noting that

$$\frac{d}{dt} \alpha_{\boldsymbol{b}_1}^2(t) = \frac{d}{dt} (\alpha_{\boldsymbol{a}_1}^2(t) + \alpha_{\boldsymbol{a}_2}^2(t)),$$

we can integrate both sides of the equation over time from $0$ to $\infty$ to obtain:

$$\alpha_{\boldsymbol{a}_1}^2(\infty) + \alpha_{\boldsymbol{a}_2}^2(\infty) = \alpha_{\boldsymbol{b}_1}^2(\infty) + \underbrace{\alpha_{\boldsymbol{a}_1}^2(0) + \alpha_{\boldsymbol{a}_2}^2(0) - \alpha_{\boldsymbol{b}_1}^2(0)}_{\triangleq \alpha_0}. \tag{20}$$

By solving equations (18), (19), and (20), we can obtain closed-form solutions of $\alpha_{\boldsymbol{a}_1}(\infty), \alpha_{\boldsymbol{a}_2}(\infty)$, and $\alpha_{\boldsymbol{b}_1}(\infty)$ as follows:

$$\alpha_{\boldsymbol{a}_1}^2(\infty) = \frac{2 w_{11}^{*2}}{\sqrt{\alpha_0^2 + 4 w_{11}^{*2} + 4 w_{21}^{*2}} - \alpha_0}, \quad \alpha_{\boldsymbol{a}_2}^2(\infty) = \frac{2 w_{21}^{*2}}{\sqrt{\alpha_0^2 + 4 w_{11}^{*2} + 4 w_{21}^{*2}} - \alpha_0}, \tag{21}$$

$$\alpha_{\boldsymbol{b}_1}^2(\infty) = \frac{\sqrt{\alpha_0^2 + 4 w_{11}^{*2} + 4 w_{21}^{*2}} - \alpha_0}{2}. \tag{22}$$

Thus, we can upper bound the proportion of the perpendicular component of $\boldsymbol{a}_1(\infty)$ and $\boldsymbol{a}_2(\infty)$ relative to its total magnitude as follows:

$$\frac{\|\boldsymbol{a}_{1\perp}(\infty)\|^2}{\|\boldsymbol{a}_1(\infty)\|^2} = \frac{\beta_{\boldsymbol{a}_1}(\infty)}{\alpha_{\boldsymbol{a}_1}^2(\infty) + \beta_{\boldsymbol{a}_1}(\infty)} \leq \frac{\beta_0 \left( \sqrt{\alpha_0^2 + 4 w_{11}^{*2} + 4 w_{21}^{*2}} - \alpha_0 \right)}{2 w_{11}^{*2}},$$

$$\frac{\|\boldsymbol{a}_{2\perp}(\infty)\|^2}{\|\boldsymbol{a}_2(\infty)\|^2} = \frac{\beta_{\boldsymbol{a}_2}(\infty)}{\alpha_{\boldsymbol{a}_2}^2(\infty) + \beta_{\boldsymbol{a}_2}(\infty)} \leq \frac{\beta_0 \left( \sqrt{\alpha_0^2 + 4 w_{11}^{*2} + 4 w_{21}^{*2}} - \alpha_0 \right)}{2 w_{21}^{*2}}.$$

To further refine these bounds, we analyze the terms $\beta_0$ and $S(\alpha_0) \triangleq \sqrt{\alpha_0^2 + 4 w_{11}^{*2} + 4 w_{21}^{*2}} - \alpha_0$. By the definition of $\beta_0$, it is upper bounded by $\|\boldsymbol{a}_1(0)\|^2 + \|\boldsymbol{a}_2(0)\|^2 = \|\boldsymbol{A}(0)\|_F^2$. Also, by the definition of $\alpha_0$, we have:

$$-\|\boldsymbol{b}_1(0)\|_2^2 \leq \alpha_0 \leq \|\boldsymbol{A}(0)\|_F^2.$$

Noting that the function $f(x) = \sqrt{x^2 + C} - x$ (where $C > 0$) is non-negative and monotonically decreasing for all $x \in \mathbb{R}$, we can upper bound $S(\alpha_0)$ using the lower bound of $\alpha_0$:

$$
\begin{aligned}
S(\alpha_0) &\leq S(-\|\boldsymbol{b}_1(0)\|_2^2) \\
&= \sqrt{(-\|\boldsymbol{b}_1(0)\|_2^2)^2 + 4({w_{11}^*}^2 + {w_{21}^*}^2)} - (-\|\boldsymbol{b}_1(0)\|_2^2) \\
&= \sqrt{\|\boldsymbol{b}_1(0)\|_2^4 + 4({w_{11}^*}^2 + {w_{21}^*}^2)} + \|\boldsymbol{b}_1(0)\|_2^2 .
\end{aligned}
$$

Substituting these bounds for $\beta_0$ and $S(\alpha_0)$ into the inequality $\frac{\|\boldsymbol{a}_{1\perp}(\infty)\|^2}{\|\boldsymbol{a}_1(\infty)\|_2^2} \leq \frac{\beta_0 S(\alpha_0)}{2{w_{11}^*}^2}$, we obtain the final upper bound for the proportion of the perpendicular component of $\boldsymbol{a}_1(\infty)$:

$$
\frac{\|\boldsymbol{a}_{1\perp}(\infty)\|^2}{\|\boldsymbol{a}_1(\infty)\|_2^2} \leq \frac{\|\boldsymbol{A}(0)\|_F^2 \left( \sqrt{\|\boldsymbol{b}_1(0)\|_2^4 + 4({w_{11}^*}^2 + {w_{21}^*}^2)} + \|\boldsymbol{b}_1(0)\|_2^2 \right)}{2{w_{11}^*}^2} .
$$

A similar bound applies to $\frac{\|\boldsymbol{a}_{2\perp}(\infty)\|^2}{\|\boldsymbol{a}_2(\infty)\|_2^2}$:

$$
\frac{\|\boldsymbol{a}_{2\perp}(\infty)\|^2}{\|\boldsymbol{a}_2(\infty)\|_2^2} \leq \frac{\|\boldsymbol{A}(0)\|_F^2 \left( \sqrt{\|\boldsymbol{b}_1(0)\|_2^4 + 4({w_{11}^*}^2 + {w_{21}^*}^2)} + \|\boldsymbol{b}_1(0)\|_2^2 \right)}{2{w_{21}^*}^2} .
$$

## D.2 Proof for Proposition 3.2

According to the definition of coupled/decoupled dynamics presented in Definition 2, for the family of initializations defined in (7) along with the block-diagonal observations

$$\Omega_{\text{block}}^{(s,n)} \triangleq \bigcup_{b=1}^{n} \Omega_b, \quad \Omega_b \triangleq \{(i,j) : i,j \in \{(b-1)s+1,\dots,bs\}\},$$

so that $d = sn$ and each $\Omega_b$ corresponds to the index set of the $b$-th $s \times s$ diagonal block. We divide the cases to ensure that all possible scenarios for this family of initializations are covered.

The derivative of a block-diagonal observed entry $w_{pq}(t) \in \Omega_{\text{block}}^{(s,n)}$ with respect to $(\boldsymbol{W}_l(t))_{ij}$ is:

$$\frac{\partial w_{pq}(t)}{\partial (\boldsymbol{W}_l(t))_{ij}} = (\boldsymbol{W}_L(t)\boldsymbol{W}_{L-1}(t)\cdots\boldsymbol{W}_{l+1}(t))_{pi} (\boldsymbol{W}_{l-1}(t)\boldsymbol{W}_{l-2}(t)\cdots\boldsymbol{W}_1(t))_{jq}, \quad (23)$$

where the first term is $(p,i)$-th element of the product $\boldsymbol{W}_L(t)\boldsymbol{W}_{L-1}(t)\cdots\boldsymbol{W}_{l+1}(t)$, and the second term is $(j,q)$-th element of the product $\boldsymbol{W}_{l-1}(t)\boldsymbol{W}_{l-2}(t)\cdots\boldsymbol{W}_1(t)$. Then, we can express the gradient of $w_{ij}(t)$ with respect to $\boldsymbol{\theta}(t)$, which is the concatenation of all trainable parameters as follows:

$$\nabla_{\boldsymbol{\theta}} w_{pq}(t) = \left( \frac{\partial w_{pq}(t)}{\partial (\boldsymbol{W}_L(t))_{11}}, \frac{\partial w_{pq}(t)}{\partial (\boldsymbol{W}_L(t))_{12}}, \cdots \frac{\partial w_{pq}(t)}{\partial (\boldsymbol{W}_1(t))_{dd}} \right). \quad (24)$$

### D.2.1 Case for $L = 2$

For $L = 2$, each observed entry can be written as

$$w_{ij}(t) \triangleq (\boldsymbol{W}_{\boldsymbol{A},\boldsymbol{B}}(t))_{ij} = \boldsymbol{a}_i(t)^\top \boldsymbol{b}_j(t),$$

where $\boldsymbol{a}_i(t)^\top$ is the $i$-th row of $\boldsymbol{A}(t)$ and $\boldsymbol{b}_j(t)$ is the $j$-th column of $\boldsymbol{B}(t)$.

Let $\boldsymbol{\theta}(t)$ be the vector obtained by stacking all entries of $\boldsymbol{A}(t)$ and $\boldsymbol{B}(t)$. The gradient of $w_{ij}(t)$ with respect to $\boldsymbol{\theta}(t)$ is given by

$$\nabla_{\boldsymbol{\theta}} w_{ij}(t) = \left( \{\nabla_{\boldsymbol{a}_r} w_{ij}(t)\}_{r=1}^d, \{\nabla_{\boldsymbol{b}_s} w_{ij}(t)\}_{s=1}^d \right),$$

where $\nabla_{\boldsymbol{a}_r}$ and $\nabla_{\boldsymbol{b}_s}$ denote derivatives with respect to the row vector $\boldsymbol{a}_r(t)$ and column vector $\boldsymbol{b}_s(t)$, respectively. Since $w_{ij}(t) = \boldsymbol{a}_i(t)^\top \boldsymbol{b}_j(t)$, we have

$$\nabla_{\boldsymbol{a}_r} w_{ij}(t) = \begin{cases} \boldsymbol{b}_j(t), & r = i, \\ \boldsymbol{0}, & r \neq i, \end{cases} \quad \nabla_{\boldsymbol{b}_s} w_{ij}(t) = \begin{cases} \boldsymbol{a}_i(t), & s = j, \\ \boldsymbol{0}, & s \neq j. \end{cases}$$

Thus $\nabla_{\boldsymbol{\theta}} w_{ij}(t)$ has nonzero components only in the coordinates corresponding to $\boldsymbol{a}_i(t)$ and $\boldsymbol{b}_j(t)$, and all other coordinates are identically zero.

Now fix two observed indices $(i,j) \in \Omega_b$ and $(p,q) \in \Omega_{b'}$ with $b \neq b'$. By the definition of $\Omega_b$, we have

$$i,j \in \{(b-1)s+1,\dots,bs\}, \quad p,q \in \{(b'-1)s+1,\dots,b's\},$$

and these index sets are disjoint. Therefore the supports of the two gradient vectors are disjoint. Hence, for all $t \geq 0$,

$$\langle \nabla_{\boldsymbol{\theta}} w_{ij}(t), \nabla_{\boldsymbol{\theta}} w_{pq}(t) \rangle = 0.$$

Therefore, the gradient flow dynamics are **decoupled** with respect to the partition $\{\Omega_b\}_{b=1}^n$ in the sense of Definition 2.

### D.2.2 Case for $L \geq 3$ and $1 < m < \infty$

For the deeper matrix case ($L \geq 3$) with $1 < m < \infty$, every entry of each weight matrix $\boldsymbol{W}_l(0)$ (for $l = 1,\dots,L$) is initialized to be positive. Then, for any $(i,j) \in \Omega_{\text{block}}^{(s,n)}$, the entry $w_{ij}(0)$ is a sum of products of these positive entries, so $w_{ij}(0) > 0$.

Evaluating (23) and (24) at $t = 0$, the derivative of $w_{ij}(0)$ with respect to any parameter in $\boldsymbol{\theta}$ is given by the product of entries derived in (23). Since $L \geq 3$, there exists at least one intermediate

layer $l \in \{2, \ldots, L-1\}$. For these intermediate layers, both products in the derivative formula are products of matrices with strictly positive entries. Consequently, for $1 < l < L$, every coordinate of $\nabla_{\boldsymbol{W}_l} w_{ij}(0)$ is strictly positive. For the boundary layers ($l = 1$ and $l = L$), the derivatives are non-negative.

Because the gradient contains a strictly positive sub-vector (corresponding to the intermediate layers) and is non-negative everywhere else, for any two distinct observed indices $(i, j)$ and $(p, q)$, their inner product satisfies:

$$\langle \nabla_{\boldsymbol{\theta}} w_{ij}(0), \nabla_{\boldsymbol{\theta}} w_{pq}(0) \rangle > 0.$$

This shows that there is no partition of $\Omega_{\text{block}}^{(s,n)}$ for which the cross-block inner products in (6) vanish at $t = 0$, so by Definition 2 the gradient flow dynamics are **coupled** for $L \geq 3$ and $1 < m < \infty$.

### D.2.3 Case for $L \geq 3$ and $m = \infty$

For $a, b \in \mathbb{R}$, define $\boldsymbol{D}(a, b) \triangleq (a - b)\boldsymbol{I}_s + b\boldsymbol{J}_s$. Lemma D.2 shows that the family

$$\mathcal{M} \triangleq \{\boldsymbol{I}_n \otimes \boldsymbol{D}(a, b) + (\boldsymbol{J}_n - \boldsymbol{I}_n) \otimes c\, \boldsymbol{J}_s\}$$

is closed under scalar multiplication, addition, and matrix multiplication, and that any two matrices in $\mathcal{M}$ commute. As a consequence, Lemma D.3 implies that if the factor matrices $\boldsymbol{W}_l(0)$ are initialized to lie in $\mathcal{M}$, then under the gradient flow dynamics in (3), each $\boldsymbol{W}_l(t)$ remains in the family $\mathcal{M}$ for all $t \geq 0$ and all $l \in [L]$.

The case $c = 0$ follows immediately as a special case. When $c = 0$, the family reduces to

$$\mathcal{M}_0 \triangleq \{\boldsymbol{I}_n \otimes \boldsymbol{D}(a, b)\},$$

and all closure, commutativity, and invariance properties follow directly from Lemma D.1 together with the Kronecker product identity.

We now focus on the setting $m = \infty$. In this case, $\boldsymbol{W}_l(t)$ lies in $\mathcal{M}_0$ for all $t \geq 0$ and all $l \in [L]$. Since every matrix in $\mathcal{M}_0$ is block-diagonal with identical $s \times s$ blocks, it follows that both products $\boldsymbol{W}_L(t) \cdots \boldsymbol{W}_{l+1}(t)$ and $\boldsymbol{W}_{l-1}(t) \cdots \boldsymbol{W}_1(t)$ inherit the same block-diagonal structure. As a result, for a prediction $w_{pq}(t)$ where $(p, q) \in \Omega_b$, the partial derivative in (23) is nonzero only if the parameter indices $i$ and $j$ also belong to the same block $b$ (i.e., $(i, j) \in \Omega_b$).

Now fix two observed indices $(i, j) \in \Omega_b$ and $(p, q) \in \Omega_{b'}$ with $b \neq b'$. Since the corresponding index sets are disjoint, the supports of $\nabla_{\boldsymbol{\theta}} w_{ij}(t)$ and $\nabla_{\boldsymbol{\theta}} w_{pq}(t)$ are disjoint as well. Consequently, for all $t \geq 0$,

$$\langle \nabla_{\boldsymbol{\theta}} w_{ij}(t), \nabla_{\boldsymbol{\theta}} w_{pq}(t) \rangle = 0.$$

This verifies that the gradient flow dynamics are **decoupled** with respect to the partition $\{\Omega_b\}_{b=1}^n$ in the sense of Definition 2.

### D.3 PROOF FOR THEOREM 3.3

For $a, b, c \in \mathbb{R}$, define

$$\boldsymbol{D}(a, b) = (a - b)\boldsymbol{I}_s + b\boldsymbol{J}_s,$$
$$\boldsymbol{O}(c) = c\boldsymbol{J}_s,$$

where $\boldsymbol{I}_s$ is the $s \times s$ identity matrix and $\boldsymbol{J}_s$ is the $s \times s$ all-ones matrix. Consider the $d \times d$ block matrix

$$\boldsymbol{M}(a, b, c) = \boldsymbol{I}_n \otimes \boldsymbol{D}(a, b) + (\boldsymbol{J}_n - \boldsymbol{I}_n) \otimes \boldsymbol{O}(c)$$

$$= \begin{bmatrix} \boldsymbol{D}(a, b) & \boldsymbol{O}(c) & \cdots & \boldsymbol{O}(c) \\ \boldsymbol{O}(c) & \boldsymbol{D}(a, b) & \cdots & \boldsymbol{O}(c) \\ \vdots & \vdots & \ddots & \vdots \\ \boldsymbol{O}(c) & \boldsymbol{O}(c) & \cdots & \boldsymbol{D}(a, b) \end{bmatrix} \in \mathbb{R}^{d \times d},$$

which is an $n \times n$ block matrix with $s \times s$ blocks. Define

$$\mathcal{M} \triangleq \{\boldsymbol{M}(a, b, c) \mid a, b, c \in \mathbb{R}\}.$$

We now state a lemma that captures the key algebraic features of this family.

**Lemma D.1.** *Let $\boldsymbol{I}_n$ denote the $n \times n$ identity matrix and $\boldsymbol{J}_n \triangleq \mathbb{1}_n \mathbb{1}_n^\top$ denote the $n \times n$ matrix with all entries equal to $1$. Then the set*

$$\mathcal{S} = \{a\boldsymbol{I}_n + b\boldsymbol{J}_n \mid a, b \in \mathbb{R}\}$$

*is closed under scalar multiplication, addition, and matrix multiplication. Also, any two matrices $\boldsymbol{A}, \boldsymbol{B} \in \mathcal{S}$ commute.*

*Proof.* Let

$$\boldsymbol{A} = a\boldsymbol{I}_n + b\boldsymbol{J}_n \quad \text{and} \quad \boldsymbol{B} = c\boldsymbol{I}_n + d\boldsymbol{J}_n,$$

with $a, b, c, d \in \mathbb{R}$, and let $\lambda \in \mathbb{R}$ be an arbitrary scalar.

**Scalar Multiplication.**

$$\lambda\boldsymbol{A} = \lambda(a\boldsymbol{I}_n + b\boldsymbol{J}_n) = (\lambda a)\boldsymbol{I}_n + (\lambda b)\boldsymbol{J}_n.$$

Since $\lambda a, \lambda b \in \mathbb{R}$, it follows that $\lambda\boldsymbol{A} \in \mathcal{S}$.

**Addition.**

$$\boldsymbol{A} + \boldsymbol{B} = (a\boldsymbol{I}_n + b\boldsymbol{J}_n) + (c\boldsymbol{I}_n + d\boldsymbol{J}_n) = (a + c)\boldsymbol{I}_n + (b + d)\boldsymbol{J}_n.$$

Since $a + c, \ b + d \in \mathbb{R}$, we have $\boldsymbol{A} + \boldsymbol{B} \in \mathcal{S}$.

**Matrix Multiplication.**

$$\boldsymbol{A}\boldsymbol{B} = (a\boldsymbol{I}_n + b\boldsymbol{J}_n)(c\boldsymbol{I}_n + d\boldsymbol{J}_n).$$

Using the distributive property and the facts that

$$\boldsymbol{I}_n\boldsymbol{J}_n = \boldsymbol{J}_n\boldsymbol{I}_n = \boldsymbol{J}_n \quad \text{and} \quad \boldsymbol{J}_n^2 = n\boldsymbol{J}_n,$$

we expand:

$$\boldsymbol{A}\boldsymbol{B} = ac\,\boldsymbol{I}_n\boldsymbol{I}_n + ad\,\boldsymbol{I}_n\boldsymbol{J}_n + bc\,\boldsymbol{J}_n\boldsymbol{I}_n + bd\,\boldsymbol{J}_n^2$$
$$= ac\,\boldsymbol{I}_n + ad\,\boldsymbol{J}_n + bc\,\boldsymbol{J}_n + bd\,(n\boldsymbol{J}_n)$$
$$= ac\,\boldsymbol{I}_n + (ad + bc + nbd)\boldsymbol{J}_n.$$

Thus, $\boldsymbol{A}\boldsymbol{B}$ is of the form $\alpha\boldsymbol{I}_n + \beta\boldsymbol{J}_n$ with $\alpha = ac$ and $\beta = ad + bc + nbd$, and hence $\boldsymbol{A}\boldsymbol{B} \in \mathcal{S}$.

**Commutativity.** By the same procedure as above,

$$\boldsymbol{A}\boldsymbol{B} = (a\boldsymbol{I}_n + b\boldsymbol{J}_n)(c\boldsymbol{I}_n + d\boldsymbol{J}_n)$$
$$= ac\boldsymbol{I}_n + (ad + bc + nbd)\boldsymbol{J}_n$$
$$= ca\boldsymbol{I}_n + (cb + da + ndb)\boldsymbol{J}_n$$
$$= \boldsymbol{B}\boldsymbol{A},$$

which completes the proof. $\qquad\square$

**Lemma D.2.** *The set $\mathcal{M}$ is closed under scalar multiplication and addition, and it is also closed under matrix multiplication. Moreover, for any $(a_1, b_1, c_1)$ and $(a_2, b_2, c_2)$, the matrices $\boldsymbol{M}(a_1, b_1, c_1)$ and $\boldsymbol{M}(a_2, b_2, c_2)$ commute.*

*Proof.* Note that by Lemma D.1, $\boldsymbol{D}(a, b)$ is closed under scalar multiplication, addition, and matrix multiplication. Since $\boldsymbol{J}_s$ is also closed under these operations, the same holds for $\boldsymbol{O}(c)$.

**Scalar multiplication.** For any scalar $\lambda \in \mathbb{R}$,

$$
\begin{aligned}
\lambda \boldsymbol{M}(a, b, c) &= \lambda \left[ \boldsymbol{I}_n \otimes \boldsymbol{D}(a, b) + (\boldsymbol{J}_n - \boldsymbol{I}_n) \otimes \boldsymbol{O}(c) \right] \\
&= \boldsymbol{I}_n \otimes (\lambda \boldsymbol{D}(a, b)) + (\boldsymbol{J}_n - \boldsymbol{I}_n) \otimes (\lambda \boldsymbol{O}(c)) \\
&= \boldsymbol{I}_n \otimes \boldsymbol{D}(\lambda a, \lambda b) + (\boldsymbol{J}_n - \boldsymbol{I}_n) \otimes \boldsymbol{O}(\lambda c) \\
&= \boldsymbol{M}(\lambda a, \lambda b, \lambda c) \in \mathcal{M}.
\end{aligned}
$$

**Addition.** For any $(a_1, b_1, c_1)$ and $(a_2, b_2, c_2)$,

$$
\begin{aligned}
\boldsymbol{M}(a_1, b_1, c_1) + \boldsymbol{M}(a_2, b_2, c_2) &= \left[ \boldsymbol{I}_n \otimes \boldsymbol{D}(a_1, b_1) + (\boldsymbol{J}_n - \boldsymbol{I}_n) \otimes \boldsymbol{O}(c_1) \right] \\
&\quad + \left[ \boldsymbol{I}_n \otimes \boldsymbol{D}(a_2, b_2) + (\boldsymbol{J}_n - \boldsymbol{I}_n) \otimes \boldsymbol{O}(c_2) \right] \\
&= \boldsymbol{I}_n \otimes (\boldsymbol{D}(a_1, b_1) + \boldsymbol{D}(a_2, b_2)) + (\boldsymbol{J}_n - \boldsymbol{I}_n) \otimes (\boldsymbol{O}(c_1) + \boldsymbol{O}(c_2)) \\
&= \boldsymbol{I}_n \otimes \boldsymbol{D}(a_1 + a_2, b_1 + b_2) + (\boldsymbol{J}_n - \boldsymbol{I}_n) \otimes \boldsymbol{O}(c_1 + c_2) \\
&= \boldsymbol{M}(a_1 + a_2, b_1 + b_2, c_1 + c_2) \in \mathcal{M}.
\end{aligned}
$$

**Matrix multiplication.** First observe that

$$
\begin{aligned}
\boldsymbol{D}(a_1, b_1) \boldsymbol{D}(a_2, b_2) &= \boldsymbol{D}(a_1 a_2 + (s-1) b_1 b_2,\ a_1 b_2 + a_2 b_1 + (s-2) b_1 b_2), \\
\boldsymbol{O}(c_1) \boldsymbol{O}(c_2) &= \boldsymbol{O}(s c_1 c_2), \\
\boldsymbol{D}(a, b) \boldsymbol{O}(c) &= \boldsymbol{O}(c) \boldsymbol{D}(a, b) = \boldsymbol{O}(ac + (s-1) bc).
\end{aligned}
$$

Multiplying $\boldsymbol{M}(a_1, b_1, c_1)$ and $\boldsymbol{M}(a_2, b_2, c_2)$ gives

$$
\boldsymbol{M}(a_1, b_1, c_1) \boldsymbol{M}(a_2, b_2, c_2) = \begin{bmatrix} \boldsymbol{T}_1 & \boldsymbol{T}_2 & \cdots & \boldsymbol{T}_2 \\ \boldsymbol{T}_2 & \boldsymbol{T}_1 & \cdots & \boldsymbol{T}_2 \\ \vdots & \vdots & \ddots & \vdots \\ \boldsymbol{T}_2 & \boldsymbol{T}_2 & \cdots & \boldsymbol{T}_1 \end{bmatrix},
$$

where

$$
\begin{aligned}
\boldsymbol{T}_1 &= \boldsymbol{D}(a_1, b_1) \boldsymbol{D}(a_2, b_2) + (n-1) \boldsymbol{O}(c_1) \boldsymbol{O}(c_2), \\
\boldsymbol{T}_2 &= \boldsymbol{O}(c_1) \boldsymbol{D}(a_2, b_2) + \boldsymbol{D}(a_1, b_1) \boldsymbol{O}(c_2) + (n-2) \boldsymbol{O}(c_1) \boldsymbol{O}(c_2).
\end{aligned}
$$

Using the identities above, we can rewrite $\boldsymbol{T}_1$ and $\boldsymbol{T}_2$ as

$$
\begin{aligned}
\boldsymbol{T}_1 &= \boldsymbol{D}(a_1 a_2 + (s-1) b_1 b_2,\ a_1 b_2 + a_2 b_1 + (s-2) b_1 b_2) + \boldsymbol{O}((n-1) s c_1 c_2) \\
&= \boldsymbol{D}\big(a_1 a_2 + (s-1) b_1 b_2 + (n-1) s c_1 c_2,\ a_1 b_2 + a_2 b_1 + (s-2) b_1 b_2 + (n-1) s c_1 c_2\big), \\
\boldsymbol{T}_2 &= \boldsymbol{O}(a_2 c_1 + (s-1) b_2 c_1) + \boldsymbol{O}(a_1 c_2 + (s-1) b_1 c_2) + \boldsymbol{O}((n-2) s c_1 c_2) \\
&= \boldsymbol{O}\big(a_1 c_2 + a_2 c_1 + (s-1) b_1 c_2 + (s-1) b_2 c_1 + (n-2) s c_1 c_2\big).
\end{aligned}
$$

Hence $\boldsymbol{M}(a_1, b_1, c_1) \boldsymbol{M}(a_2, b_2, c_2)$ again has the same block structure as $\boldsymbol{M}(\cdot, \cdot, \cdot)$, so $\mathcal{M}$ is closed under matrix multiplication.

**Commutativity.** The expressions for $\boldsymbol{T}_1$ and $\boldsymbol{T}_2$ above are symmetric in $(a_1, b_1, c_1)$ and $(a_2, b_2, c_2)$. In particular, if we interchange $(a_1, b_1, c_1)$ and $(a_2, b_2, c_2)$ in the formulas for $\boldsymbol{T}_1$ and $\boldsymbol{T}_2$, we obtain the same matrices. Therefore

$$
\boldsymbol{M}(a_1, b_1, c_1) \boldsymbol{M}(a_2, b_2, c_2) = \boldsymbol{M}(a_2, b_2, c_2) \boldsymbol{M}(a_1, b_1, c_1),
$$

and the matrices in $\mathcal{M}$ commute pairwise. $\qquad\square$

Using the above lemma, we show that if all factor matrices $\boldsymbol{W}_l$ are initialized according to (7), then $\boldsymbol{W}_l(t)$ stays in $\mathcal{M}$ for every $t \geq 0$.

**Lemma D.3.** *Let $s, n \in \mathbb{N}$ and set $d = sn$. Consider a ground truth matrix $\boldsymbol{W}^* \in \mathbb{R}^{d \times d}$ with observation set $\Omega_{\text{block}}^{(s,n)}$ with all observed entries sharing the same positive value $w^* > 0$. Consider the product matrix $\boldsymbol{W}_{L:1}$, where the factor matrices $\boldsymbol{W}_l \in \mathbb{R}^{d \times d}$ are initialized according to (7). Under the gradient flow dynamics (3), $\boldsymbol{W}_l(t)$ remains in the family $\mathcal{M}$ for all $t \geq 0$ and all $l \in [L]$.*

*Proof.* First note that the initialization in (7) belongs to the family $\mathcal{M}$, since each factor is of the form

$$\boldsymbol{W}_l(0) = \boldsymbol{M}(\alpha, \alpha/m, \alpha/m), \quad l \in [L].$$

We will show that $\mathcal{M}$ is invariant under the gradient flow.

Fix any time $t \geq 0$ and assume that $\boldsymbol{W}_l(t) \in \mathcal{M}$ for all $l \in [L]$. By Lemma D.2, $\mathcal{M}$ is closed under matrix multiplication and every matrix in $\mathcal{M}$ is symmetric by construction, so it is also closed under transpose. Hence the product matrix $\boldsymbol{W}_{L:1}(t) = \boldsymbol{W}_L(t) \cdots \boldsymbol{W}_1(t)$ lies in $\mathcal{M}$. In particular, there exist scalars $A, B, C \in \mathbb{R}$ such that $\boldsymbol{W}_{L:1}(t) = \boldsymbol{M}(A, B, C)$.

By the definition of the observation set $\Omega$ and the assumption that all observed entries share the same ground-truth value $w^*$, the loss has the form

$$\ell(\boldsymbol{W}_{L:1}) = \frac{1}{2} \sum_{(i,j) \in \Omega} \left((\boldsymbol{W}_{L:1})_{ij} - w^*\right)^2.$$

Since $\Omega$ contains exactly the entries inside each diagonal block, and $\boldsymbol{W}_{L:1}(t) = \boldsymbol{M}(A, B, C)$ has diagonal blocks with diagonal entries $A$ and off-diagonal entries $B$, a direct computation gives

$$\nabla\ell(\boldsymbol{W}_{L:1}(t)) = \boldsymbol{M}(A - w^*, B - w^*, 0) \in \mathcal{M}.$$

The gradient flow dynamics for each factor matrix are

$$\dot{\boldsymbol{W}}_l(t) = -\left(\prod_{i=l+1}^{L} \boldsymbol{W}_i(t)^\top\right) \nabla\ell(\boldsymbol{W}_{L:1}(t)) \left(\prod_{i=1}^{l-1} \boldsymbol{W}_i(t)^\top\right), \quad l \in [L].$$

Each factor in the products on the right-hand side belongs to $\mathcal{M}$, and by Lemma D.2 the product of matrices in $\mathcal{M}$ remains in $\mathcal{M}$. Since $\nabla\ell(\boldsymbol{W}_{L:1}(t)) \in \mathcal{M}$ as well, it follows that

$$\dot{\boldsymbol{W}}_l(t) \in \mathcal{M} \quad \text{for all } l \in [L].$$

Since the initial condition satisfies $\boldsymbol{W}_l(0) \in \mathcal{M}$ for all $l \in [L]$, we conclude that

$$\boldsymbol{W}_l(t) \in \mathcal{M} \quad \text{for all } t \geq 0, l \in [L].$$

$\square$

Beyond showing that every factor matrix remains in the family $\mathcal{M}$, we further establish that all layers evolve identically with below lemma:

**Lemma D.4.** *Under the setting of Lemma D.3,*

$$\boldsymbol{W}_L(t) = \boldsymbol{W}_{L-1}(t) = \cdots = \boldsymbol{W}_1(t)$$

*holds for all $t \geq 0$.*

*Proof.* By Lemma D.3 and Lemma D.2, we know that for all $t \geq 0$ and all $l \in [L]$ we have $\boldsymbol{W}_l(t) \in \mathcal{M}$, and that matrices in $\mathcal{M}$ are closed under matrix multiplication, transpose, and commute pairwise. Moreover, as shown in the proof of Lemma D.3, the loss gradient $\nabla\ell(\boldsymbol{W}_{L:1}(t))$ also lies in $\mathcal{M}$.

Fix any time $t$ and suppose that

$$\boldsymbol{W}_L(t) = \boldsymbol{W}_{L-1}(t) = \cdots = \boldsymbol{W}_1(t) =: \boldsymbol{U}(t).$$

Then the product matrix satisfies $\boldsymbol{W}_{L:1}(t) = \boldsymbol{U}(t)^L$, and the gradient flow dynamics for each layer can be written as

$$\dot{\boldsymbol{W}}_l(t) = -\left(\prod_{i=l+1}^{L} \boldsymbol{W}_i(t)^\top\right) \nabla\ell(\boldsymbol{W}_{L:1}(t)) \left(\prod_{i=1}^{l-1} \boldsymbol{W}_i(t)^\top\right)$$

$$= -\boldsymbol{U}(t)^{L-l} \nabla\ell\left(\boldsymbol{U}(t)^L\right) \boldsymbol{U}(t)^{l-1}.$$

Since $\boldsymbol{U}(t)$ and $\nabla \ell(\boldsymbol{U}(t)^L)$ both lie in $\mathcal{M}$ and matrices in $\mathcal{M}$ commute pairwise, we can reorder the factors to obtain

$$\dot{\boldsymbol{W}}_l(t) = -\nabla \ell \left(\boldsymbol{U}(t)^L\right) \boldsymbol{U}(t)^{L-1} \quad \text{for all } l \in [L].$$

Thus, whenever $\boldsymbol{W}_1(t) = \cdots = \boldsymbol{W}_L(t)$ holds at some time $t$, the time derivatives of all layers coincide at that time:

$$\dot{\boldsymbol{W}}_L(t) = \dot{\boldsymbol{W}}_{L-1}(t) = \cdots = \dot{\boldsymbol{W}}_1(t).$$

By the initialization scheme (7) we have

$$\boldsymbol{W}_L(0) = \boldsymbol{W}_{L-1}(0) = \cdots = \boldsymbol{W}_1(0).$$

Since the gradient flow admits a unique solution for this initial condition, it follows that the equalities between the layers are preserved for all times $t \geq 0$, that is,

$$\boldsymbol{W}_L(t) = \boldsymbol{W}_{L-1}(t) = \cdots = \boldsymbol{W}_1(t) \quad \text{for all } t \geq 0.$$

$\square$

Using the lemma above, we can parameterize every factor matrix as $\boldsymbol{W}_l(t) = \boldsymbol{M}(a(t), b(t), c(t))$ for all $l \in [L]$, where $(a(t), b(t), c(t))$ are shared coefficients. Likewise, we write the product matrix as $\boldsymbol{W}_{L:1}(t) = \boldsymbol{M}(A(t), B(t), C(t))$. We now derive the eigenvalues of each factor matrix.

**Lemma D.5.** *Let $s, n \in \mathbb{N}$ and $d = sn$. For $a, b, c \in \mathbb{R}$, let $\boldsymbol{M}(a, b, c) \in \mathbb{R}^{d \times d}$ be the block matrix defined by*

$$\boldsymbol{M}(a, b, c) = \boldsymbol{I}_n \otimes \boldsymbol{D}(a, b) + (\boldsymbol{J}_n - \boldsymbol{I}_n) \otimes \boldsymbol{O}(c),$$

*where $\boldsymbol{D}(a, b) = (a - b)\boldsymbol{I}_s + b\boldsymbol{J}_s$ and $\boldsymbol{O}(c) = c\boldsymbol{J}_s$. The eigenvalues of $\boldsymbol{M}(a, b, c)$ and their corresponding multiplicities are:*

$$\lambda_1 = a + (s-1)b + s(n-1)c \text{ with multiplicity } 1,$$
$$\lambda_2 = a + (s-1)b - sc \text{ with multiplicity } n-1,$$
$$\lambda_3 = a - b \text{ with multiplicity } n(s-1).$$

*Proof.* First, we express $\boldsymbol{M}(a, b, c)$ in terms of Kronecker products of identity matrices $\boldsymbol{I}$ and all-ones matrices $\boldsymbol{J}$. Substituting the definitions of $\boldsymbol{D}$ and $\boldsymbol{O}$:

$$\boldsymbol{M} = \boldsymbol{I}_n \otimes ((a - b)\boldsymbol{I}_s + b\boldsymbol{J}_s) + (\boldsymbol{J}_n - \boldsymbol{I}_n) \otimes (c\boldsymbol{J}_s)$$
$$= (a - b)(\boldsymbol{I}_n \otimes \boldsymbol{I}_s) + b(\boldsymbol{I}_n \otimes \boldsymbol{J}_s) + c(\boldsymbol{J}_n \otimes \boldsymbol{J}_s) - c(\boldsymbol{I}_n \otimes \boldsymbol{J}_s)$$
$$= (a - b)(\boldsymbol{I}_n \otimes \boldsymbol{I}_s) + (b - c)(\boldsymbol{I}_n \otimes \boldsymbol{J}_s) + c(\boldsymbol{J}_n \otimes \boldsymbol{J}_s).$$

The matrix $\boldsymbol{J}_n$ has two distinct eigenvalues: $n$ (corresponding to eigenvector $\mathbb{1}_n$) and $0$ (corresponding to the orthogonal complement $\mathbb{1}_n^\perp$). We construct the eigenbasis of $\boldsymbol{M}$ using tensor products of the eigenvectors of $\boldsymbol{J}_n$ and $\boldsymbol{J}_s$.

**Case 1.** Consider the eigenvector $\boldsymbol{v}_1 = \mathbb{1}_n \otimes \mathbb{1}_s$. Since $\boldsymbol{J}_n \mathbb{1}_n = n\mathbb{1}_n$ and $\boldsymbol{J}_s \mathbb{1}_s = s\mathbb{1}_s$, we have:

$$\boldsymbol{M}\boldsymbol{v}_1 = ((a - b) + (b - c)s + c(ns)) \, \boldsymbol{v}_1$$
$$= (a + (s-1)b + s(n-1)c) \, \boldsymbol{v}_1.$$

This subspace has dimension $1 \times 1 = 1$.

**Case 2.** Consider eigenvectors $\boldsymbol{v}_2 = \boldsymbol{u} \otimes \mathbb{1}_s$, where $\boldsymbol{u} \in \mathbb{1}_n^\perp \subset \mathbb{R}^n$. Here $\boldsymbol{J}_n \boldsymbol{u} = \boldsymbol{0}$ and $\boldsymbol{J}_s \mathbb{1}_s = s\mathbb{1}_s$. Thus:

$$\boldsymbol{M}\boldsymbol{v}_2 = ((a - b) + (b - c)s + c(0 \cdot s)) \, \boldsymbol{v}_2$$
$$= (a + (s-1)b - sc) \, \boldsymbol{v}_2.$$

The dimension of $\mathbb{1}_n^\perp$ is $n - 1$, so the multiplicity is $(n - 1) \times 1 = n - 1$.

**Case 3.** Consider eigenvectors $\boldsymbol{v}_3 = \boldsymbol{w} \otimes \boldsymbol{z}$, where $\boldsymbol{w} \in \mathbb{R}^n$ is arbitrary and $\boldsymbol{z} \in \mathbb{1}_s^\perp \subset \mathbb{R}^s$. Here $\boldsymbol{J}_s \boldsymbol{z} = \boldsymbol{0}$. Consequently, any term containing $\boldsymbol{J}_s$ in the Kronecker product sends this vector to zero:

$$(\boldsymbol{A} \otimes \boldsymbol{J}_s)(\boldsymbol{w} \otimes \boldsymbol{z}) = \boldsymbol{A}\boldsymbol{w} \otimes \boldsymbol{J}_s \boldsymbol{z} = \boldsymbol{A}\boldsymbol{w} \otimes \boldsymbol{0} = \boldsymbol{0}.$$

Therefore, only the identity term remains:

$$\boldsymbol{M}\boldsymbol{v}_3 = (a - b)\boldsymbol{I}_{ns}\boldsymbol{v}_3 + \boldsymbol{0} + \boldsymbol{0}$$
$$= (a - b)\boldsymbol{v}_3.$$

The dimension of $\mathbb{R}^n$ is $n$ and the dimension of $\mathbb{1}_s^\perp$ is $s - 1$. Thus, the multiplicity is $n(s - 1)$. $\square$

**Lemma D.6.** *Let $\lambda_i(t)$ for $i \in \{1, 2, 3\}$ denote the eigenvalues of the factor matrix $\boldsymbol{W}_l(t)$ from Lemma D.5. Under gradient flow (3), the evolution of these eigenvalues is governed by the following system of ODE:*

$$\dot{\lambda}_1(t) = -\left(\frac{\lambda_1^L(t) + (n-1)\lambda_2^L(t)}{n} - sw^*\right)\lambda_1^{L-1}(t),$$

$$\dot{\lambda}_2(t) = -\left(\frac{\lambda_1^L(t) + (n-1)\lambda_2^L(t)}{n} - sw^*\right)\lambda_2^{L-1}(t),$$

$$\dot{\lambda}_3(t) = -\lambda_3^{2L-1}(t).$$

*Proof.* Given that the factor matrices $\boldsymbol{W}_l(t)$ share the same form (Lemma D.4), let $\lambda_i(t)$ denote their eigenvalues. Consequently, the eigenvalues of the product matrix $\boldsymbol{W}_{L:1}(t)$ are $\lambda_i^L(t)$. Using Lemma D.5 to invert the eigenvalue relations, we can express the parameters of $\boldsymbol{W}_{L:1} = \boldsymbol{M}(A, B, C)$ as follows:

$$A = \frac{\lambda_1^L + (n-1)\lambda_2^L + n(s-1)\lambda_3^L}{sn}.$$

$$B = \frac{\lambda_1^L + (n-1)\lambda_2^L - n\lambda_3^L}{sn},$$

$$C = \frac{\lambda_1^L - \lambda_2^L}{sn}.$$

Recall from the proof of Lemma D.3 that the gradient takes the form $\nabla\ell(\boldsymbol{W}_{L:1}) = \boldsymbol{M}(A - w^*, B - w^*, 0)$. Let $\gamma_i$ denote the eigenvalue of $\nabla\ell(\boldsymbol{W}_{L:1})$ corresponding to the $i$-th eigenvalue class in Lemma D.5. Note that for the gradient matrix, the off-diagonal block parameter is zero ($c = 0$). Consequently, the eigenvalues for $\gamma_1$ and $\gamma_2$ coincide. Specifically:

$$\gamma_1 = (A - w^*) + (s-1)(B - w^*) + s(n-1)(0)$$
$$= (A - w^*) + (s-1)(B - w^*),$$
$$\gamma_2 = (A - w^*) + (s-1)(B - w^*) - s(0)$$
$$= \gamma_1,$$
$$\gamma_3 = (A - w^*) - (B - w^*).$$

Substituting the expressions for $A$ and $B$ into the equations above yields $\gamma_i$ in terms of $\lambda_i^L$:

$$\gamma_1 = \gamma_2 = \left(\frac{\lambda_1^L + (n-1)\lambda_2^L + n(s-1)\lambda_3^L}{sn} - w^*\right) + (s-1)\left(\frac{\lambda_1^L + (n-1)\lambda_2^L - n\lambda_3^L}{sn} - w^*\right)$$

$$= \frac{\lambda_1^L + (n-1)\lambda_2^L}{n} - sw^*,$$

$$\gamma_3 = \left(\frac{\lambda_1^L + (n-1)\lambda_2^L + n(s-1)\lambda_3^L}{sn} - w^*\right) - \left(\frac{\lambda_1^L + (n-1)\lambda_2^L - n\lambda_3^L}{sn} - w^*\right)$$

$$= \lambda_3^L.$$

Finally, recall that the gradient flow dynamics for each layer are governed by

$$\dot{\boldsymbol{W}}_l(t) = -\left(\prod_{j=l+1}^{L}\boldsymbol{W}_j(t)^\top\right)\nabla\ell(\boldsymbol{W}_{L:1}(t))\left(\prod_{j=1}^{l-1}\boldsymbol{W}_j(t)^\top\right).$$

Since the weight matrices $\boldsymbol{W}_l(t)$ and the gradient matrix $\nabla\ell(\boldsymbol{W}_{L:1}(t))$ belong to $\mathcal{M}$, they are commutative and simultaneously diagonalizable. Let $\boldsymbol{P} \in \mathbb{R}^{d \times d}$ be the time-independent common orthogonal matrix such that $\boldsymbol{W}_l(t) = \boldsymbol{P}\Lambda(t)\boldsymbol{P}^\top$ and $\nabla\ell(\boldsymbol{W}_{L:1}(t)) = \boldsymbol{P}\Gamma(t)\boldsymbol{P}^\top$, where $\Lambda(t)$ and $\Gamma(t)$ are diagonal matrices containing the eigenvalues $\lambda_i(t)$ and $\gamma_i(t)$, respectively.

Projecting the gradient flow dynamics onto the eigenspace spanned by the $i$-th eigenvector, we obtain the evolution of the eigenvalues. Using the fact that $\boldsymbol{W}_l(t)^\top = \boldsymbol{W}_l(t)$ due to symmetry, the dynamics

for the $l$-th layer become:

$$\dot{\boldsymbol{W}}_l(t) = \boldsymbol{P}\dot{\Lambda}(t)\boldsymbol{P}^\top = -\left(\boldsymbol{P}\Lambda(t)\boldsymbol{P}^\top\right)^{L-l}\left(\boldsymbol{P}\Gamma(t)\boldsymbol{P}^\top\right)\left(\boldsymbol{P}\Lambda(t)\boldsymbol{P}^\top\right)^{l-1}$$
$$= -\boldsymbol{P}\left(\Lambda^{L-l}(t)\Gamma(t)\Lambda^{l-1}(t)\right)\boldsymbol{P}^\top.$$

Multiplying by $\boldsymbol{P}^\top$ on the left and $\boldsymbol{P}$ on the right yields the diagonal evolution:

$$\dot{\Lambda}(t) = -\Gamma(t)\Lambda^{L-1}(t).$$

For each distinct eigenvalue index $i \in \{1, 2, 3\}$, the scalar dynamics simplify to:

$$\dot{\lambda}_i(t) = -\gamma_i(t)\lambda_i^{L-1}(t).$$

Substituting the values of $\gamma_i$ derived previously, we obtain the specific evolution equations for each eigenvalue:

$$\dot{\lambda}_1(t) = -\gamma_1(t)\lambda_1^{L-1}(t) = -\left(\frac{\lambda_1^L(t) + (n-1)\lambda_2^L(t)}{n} - sw^*\right)\lambda_1^{L-1}(t),$$

$$\dot{\lambda}_2(t) = -\gamma_2(t)\lambda_2^{L-1}(t) = -\left(\frac{\lambda_1^L(t) + (n-1)\lambda_2^L(t)}{n} - sw^*\right)\lambda_2^{L-1}(t),$$

$$\dot{\lambda}_3(t) = -\gamma_3(t)\lambda_3^{L-1}(t) = -\lambda_3^{2L-1}(t).$$

$\square$

Building on the lemma above, we can identify a conserved quantity that depends on the depth.

**Lemma D.7.** *Under the gradient flow dynamics defined in Lemma D.6, the eigenvalues $\lambda_1(t)$ and $\lambda_2(t)$ satisfy the following conservation laws for all $t \geq 0$:*

  *1. If $L = 2$, the ratio of the eigenvalues is conserved:*
$$\frac{\lambda_1(t)}{\lambda_2(t)} = \frac{\lambda_1(0)}{\lambda_2(0)}.$$

  *2. If $L \geq 3$, the difference of the negated powers is conserved:*
$$\lambda_1^{2-L}(t) - \lambda_2^{2-L}(t) = \lambda_1^{2-L}(0) - \lambda_2^{2-L}(0).$$

*Proof.* From Lemma D.6, the scalar dynamics for the first two eigenvalues are given by:

$$\dot{\lambda}_i(t) = -\gamma(t)\lambda_i^{L-1}(t) \quad \text{for } i \in \{1, 2\},$$

where $\gamma(t) = \frac{\lambda_1(t)^L + (n-1)\lambda_2(t)^L}{n} - sw^*$. We consider the two cases based on the depth $L$.

**Case 1:** ($L = 2$). In this case, the dynamics simplify to $\dot{\lambda}_i(t) = -\gamma(t)\lambda_i(t)$. Rearranging the terms to separate variables, we have:

$$\frac{\dot{\lambda}_1(t)}{\lambda_1(t)} = -\gamma(t), \quad \frac{\dot{\lambda}_2(t)}{\lambda_2(t)} = -\gamma(t).$$

Subtracting the second equation from the first eliminates $\gamma(t)$:

$$\frac{d}{dt}\log|\lambda_1(t)| - \frac{d}{dt}\log|\lambda_2(t)| = 0$$
$$\frac{d}{dt}\log\left|\frac{\lambda_1(t)}{\lambda_2(t)}\right| = 0.$$

This implies that the ratio $\lambda_1(t)/\lambda_2(t)$ is constant in time.

**Case 2:** ($L \geq 3$). Consider the time derivative of the quantity $Q(t) = \lambda_1^{2-L}(t) - \lambda_2^{2-L}(t)$. Applying the chain rule:

$$\frac{d}{dt}\left(\lambda_1^{2-L}(t)\right) = (2-L)\lambda_1^{1-L}(t) \cdot \dot{\lambda}_1(t)$$
$$= (2-L)\lambda_1^{1-L}(t) \cdot \left(-\gamma(t)\lambda_1^{L-1}(t)\right)$$
$$= -(2-L)\gamma(t).$$

Similarly, for the second term:

$$\frac{d}{dt}\left(\lambda_2^{2-L}(t)\right) = (2-L)\lambda_2^{1-L}(t) \cdot \left(-\gamma(t)\lambda_2^{L-1}(t)\right)$$
$$= -(2-L)\gamma(t).$$

Subtracting the two derivatives yields:

$$\frac{d}{dt}\left(\lambda_1^{2-L}(t) - \lambda_2^{2-L}(t)\right) = (-(2-L)\gamma(t)) - (-(2-L)\gamma(t)) = 0.$$

Since the time derivative is zero, the quantity is conserved throughout the training, proving the statement. $\qquad\square$

We are now ready to prove Theorem 3.3.

*Proof.* Using the inverse relations from Lemma D.5, we express the parameters $A(t)$ and $B(t)$ in terms of the eigenvalues:

$$A(t) = \frac{\lambda_1^L(t) + (n-1)\lambda_2^L(t) + n(s-1)\lambda_3^L(t)}{sn},$$
$$B(t) = \frac{\lambda_1^L(t) + (n-1)\lambda_2^L(t) - n\lambda_3^L(t)}{sn}.$$

Consider the difference between the parameters:

$$A(t) - B(t) = \frac{ns\lambda_3^L(t)}{sn} = \lambda_3^L(t).$$

Since loss converges to zero (Proposition D.1), implies global optimality, which requires $A(\infty) = B(\infty) = w^*$. Taking the limit $t \to \infty$, the difference vanishes, yielding:

$$\lambda_3(\infty) = 0.$$

Next, substituting $\lambda_3(\infty) = 0$ and $A(\infty) = w^*$ into the expression for $A(t)$, we obtain:

$$w^* = \frac{\lambda_1^L(\infty) + (n-1)\lambda_2^L(\infty)}{sn}.$$

Multiplying by $sn = d$, we arrive at the first constraint:

$$\lambda_1^L(\infty) + (n-1)\lambda_2^L(\infty) = dw^*. \tag{25}$$

Let $\sigma_1 \geq \sigma_2 \geq \cdots \geq \sigma_d$ denote the singular values of the limiting product matrix $\boldsymbol{W}_{L:1}(\infty)$. Under our initialization scheme and Lemma F.3, the factor matrices remain positive definite, implying that the singular values of the product matrix coincide with the $L$-th power of the eigenvalues. Based on the multiplicities derived in Lemma D.5, we identify:

$$\sigma_1 = \lambda_1^L(\infty), \; \sigma_i = \lambda_2^L(\infty) \text{ for } i \in \{2, \ldots, n\}, \quad \sigma_j = \lambda_3^L(\infty) = 0 \text{ for } j > n.$$

We now solve for the non-zero singular values by considering two cases based on the depth $L$.

**Case 1:** ($L = 2$). For $L = 2$, Lemma D.7 states that the ratio of eigenvalues is preserved. Using the initialization values from (7) and Lemma D.5, this ratio is given by:

$$\frac{\lambda_1(\infty)}{\lambda_2(\infty)} = \frac{\lambda_1(0)}{\lambda_2(0)} = \frac{m+d-1}{m-1}. \tag{26}$$

Substituting $\lambda_i^2(\infty) = \sigma_i$ into (25) and combining it with the squared ratio from (26), we can solve for $\sigma_1$ and $\sigma_i$:

$$\sigma_1 = \frac{w^* d(m+d-1)^2}{(m+d-1)^2 + (n-1)(m-1)^2},$$
$$\sigma_i = \frac{w^* d(m-1)^2}{(m+d-1)^2 + (n-1)(m-1)^2} \quad \text{for all } i \in \{2, \ldots, n\}.$$

**Case 2: ($L \geq 3$ and finite $m$).** For $L \geq 3$ with $1 < m < \infty$, Lemma D.7 ensures the conservation of the difference of negated powers:

$$\lambda_1^{2-L}(\infty) - \lambda_2^{2-L}(\infty) = \lambda_1^{2-L}(0) - \lambda_2^{2-L}(0).$$

Substituting the initial eigenvalues from (7), the right-hand side becomes:

$$\lambda_1^{2-L}(\infty) - \lambda_2^{2-L}(\infty) = \left(\frac{\alpha}{m}\right)^{2-L}\left((m+d-1)^{2-L} - (m-1)^{2-L}\right). \tag{27}$$

Finally, expressing the eigenvalues in terms of singular values via $\lambda_i(\infty) = \sigma_i^{1/L}$ (implying $\lambda_i^{2-L} = \sigma_i^{\frac{2-L}{L}}$) and combining (25) with (27), we obtain the system of implicit equations:

$$\sigma_1^{\frac{2-L}{L}} - \left(\frac{w^* d - \sigma_1}{n-1}\right)^{\frac{2-L}{L}} = C_{\alpha,m,L,d},$$

$$(w^* d - (n-1)\sigma_i)^{\frac{2-L}{L}} - \sigma_i^{\frac{2-L}{L}} = C_{\alpha,m,L,d} \quad \text{for all } i \in \{2,\ldots,n\},$$

where $C_{\alpha,m,L,d} \triangleq \left(\frac{\alpha}{m}\right)^{2-L}\left((m+d-1)^{2-L} - (m-1)^{2-L}\right).$

**Case 3: ($L \geq 3$ and $m = \infty$).** In this case, the initial eigenvalues of the factor matrices become:

$$\lambda_1(0) = \lim_{m \to \infty} \alpha\left(1 + \frac{d-1}{m}\right) = \alpha,$$

$$\lambda_2(0) = \lim_{m \to \infty} \alpha\left(1 - \frac{1}{m}\right) = \alpha.$$

Since the initial eigenvalues are identical, i.e., $\lambda_1(0) = \lambda_2(0)$, the conserved quantities derived in Lemma D.7 dictate that the limiting values must also be identical. When $L \geq 3$, the conservation law states:

$$\lambda_1^{2-L}(\infty) - \lambda_2^{2-L}(\infty) = \lambda_1^{2-L}(0) - \lambda_2^{2-L}(0) = \alpha^{2-L} - \alpha^{2-L} = 0.$$

This implies $\lambda_1(\infty) = \lambda_2(\infty)$. Consequently, the singular values of the product matrix satisfy $\sigma_1 = \sigma_i$ for all $i \in \{2,\ldots,n\}$.

In the case where $L = 2$, the conservation law states:

$$\frac{\lambda_1(\infty)}{\lambda_2(\infty)} = \frac{\lambda_1(0)}{\lambda_2(0)} = \frac{\alpha}{\alpha} = 1.$$

This also implies $\lambda_1(\infty) = \lambda_2(\infty)$ and thus $\sigma_1 = \sigma_i$ for all $i \in \{2,\ldots,n\}$. Consequently, by (25), we have $\sigma_1 = \sigma_i = sw^*$. $\qquad \square$

### D.4 LOSS CONVERGENCE

We further establish loss convergence in the following proposition.

**Proposition D.1.** *Under the setting of Theorem 3.3, suppose the factor matrices are initialized according to (7). Then, under the gradient flow dynamics (3), the loss converges to zero:*

$$\lim_{t \to \infty} \ell(\boldsymbol{W}_{L:1}(t); \Omega_{\text{block}}^{(s,n)}) = 0.$$

*Proof.* According to Lemma D.6, the dynamics of the third eigenvalue are governed by $\dot{\lambda}_3(t) = -\lambda_3^{2L-1}(t)$. For $L \geq 2$, this is a separable ODE whose explicit solution is given by

$$\lambda_3(t) = \left( (2L-2)t + \lambda_3^{-(2L-2)}(0) \right)^{-\frac{1}{2L-2}}.$$

As $t \to \infty$, the term inside the parenthesis grows without bound, which directly implies $\lambda_3(\infty) = 0$.

Next, we consider the global error term

$$\gamma(t) \triangleq \frac{\lambda_1^L(t) + (n-1)\lambda_2^L(t)}{n} - sw^*. \tag{28}$$

By substituting the dynamics from Lemma D.6, we obtain its time derivative:

$$\begin{aligned}
\dot{\gamma}(t) &= \frac{L}{n} \left( \lambda_1^{L-1}(t)\dot{\lambda}_1(t) + (n-1)\lambda_2^{L-1}(t)\dot{\lambda}_2(t) \right) \\
&= -\underbrace{\frac{L}{n} \left( \lambda_1^{2L-2}(t) + (n-1)\lambda_2^{2L-2}(t) \right)}_{\triangleq K(t)} \gamma(t).
\end{aligned}$$

The solution to this linear ODE is $\gamma(t) = \gamma(0) \exp\left( -\int_0^t K(\tau)d\tau \right)$, which implies that $\gamma(t)$ preserves its initial sign for all $t \geq 0$. Since $L \geq 2$, the map $f(x) = x^{\frac{2L-2}{L}}$ is convex on $\mathbb{R}_+$. By Jensen's inequality, we can lower bound $K(t)$ for any fixed $t$:

$$\begin{aligned}
K(t) &= L \left( \frac{(\lambda_1^L(t))^{\frac{2L-2}{L}} + (n-1)(\lambda_2^L(t))^{\frac{2L-2}{L}}}{n} \right) \\
&\geq L \left( \frac{\lambda_1^L(t) + (n-1)\lambda_2^L(t)}{n} \right)^{\frac{2L-2}{L}}. \tag{29}
\end{aligned}$$

**Case 1** ($\gamma(0) \leq 0$). Suppose the initialization scale $\alpha$ satisfies

$$0 < \alpha^L \leq \frac{w^* d m^L}{(m+d-1)^L + (n-1)(m-1)^L}.$$

This ensures $\gamma(0) \leq 0$, and consequently $\gamma(t) \leq 0$ for all $t \geq 0$. From $\dot{\lambda}_1(t) = -\gamma(t)\lambda_1^{L-1}(t) \geq 0$, it follows that $\lambda_1(t) \geq \lambda_1(0) > 0$. Similarly, $\lambda_2(t) \geq \lambda_2(0) > 0$. Therefore, $K(t)$ is strictly lower bounded by its initial value:

$$K(t) \geq K(0) = \frac{L}{n} \left( \lambda_1^{2L-2}(0) + (n-1)\lambda_2^{2L-2}(0) \right) > 0.$$

**Case 2** ($\gamma(0) > 0$). Suppose the initialization scale $\alpha$ satisfies

$$\alpha^L > \frac{w^* d m^L}{(m+d-1)^L + (n-1)(m-1)^L}$$

so that $\gamma(0) > 0$, which implies $\gamma(t) > 0$ for all $t \geq 0$. Using the lower bound from (29) and the fact that $\frac{\lambda_1^L(t) + (n-1)\lambda_2^L(t)}{n} = \gamma(t) + sw^*$, we have:

$$K(t) \geq L \left( \gamma(t) + sw^* \right)^{\frac{2L-2}{L}}.$$

Since $\gamma(t) > 0$ for all $t \geq 0$, we can further lower bound $K(t)$ by a constant:

$$K(t) > L(sw^*)^{\frac{2L-2}{L}} > 0.$$

In both cases, there exists a uniform lower bound $K_{\min} > 0$ such that $K(t) \geq K_{\min}$ for all $t \geq 0$. Substituting this into (28), we have

$$|\gamma(t)| = |\gamma(0)| \exp\left(-\int_0^t K(\tau)d\tau\right) \leq |\gamma(0)|e^{-K_{\min}t}.$$

Taking the limit as $t \to \infty$, we obtain $\lim_{t\to\infty} \gamma(t) = 0$, which implies

$$\lim_{t\to\infty} \frac{\lambda_1^L(t) + (n-1)\lambda_2^L(t)}{n} = sw^*. \tag{30}$$

Recall from the inverse relations in Lemma D.5 and Lemma D.6 that the parameters of $\boldsymbol{W}_{L:1}(t) = \boldsymbol{M}(A(t), B(t), C(t))$ are given by

$$A(t) = \frac{\lambda_1^L(t) + (n-1)\lambda_2^L(t) + n(s-1)\lambda_3^L(t)}{sn},$$

$$B(t) = \frac{\lambda_1^L(t) + (n-1)\lambda_2^L(t) - n\lambda_3^L(t)}{sn}.$$

Combining the results $\lambda_3(\infty) = 0$ and (30), the limits of these parameters are

$$\lim_{t\to\infty} A(t) = \frac{nsw^* + n(s-1)\cdot 0}{sn} = w^*,$$

$$\lim_{t\to\infty} B(t) = \frac{nsw^* - n\cdot 0}{sn} = w^*.$$

The loss function is defined by the error on the observed set $\Omega_{\text{block}}^{(s,n)}$, which consists of the diagonal blocks. Specifically, for each block, the diagonal entries converge to $A(\infty) = w^*$ and the off-diagonal entries converge to $B(\infty) = w^*$. Since all observed entries in $\boldsymbol{W}^*$ share the same value $w^*$, we conclude

$$\lim_{t\to\infty} \ell\left(\boldsymbol{W}_{L:1}(t); \Omega_{\text{block}}^{(s,n)}\right) = \frac{1}{2}\sum_{(i,j)\in\Omega_{\text{block}}^{(s,n)}} (w_{ij}(\infty) - w^*)^2 = 0,$$

which completes the proof.

$\square$

### D.5 UNIQUENESS OF THE LIMITING SINGULAR VALUES

**Proposition D.2.** *Under the setting of Theorem 3.3,*

$$f_1(\sigma) \triangleq \sigma^{\frac{2-L}{L}} - \left(\frac{w^*d - \sigma}{n-1}\right)^{\frac{2-L}{L}}$$

*strictly decreases in* $\sigma \in (0, w^*d)$. *Also,*

$$f_2(\sigma) \triangleq (w^*d - (n-1)\sigma)^{\frac{2-L}{L}} - \sigma^{\frac{2-L}{L}}$$

*strictly increases in* $\sigma \in \left(0, \frac{w^*d}{n-1}\right)$. *Therefore, Equations (8) and (9) admit a unique solution* $(\sigma_1, \sigma_i)$.

*Proof.* Under Theorem 3.3 with $L \geq 3$ and $1 < m < \infty$, the limiting nonzero singular values satisfy

$$\sigma_1 + (n-1)\sigma_i = w^*d, \tag{31}$$

where $i \in \{2, \ldots, n\}$. Since these are the nonzero singular values of the limiting product matrix, we have

$$\sigma_1 > 0, \qquad \sigma_i > 0.$$

Combining this with (31) yields

$$0 < \sigma_1 < w^*d, \qquad 0 < \sigma_i < \frac{w^*d}{n-1}.$$

Hence the implicit equations are naturally defined on the intervals $(0, w^*d)$ and $\left(0, \frac{w^*d}{n-1}\right)$. For brevity, set

$$a \triangleq \frac{2-L}{L} < 0.$$

**Uniqueness of $\sigma_1$.** Define

$$f_1(\sigma) \triangleq \sigma^a - \left(\frac{w^*d - \sigma}{n-1}\right)^a, \qquad \sigma \in (0, w^*d).$$

Then

$$f_1'(\sigma) = a\sigma^{a-1} + \frac{a}{n-1}\left(\frac{w^*d - \sigma}{n-1}\right)^{a-1}.$$

Since $a < 0$ and both terms inside the powers are positive on $(0, w^*d)$, we obtain

$$f_1'(\sigma) < 0 \qquad \text{for all } \sigma \in (0, w^*d).$$

Therefore $f_1$ is strictly decreasing on $(0, w^*d)$.

Moreover,

$$\lim_{\sigma \to 0^+} f_1(\sigma) = +\infty, \qquad \lim_{\sigma \to (w^*d)^-} f_1(\sigma) = -\infty.$$

Hence $f_1$ is a continuous bijection from $(0, w^*d)$ onto $\mathbb{R}$. Therefore, for any constant $C_{\alpha,m,L,d} \in \mathbb{R}$, there exists a unique $\sigma_1 \in (0, w^*d)$ such that

$$f_1(\sigma_1) = C_{\alpha,m,L,d}.$$

**Uniqueness of $\sigma_i$.** Define

$$f_2(\sigma) \triangleq (w^*d - (n-1)\sigma)^a - \sigma^a, \qquad \sigma \in \left(0, \frac{w^*d}{n-1}\right).$$

Differentiating gives

$$f_2'(\sigma) = -a(n-1)(w^*d - (n-1)\sigma)^{a-1} - a\sigma^{a-1}.$$

Since $-a > 0$ and both bases are positive on $\left(0, \frac{w^* d}{n-1}\right)$, we have

$$f_2'(\sigma) > 0 \qquad \text{for all } \sigma \in \left(0, \frac{w^* d}{n-1}\right).$$

Thus $f_2$ is strictly increasing on that interval.

Also,

$$\lim_{\sigma \to 0^+} f_2(\sigma) = -\infty, \qquad \lim_{\sigma \to \left(\frac{w^* d}{n-1}\right)^-} f_2(\sigma) = +\infty.$$

Hence $f_2$ is a continuous bijection from $\left(0, \frac{w^* d}{n-1}\right)$ onto $\mathbb{R}$. Therefore, for any constant $C_{\alpha,m,L,d} \in \mathbb{R}$, there exists a unique $\sigma_i \in \left(0, \frac{w^* d}{n-1}\right)$ such that

$$f_2(\sigma_i) = C_{\alpha,m,L,d}.$$

This proves that (8) and (9) admit unique solutions. $\qquad \square$

### D.6 PROOF FOR COROLLARY 3.4

**Corollary 3.4.** *Let $1 < m < \infty$, $n \geq 2$, $s \geq 1$, $w^* > 0$, and $L \geq 3$ be fixed. Then, as $\alpha \to 0$, the stable rank of the limit product matrix $\boldsymbol{W}_{L:1}(\infty)$ converges to one; that is,*

$$\mathrm{srank}\big(\boldsymbol{W}_{L:1}(\infty)\big) \to 1.$$

*Proof.* Fix $m > 1$, $n \geq 2$, $s \geq 1$, $w^* > 0$, and $L \geq 3$. Let

$$a \triangleq \frac{2-L}{L} < 0.$$

First, we analyze the behavior of

$$C_{\alpha,m,L,d} = \left(\frac{\alpha}{m}\right)^{2-L} \big((m+d-1)^{2-L} - (m-1)^{2-L}\big)$$

as $\alpha \to 0$. Since $L \geq 3$, we have $2 - L < 0$. The map $x \mapsto x^{2-L}$ is strictly decreasing on $(0, \infty)$, and because $m + d - 1 > m - 1 > 0$,

$$(m+d-1)^{2-L} - (m-1)^{2-L} < 0.$$

Moreover, $\left(\frac{\alpha}{m}\right)^{2-L} \to +\infty$ as $\alpha \to 0$. Hence

$$C_{\alpha,m,L,d} \to -\infty \quad \text{as } \alpha \to 0.$$

Next, consider the function from (9):

$$f_2(\sigma) = (w^*d - (n-1)\sigma)^a - \sigma^a, \quad \sigma \in \left(0, \frac{w^*d}{n-1}\right).$$

By Proposition D.2, we know that $f_2$ is a continuous, strictly increasing bijection from $\left(0, \frac{w^*d}{n-1}\right)$ onto $\mathbb{R}$, and for each $C \in \mathbb{R}$ there is a unique $\sigma(C)$ such that $f_2\big(\sigma(C)\big) = C$.

Now we apply this to $C_{\alpha,m,L,d}$. Since $C_{\alpha,m,L,d} \to -\infty$ as $\alpha \to 0$ and $f_2$ is strictly increasing with $\lim_{\sigma \to 0^+} f_2(\sigma) = -\infty$, it follows that

$$\sigma_i(\alpha) \to 0 \quad \text{as } \alpha \to 0.$$

Using the linear constraint (25), we then obtain

$$\sigma_1(\alpha) = w^*d - (n-1)\sigma_i(\alpha) \to w^*d \quad \text{as } \alpha \to 0.$$

The stable rank of $\boldsymbol{W}_{L:1}(\infty)$ is

$$\mathrm{srank}\big(\boldsymbol{W}_{L:1}(\infty)\big) = \frac{\sigma_1(\alpha)^2 + (n-1)\sigma_i(\alpha)^2 + n(s-1)\sigma_j^2}{\sigma_1(\alpha)^2} = 1 + (n-1)\left(\frac{\sigma_i(\alpha)}{\sigma_1(\alpha)}\right)^2.$$

Since $\sigma_i(\alpha) \to 0$ and $\sigma_1(\alpha) \to w^*d > 0$, we have

$$\frac{\sigma_i(\alpha)}{\sigma_1(\alpha)} \to 0,$$

and therefore

$$\mathrm{srank}\big(\boldsymbol{W}_{L:1}(\infty)\big) \to 1 \quad \text{as } \alpha \to 0.$$

$\square$

# E   PROOF FOR SECTION 4

In this section, we provide the proofs for the propositions and theorems presented in Section 4. First, Subsection E.1 presents the general form of Proposition 4.1 along with its proof. Next, Subsection E.2 details the proof of Theorem 4.2, focusing on the $2 \times 2$ matrix case. Lastly, Subsection E.3 generalizes the core ideas of Theorem 4.2 to $d \times d$ matrices and provides the formal statement and the proof of Theorem 4.3.

## E.1   GENERAL FORM AND PROOF OF PROPOSITION 4.1

We first present the general form of Proposition 4.1. This proposition applies to any "fully disconnected case", a scenario that involves the diagonal entries introduced within this same proposition.

For a $d \times d$ ground truth matrix $\boldsymbol{W}^*$, the observed entries are given by $\Omega = \{(i_n, j_n)\}_{n=1}^d$. Since we consider the fully disconnected case, $i_n \neq i_m, j_n \neq j_m$ for all $n \neq m \in [d]$. We factorize the solution model at time $t$ as $\boldsymbol{W}_{\boldsymbol{A},\boldsymbol{B}}(t) = \boldsymbol{A}(t)\boldsymbol{B}(t)$, where $\boldsymbol{W}_{\boldsymbol{A},\boldsymbol{B}}(t), \boldsymbol{A}(t), \boldsymbol{B}(t) \in \mathbb{R}^{d \times d}$. We consider the gradient flow dynamics with the loss function defined as in (2).

For a given row index $k$, since there exists a unique entry $(k, j) \in \Omega$, we denote this unique column index by $j^{(k)}$. Thus, $w_{k,j^{(k)}}^*$ and $w_{k,j^{(k)}}(t)$ refer to the ground truth weight $w_{k,j}^*$ and the time-varying weight $w_{k,j}(t)$ respectively, where $j = j^{(k)}$. Similarly, for a given column index $l$, since there exists a unique entry $(i, l) \in \Omega$, we denote this unique row index by $i^{(l)}$. Thus $w_{i^{(l)},l}^*$ and $w_{i^{(l)},l}$ refer to the ground truth weight $w_{i,l}^*$ and the time-varying weight $w_{i,l}(t)$ respectively, where $i = i^{(l)}$. Defining the residuals as $r_{ij}(t) := w_{ij}^* - w_{ij}(t)$, we adopt this compact notation for residuals as well. Then, we can derive a closed-form solution for *arbitrary initialization* with below proposition.

**Proposition E.1.** *Consider a ground truth matrix $\boldsymbol{W}^* \in \mathbb{R}^{d \times d}$ and a set of $d$ fully disconnected observations $\Omega = \{(i_n, j_n)\}_{n=1}^d$. The model is factorized as $\boldsymbol{W}_{\boldsymbol{A},\boldsymbol{B}}(t) = \boldsymbol{A}(t)\boldsymbol{B}(t)$, where the factors $\boldsymbol{A}(t), \boldsymbol{B}(t) \in \mathbb{R}^{d \times d}$. For each observed pair $(i_n, j_n) \in \Omega$, define the constants $P_{i_n,j_n}$ and $Q_{i_n,j_n}$ based on the initial values $\boldsymbol{A}(0)$ and $\boldsymbol{B}(0)$:*

$$P_{i_n,j_n} \triangleq \sum_{k=1}^d a_{i_n,k}(0)b_{k,j_n}(0) \quad and \quad Q_{i_n,j_n} \triangleq \sum_{k=1}^d \left(a_{i_n,k}(0)^2 + b_{k,j_n}(0)^2\right).$$

*Furthermore, for each such observed pair $(i_n, j_n)$, let the parameter $\bar{r}_{i_n,j_n}$ be determined from the ground truth entry $w_{i_n,j_n}^*$ and the constants defined above, as follows:*

$$\bar{r}_{i_n,j_n} \triangleq \frac{1}{2} \log \left( \frac{P_{i_n,j_n} + \frac{Q_{i_n,j_n}}{2}}{w_{i_n,j_n}^* + \sqrt{{w_{i_n,j_n}^*}^2 - P_{i_n,j_n}^2 + \left(\frac{Q_{i_n,j_n}}{2}\right)^2}} \right).$$

*Then, assuming convergence to a zero-loss solution (i.e., $w_{i_n,j_n}(\infty) = w_{i_n,j_n}^*$ for all $(i_n, j_n) \in \Omega$), any entry $a_{p,q}(\infty)$ of the converged matrix $\boldsymbol{A}(\infty)$ and any entry $b_{p,q}(\infty)$ of the converged matrix $\boldsymbol{B}(\infty)$ (for arbitrary indices $p, q \in [d]$) are explicitly given by:*

$$a_{p,q}(\infty) = a_{p,q}(0) \cosh\left(\bar{r}_{p,j^{(p)}}\right) - b_{q,j^{(p)}}(0) \sinh\left(\bar{r}_{p,j^{(p)}}\right),$$
$$b_{p,q}(\infty) = b_{p,q}(0) \cosh\left(\bar{r}_{i^{(q)},q}\right) - a_{i^{(q)},p}(0) \sinh\left(\bar{r}_{i^{(q)},q}\right).$$

*Proof.* We can express their evolution in the following vector form using the vectorized parameter $\boldsymbol{\theta}(t) := \begin{bmatrix} \mathrm{vec}(\boldsymbol{A}(t)) \\ \mathrm{vec}(\boldsymbol{B}(t)) \end{bmatrix} \in \mathbb{R}^{2d^2}$:

$$\dot{\boldsymbol{\theta}}(t) = - \begin{bmatrix} \boldsymbol{0}_{d^2,d^2} & \boldsymbol{R}(t) \\ \boldsymbol{R}(t)^\top & \boldsymbol{0}_{d^2,d^2} \end{bmatrix} \boldsymbol{\theta}(t) \tag{32}$$

where $\boldsymbol{R}(t) \in \mathbb{R}^{d^2 \times d^2}$ is defined as:

$$
\boldsymbol{R}(t) = \begin{bmatrix} r_{1,j^{(1)}}(t)\boldsymbol{e}_{j^{(1)}}^\top \\ r_{1,j^{(1)}}(t)\boldsymbol{e}_{j^{(1)}+d}^\top \\ \vdots \\ r_{1,j^{(1)}}(t)\boldsymbol{e}_{j^{(1)}+(d-1)d}^\top \\ r_{2,j^{(2)}}(t)\boldsymbol{e}_{j^{(2)}}^\top \\ r_{2,j^{(2)}}(t)\boldsymbol{e}_{j^{(2)}+d}^\top \\ \vdots \\ r_{d,j^{(d)}}(t)\boldsymbol{e}_{j^{(d)}+(d-1)d}^\top \end{bmatrix}
\tag{33}
$$

for $\boldsymbol{e}_i \in \mathbb{R}^{d^2}$ form the standard basis. Since $\begin{bmatrix} \boldsymbol{0}_{d^2,d^2} & \boldsymbol{R}(t) \\ \boldsymbol{R}(t)^\top & \boldsymbol{0}_{d^2,d^2} \end{bmatrix}$ commutes with any other $t$ values, the solution is given as:

$$
\boldsymbol{\theta}(t) = \exp\left( -\int_0^\tau \begin{bmatrix} \boldsymbol{0}_{d^2,d^2} & \boldsymbol{R}(t) \\ \boldsymbol{R}(t)^\top & \boldsymbol{0}_{d^2,d^2} \end{bmatrix} \mathrm{d}\tau \right) \cdot \boldsymbol{\theta}(0)
\tag{34}
$$

$$
= \exp\left( -\begin{bmatrix} \boldsymbol{0}_{d^2,d^2} & \bar{\boldsymbol{R}}(t) \\ \bar{\boldsymbol{R}}(t)^\top & \boldsymbol{0}_{d^2,d^2} \end{bmatrix} \mathrm{d}\tau \right) \cdot \boldsymbol{\theta}(0)
\tag{35}
$$

where

$$
\bar{\boldsymbol{R}}(t) := \int_0^t \boldsymbol{R}(\tau)\mathrm{d}\tau = \begin{bmatrix} \bar{r}_{1,j^{(1)}}(t)\boldsymbol{e}_{j^{(1)}}^\top \\ \bar{r}_{1,j^{(1)}}(t)\boldsymbol{e}_{j^{(1)}+d}^\top \\ \vdots \\ \bar{r}_{1,j^{(1)}}(t)\boldsymbol{e}_{j^{(1)}+(d-1)d}^\top \\ \bar{r}_{2,j^{(2)}}(t)\boldsymbol{e}_{j^{(2)}}^\top \\ \bar{r}_{2,j^{(2)}}(t)\boldsymbol{e}_{j^{(2)}+d}^\top \\ \vdots \\ \bar{r}_{d,j^{(d)}}(t)\boldsymbol{e}_{j^{(d)}+(d-1)d}^\top \end{bmatrix}
$$

for $\bar{r}_{i,j}(t) = \int_0^t r_{i,j}(\tau)\mathrm{d}\tau$. If we assume convergence, we get:

$$
\boldsymbol{\theta}(\infty) = \exp\left( -\begin{bmatrix} \boldsymbol{0}_{d^2,d^2} & \bar{\boldsymbol{R}}(\infty) \\ \bar{\boldsymbol{R}}(\infty)^\top & \boldsymbol{0}_{d^2,d^2} \end{bmatrix} \mathrm{d}\tau \right) \cdot \boldsymbol{\theta}(0)
\tag{36}
$$

$$
= \left( \begin{bmatrix} \boldsymbol{I}_{d^2} & \boldsymbol{0}_{d^2,d^2} \\ \boldsymbol{0}_{d^2,d^2} & \boldsymbol{I}_{d^2} \end{bmatrix} - \begin{bmatrix} \boldsymbol{0}_{d^2,d^2} & \bar{\boldsymbol{R}}(t) \\ \bar{\boldsymbol{R}}(t)^\top & \boldsymbol{0}_{d^2,d^2} \end{bmatrix} + \frac{1}{2}\begin{bmatrix} \bar{\boldsymbol{R}}(t)\bar{\boldsymbol{R}}(t)^\top & \boldsymbol{0}_{d^2,d^2} \\ \boldsymbol{0}_{d^2,d^2} & \bar{\boldsymbol{R}}(t)^\top \bar{\boldsymbol{R}}(t) \end{bmatrix} \right.
\tag{37}
$$

$$
-\frac{1}{6}\begin{bmatrix} \boldsymbol{0}_{d^2,d^2} & \bar{\boldsymbol{R}}(t)\bar{\boldsymbol{R}}(t)^\top \bar{\boldsymbol{R}}(t) \\ \bar{\boldsymbol{R}}(t)^\top \bar{\boldsymbol{R}}(t)\bar{\boldsymbol{R}}(t)^\top & \boldsymbol{0}_{d^2,d^2} \end{bmatrix} + \frac{1}{24}\begin{bmatrix} \left(\bar{\boldsymbol{R}}(t)\bar{\boldsymbol{R}}(t)^\top\right)^2 & \boldsymbol{0}_{d^2,d^2} \\ \boldsymbol{0}_{d^2,d^2} & \left(\bar{\boldsymbol{R}}(t)^\top \bar{\boldsymbol{R}}(t)\right)^2 \end{bmatrix}
\tag{38}
$$

$$
\left. -\cdots \right) \cdot \boldsymbol{\theta}(0),
\tag{39}
$$

which can be simplified as:

$$
\boldsymbol{\theta}(\infty) = \begin{bmatrix} \boldsymbol{C} & \boldsymbol{D} \\ \boldsymbol{E} & \boldsymbol{F} \end{bmatrix} \boldsymbol{\theta}(0),
\tag{40}
$$

with $\boldsymbol{C}, \boldsymbol{D}, \boldsymbol{E}$ and $\boldsymbol{F}$ are defined as following:

$$\boldsymbol{C} = \cosh\left(\operatorname{diag}\left(\bar{r}_{1,j^{(1)}}, \ldots, \bar{r}_{1,j^{(1)}}, \bar{r}_{2,j^{(2)}}, \ldots, \bar{r}_{2,j^{(2)}}, \ldots, \bar{r}_{d,j^{(d)}}, \ldots, \bar{r}_{d,j^{(d)}}\right)\right),$$

$$\boldsymbol{F} = \cosh\left(\operatorname{diag}\left(\bar{r}_{i^{(1)},1}, \bar{r}_{i^{(2)},2}, \ldots, \bar{r}_{i^{(d)},d}, \ldots, \bar{r}_{i^{(1)},1}, \bar{r}_{i^{(2)},2}, \ldots, \bar{r}_{i^{(d)},d}\right)\right),$$

$$\boldsymbol{D} = -\sinh\left(\left[\bar{r}_{1,j^{(1)}}\boldsymbol{e}_{j^{(1)}}^{\top}, \ldots, \bar{r}_{1,j^{(1)}}\boldsymbol{e}_{j^{(1)}+(d-1)d}^{\top}, \ldots, \bar{r}_{d,j^{(d)}}\boldsymbol{e}_{j^{(d)}}^{\top}, \ldots, \bar{r}_{d,j^{(d)}}\boldsymbol{e}_{j^{(d)}+(d-1)d}^{\top}\right]^{\top}\right),$$

$$\boldsymbol{E} = -\sinh\left(\left[\bar{r}_{1,j^{(1)}}\boldsymbol{e}_{j^{(1)}}, \ldots, \bar{r}_{1,j^{(1)}}\boldsymbol{e}_{j^{(1)}+(d-1)d}, \ldots, \bar{r}_{d,j^{(d)}}\boldsymbol{e}_{j^{(d)}}, \ldots, \bar{r}_{d,j^{(d)}}\boldsymbol{e}_{j^{(d)}+(d-1)d}\right]\right).$$

Here, for any matrix $\boldsymbol{P}$, the operations $\cosh(\boldsymbol{P})$ and $\sinh(\boldsymbol{P})$ are performed elementwise. For a set of $d$ observed indices $\Omega$, there exists $d$ corresponding unknown variables, $\bar{r}_{i_k,j_k}$. If convergence is guaranteed, the model yields $d$ equations relating these variables to the $d$ ground truth values. This implies that the variables $\bar{r}_{i_k,j_k}$ can be characterized as a closed-form. To characterize more rigorously, we substitute $\boldsymbol{C}, \boldsymbol{D}, \boldsymbol{E}$, and $\boldsymbol{F}$ into (40):

$$\boldsymbol{\theta}(\infty) = \begin{bmatrix} a_{1,1}(\infty) \\ a_{1,2}(\infty) \\ \vdots \\ a_{1,d}(\infty) \\ a_{2,1}(\infty) \\ a_{2,2}(\infty) \\ \vdots \\ a_{2,d}(\infty) \\ \vdots \\ a_{d,1}(\infty) \\ \vdots \\ a_{d,d}(\infty) \\ b_{1,1}(\infty) \\ b_{1,2}(\infty) \\ \vdots \\ b_{1,d}(\infty) \\ b_{2,1}(\infty) \\ b_{2,2}(\infty) \\ \vdots \\ b_{2,d}(\infty) \\ \vdots \\ b_{d,1}(\infty) \\ \vdots \\ b_{d,d}(\infty) \end{bmatrix} = \begin{bmatrix} a_{1,1}(0)\cosh(\bar{r}_{1,j^{(1)}}) - b_{1,j^{(1)}}(0)\sinh(\bar{r}_{1,j^{(1)}}) \\ a_{1,2}(0)\cosh(\bar{r}_{1,j^{(1)}}) - b_{2,j^{(1)}}(0)\sinh(\bar{r}_{1,j^{(1)}}) \\ \vdots \\ a_{1,d}(0)\cosh(\bar{r}_{1,j^{(1)}}) - b_{d,j^{(1)}}(0)\sinh(\bar{r}_{1,j^{(1)}}) \\ a_{2,1}(0)\cosh(\bar{r}_{2,j^{(2)}}) - b_{1,j^{(2)}}(0)\sinh(\bar{r}_{2,j^{(2)}}) \\ a_{2,2}(0)\cosh(\bar{r}_{2,j^{(2)}}) - b_{2,j^{(2)}}(0)\sinh(\bar{r}_{2,j^{(2)}}) \\ \vdots \\ a_{2,d}(0)\cosh(\bar{r}_{2,j^{(2)}}) - b_{d,j^{(2)}}(0)\sinh(\bar{r}_{2,j^{(2)}}) \\ \vdots \\ a_{d,1}(0)\cosh(\bar{r}_{d,j^{(d)}}) - b_{1,j^{(d)}}(0)\sinh(\bar{r}_{d,j^{(d)}}) \\ \vdots \\ a_{d,d}(0)\cosh(\bar{r}_{d,j^{(d)}}) - b_{d,j^{(d)}}(0)\sinh(\bar{r}_{d,j^{(d)}}) \\ -a_{i^{(1)},1}(0)\sinh(\bar{r}_{i^{(1)},1}) + b_{1,1}(0)\cosh(\bar{r}_{i^{(1)},1}) \\ -a_{i^{(2)},1}(0)\sinh(\bar{r}_{i^{(2)},2}) + b_{1,2}(0)\cosh(\bar{r}_{i^{(2)},2}) \\ \vdots \\ -a_{i^{(d)},1}(0)\sinh(\bar{r}_{i^{(d)},d}) + b_{1,d}(0)\cosh(\bar{r}_{i^{(d)},d}) \\ -a_{i^{(1)},2}(0)\sinh(\bar{r}_{i^{(1)},1}) + b_{2,1}(0)\cosh(\bar{r}_{i^{(1)},1}) \\ -a_{i^{(2)},2}(0)\sinh(\bar{r}_{i^{(2)},2}) + b_{2,2}(0)\cosh(\bar{r}_{i^{(2)},2}) \\ \vdots \\ -a_{i^{(d)},2}(0)\sinh(\bar{r}_{i^{(d)},d}) + b_{2,d}(0)\cosh(\bar{r}_{i^{(d)},d}) \\ \vdots \\ -a_{i^{(1)},d}(0)\sinh(\bar{r}_{i^{(1)},1}) + b_{d,1}(0)\cosh(\bar{r}_{i^{(1)},1}) \\ \vdots \\ -a_{i^{(d)},d}(0)\sinh(\bar{r}_{i^{(d)},d}) + b_{d,d}(0)\cosh(\bar{r}_{i^{(d)},d}) \end{bmatrix}. \tag{41}$$

Then, assuming convergence, for each observation $(i_n, j_n) \in \Omega$ (for $n = 1, \ldots, d$), we obtain the equation:

$$w_{i_n,j_n}^* = w_{i_n,j_n}(\infty) = a_{i_n,1}(\infty)b_{1,j_n}(\infty) + \cdots + a_{i_n,d}(\infty)b_{d,j_n}(\infty)$$

$$= \sum_{k=1}^{d}\left[\left(a_{i_n,k}(0)\cosh(\bar{r}_{i_n,j_n}) - b_{k,j^{(i_n)}}(0)\sinh(\bar{r}_{i_n,j_n})\right)\right.$$

$$\left. \cdot \left(b_{k,j_n}(0)\cosh(\bar{r}_{i_n,j_n}) - a_{i_n,k}(0)\sinh(\bar{r}_{i_n,j_n})\right)\right].$$

Let $C_n = \cosh(\bar{r}_{i_n,j_n})$ and $S_n = \sinh(\bar{r}_{i_n,j_n})$. Then we can rewrite the above equation as:

$$
\begin{aligned}
w^*_{i_n,j_n} &= \sum_{k=1}^{d} \left( a_{i_n,k}(0)b_{k,j_n}(0)C_n^2 - a_{i_n,k}(0)^2 C_n S_n - b_{k,j_n}(0)^2 C_n S_n + a_{i_n,k}(0)b_{k,j_n}(0)S_n^2 \right) \\
&= \left( \sum_{k=1}^{d} a_{i_n,k}(0)b_{k,j_n}(0) \right) \left( C_n^2 + S_n^2 \right) - \left( \sum_{k=1}^{d} \left( a_{i_n,k}(0)^2 + b_{k,j_n}(0)^2 \right) \right) C_n S_n \\
&= P_{i_n,j_n} \cosh(2\bar{r}_{i_n,j_n}) - \frac{Q_{i_n,j_n}}{2} \sinh(2\bar{r}_{i_n,j_n}),
\end{aligned}
\tag{42}
$$

where $P_{i_n,j_n} = \sum_{k=1}^{d} a_{i_n,k}(0)b_{k,j_n}(0)$ and $Q_{i_n,j_n} = \sum_{k=1}^{d} \left( a_{i_n,k}(0)^2 + b_{k,j_n}(0)^2 \right)$.

By solving (42) with respect to $\bar{r}_{i_n,j_n}$, we can get:

$$
\begin{aligned}
2w^*_{i_n,j_n} &= P_{i_n,j_n} \left( e^{2\bar{r}_{i_n,j_n}} + e^{-2\bar{r}_{i_n,j_n}} \right) - \frac{Q_{i_n,j_n}}{2} \left( e^{2\bar{r}_{i_n,j_n}} - e^{-2\bar{r}_{i_n,j_n}} \right) \\
&= e^{2\bar{r}_{i_n,j_n}} \left( P_{i_n,j_n} - \frac{Q_{i_n,j_n}}{2} \right) + e^{-2\bar{r}_{i_n,j_n}} \left( P_{i_n,j_n} + \frac{Q_{i_n,j_n}}{2} \right).
\end{aligned}
$$

Multiply by $e^{2\bar{r}_{i_n,j_n}}$ leads to:

$$
2w^*_{i_n,j_n} e^{2\bar{r}_{i_n,j_n}} = e^{4\bar{r}_{i_n,j_n}} \left( P_{i_n,j_n} - \frac{Q_{i_n,j_n}}{2} \right) + P_{i_n,j_n} + \frac{Q_{i_n,j_n}}{2}.
$$

Rearrange into a quadratic equation by setting $u = e^{2\bar{r}_{i_n,j_n}}$:

$$
\left( P_{i_n,j_n} - \frac{Q_{i_n,j_n}}{2} \right) u^2 - 2w^*_{i_n,j_n} u + P_{i_n,j_n} + \frac{Q_{i_n,j_n}}{2} = 0.
$$

By solving the above equation while noting that $P_{i_n,j_n} - \frac{Q_{i_n,j_n}}{2} \leq 0$ by the definition, we can get explicit solutions for $\bar{r}_{i_n,j_n}$:

$$
\bar{r}_{i_n,j_n} = \frac{1}{2} \log \left( \frac{P_{i_n,j_n} + \frac{Q_{i_n,j_n}}{2}}{w^*_{i_n,j_n} + \sqrt{{w^*_{i_n,j_n}}^2 - P^2_{i_n,j_n} + \left( \frac{Q_{i_n,j_n}}{2} \right)^2}} \right).
$$

Note that each $\bar{r}_{i_n,j_n}$ is solely determined by the initial points $\boldsymbol{\theta}(0)$. With $\bar{r}_{i_n,j_n}$ determined for each observed entry, we have closed-form expressions characterizing the model's learned relationship for these observations. Consequently, by (41), we have:

$$
\begin{aligned}
a_{p,q}(\infty) &= a_{p,q}(0) \cosh\left( \bar{r}_{p,j^{(p)}} \right) - b_{q,j^{(p)}}(0) \sinh\left( \bar{r}_{p,j^{(p)}} \right), \\
b_{p,q}(\infty) &= b_{p,q}(0) \cosh\left( \bar{r}_{i^{(q)},q} \right) - a_{i^{(q)},p}(0) \sinh\left( \bar{r}_{i^{(q)},q} \right).
\end{aligned}
$$

$\square$

### E.2  PROOF OF THEOREM 4.2

In this section, we will provide the analysis of $2 \times 2$ matrix that starts from pre-trained weights with diagonal observations $w^* \triangleq w_{11}^* = w_{22}^*$, $\boldsymbol{W_{A,B}}(t)$ cannot converge to a low-rank solution. Let $T_1 > t_1$ be the timestep that concludes the pre-train phase. For the sake of simplicity, we omit the $\epsilon$ term introduced in the pre-training phase. Then, we know from Proposition E.1, we have:

$$\boldsymbol{A}(T_1) = \boldsymbol{B}(T_1) = \begin{pmatrix} \sqrt{w^*} & 0 \\ 0 & \sqrt{w^*} \end{pmatrix}. \tag{43}$$

In the post-train phase, we introduce an additional observation in the off-diagonal entries, specifically $w_{12}^*$ or $w_{21}^*$. Without loss of generality, we assume $w_{12}^* > 0$ is revealed while other observations remain the same, i.e., $\Omega_{\text{post}} = \{(1,1),(1,2),(2,2)\}$. Note that the gradient of the post-train loss is:

$$\nabla \ell(\boldsymbol{W_{A,B}}) = \begin{pmatrix} w_{11} - w^* & w_{12} - w_{12}^* \\ 0 & w_{22} - w^* \end{pmatrix}$$
$$= \begin{pmatrix} a_{11}b_{11} + a_{12}b_{21} - w^* & a_{11}b_{12} + a_{12}b_{22} - w_{12}^* \\ 0 & a_{21}b_{12} + a_{22}b_{22} - w^* \end{pmatrix}.$$

For simplicity, we again omit the $\Omega$ term in the loss specification. We define the residuals for the relevant matrix elements as $r_{11} := w_{11} - w^*$, $r_{12} := w_{12} - w_{12}^*$, and $r_{22} := w_{22} - w^*$.

We begin by demonstrating a pairwise symmetry between the entries of $\boldsymbol{A}(t)$ and $\boldsymbol{B}(t)$, which simplifies subsequent analysis. To this end, we first provide the time derivatives for the elements of $\boldsymbol{A}(t)$ and $\boldsymbol{B}(t)$. Given the general gradient flow dynamics $\dot{\boldsymbol{A}}(t) = -\nabla \ell(\boldsymbol{W_{A,B}}(t))\boldsymbol{B}^\top(t)$ and $\dot{\boldsymbol{B}}(t) = -\boldsymbol{A}^\top(t)\nabla \ell(\boldsymbol{W_{A,B}}(t))$, the component-wise updates are as follows. For $\boldsymbol{A}(t)$:

$$\begin{aligned}
\dot{a}_{11}(t) &= b_{11}(t)(w^* - w_{11}(t)) + b_{12}(t)(w_{12}^* - w_{12}(t)), \\
\dot{a}_{12}(t) &= b_{21}(t)(w^* - w_{11}(t)) + b_{22}(t)(w_{12}^* - w_{12}(t)), \\
\dot{a}_{21}(t) &= b_{12}(t)(w^* - w_{22}(t)), \\
\dot{a}_{22}(t) &= b_{22}(t)(w^* - w_{22}(t)),
\end{aligned} \tag{44}$$

and for $\boldsymbol{B}(t)$:

$$\begin{aligned}
\dot{b}_{11}(t) &= a_{11}(t)(w^* - w_{11}(t)), \\
\dot{b}_{12}(t) &= a_{11}(t)(w_{12}^* - w_{12}(t)) + a_{21}(t)(w^* - w_{22}(t)), \\
\dot{b}_{21}(t) &= a_{12}(t)(w^* - w_{11}(t)), \\
\dot{b}_{22}(t) &= a_{12}(t)(w_{12}^* - w_{12}(t)) + a_{22}(t)(w^* - w_{22}(t)).
\end{aligned} \tag{45}$$

Using the equations above, we first present a result showing that the $k$-th derivative of each element in $\boldsymbol{A}(t)$ and $\boldsymbol{B}(t)$ at initialization exhibits a pairwise symmetry:

**Lemma E.1.** *Let $\boldsymbol{W_{A,B}}(T_1) = \boldsymbol{A}(T_1)\boldsymbol{B}(T_1) \in \mathbb{R}^{2 \times 2}$ be a product matrix, where $\boldsymbol{A}(T_1)$ and $\boldsymbol{B}(T_1)$ are matrices that are obtained at the end of the pre-training phase. Suppose the ground truth matrix satisfies $w_{11}^* = w_{22}^*$. Then for every $k \in \mathbb{N} \cup \{0\}$, the following identities hold:*

$$\begin{aligned}
a_{11}^{(k)}(T_1) &= b_{22}^{(k)}(T_1), & a_{12}^{(k)}(T_1) &= b_{12}^{(k)}(T_1), \\
a_{21}^{(k)}(T_1) &= b_{21}^{(k)}(T_1), & a_{22}^{(k)}(T_1) &= b_{11}^{(k)}(T_1),
\end{aligned} \tag{46}$$

*and consequently,*

$$w_{11}^{(k)}(T_1) = w_{22}^{(k)}(T_1). \tag{47}$$

*Proof.* We prove the statement by induction on $k$. When $k = 0$, by the initialization assumption, we have

$$a_{11}(T_1) = b_{22}(T_1), \quad a_{12}(T_1) = b_{12}(T_1), \quad a_{21}(T_1) = b_{21}(T_1), \quad a_{22}(T_1) = b_{11}(T_1),$$

and therefore $w_{11}(T_1) = w_{22}(T_1)$.

Assume that for all orders $m < k$ (with $k \geq 1$) the identities

$$a_{11}^{(m)}(T_1) = b_{22}^{(m)}(T_1), \quad a_{12}^{(m)}(T_1) = b_{12}^{(m)}(T_1), \quad a_{21}^{(m)}(T_1) = b_{21}^{(m)}(T_1), \quad a_{22}^{(m)}(T_1) = b_{11}^{(m)}(T_1),$$

hold, and hence also $w_{11}^{(m)}(T_1) = w_{22}^{(m)}(T_1)$. By the Leibniz rule, each element of the $k$-th derivative can be written as a finite sum involving derivatives of orders strictly less than $k$. For $\boldsymbol{A}(t)$:

$$a_{11}^{(k)}(t) = -\sum_{j=0}^{k-1} \binom{k-1}{j} \left( b_{11}^{(k-1-j)}(t) r_{11}^{(j)}(t) + b_{12}^{(k-1-j)}(t) r_{12}^{(j)}(t) \right),$$

$$a_{12}^{(k)}(t) = -\sum_{j=0}^{k-1} \binom{k-1}{j} \left( b_{21}^{(k-1-j)}(t) r_{11}^{(j)}(t) + b_{22}^{(k-1-j)}(t) r_{12}^{(j)}(t) \right),$$

$$a_{21}^{(k)}(t) = -\sum_{j=0}^{k-1} \binom{k-1}{j} b_{12}^{(k-1-j)}(t) r_{22}^{(j)}(t),$$

$$a_{22}^{(k)}(t) = -\sum_{j=0}^{k-1} \binom{k-1}{j} b_{22}^{(k-1-j)}(t) r_{22}^{(j)}(t),$$

and for $\boldsymbol{B}(t)$:

$$b_{11}^{(k)}(t) = -\sum_{j=0}^{k-1} \binom{k-1}{j} a_{11}^{(k-1-j)}(t) r_{11}^{(j)}(t),$$

$$b_{12}^{(k)}(t) = -\sum_{j=0}^{k-1} \binom{k-1}{j} \left( a_{11}^{(k-1-j)}(t) r_{12}^{(j)}(t) + a_{21}^{(k-1-j)}(t) r_{22}^{(j)}(t) \right),$$

$$b_{21}^{(k)}(t) = -\sum_{j=0}^{k-1} \binom{k-1}{j} a_{12}^{(k-1-j)}(t) r_{11}^{(j)}(t),$$

$$b_{22}^{(k)}(t) = -\sum_{j=0}^{k-1} \binom{k-1}{j} \left( a_{12}^{(k-1-j)}(t) r_{12}^{(j)}(t) + a_{22}^{(k-1-j)}(t) r_{22}^{(j)}(t) \right).$$

By the inductive hypothesis, all derivatives of order less than $k$ satisfy the symmetric relations at $t = T_1$. Inserting these equalities into the expressions with $t = T_1$ above shows that the symmetry is maintained at the $k$-th order:

$$a_{11}^{(k)}(T_1) = b_{22}^{(k)}(T_1), \quad a_{12}^{(k)}(T_1) = b_{12}^{(k)}(T_1), \quad a_{21}^{(k)}(T_1) = b_{21}^{(k)}(T_1), \quad a_{22}^{(k)}(T_1) = b_{11}^{(k)}(T_1),$$

proving equations (46) and (47). $\qquad\square$

**Lemma E.2.** *Under the setting of Lemma E.1, below relationships hold for all $t \geq T_1$:*

$$\begin{aligned} a_{11}(t) = b_{22}(t), \quad a_{12}(t) = b_{12}(t), \\ a_{21}(t) = b_{21}(t), \quad a_{22}(t) = b_{11}(t), \end{aligned} \tag{48}$$

*which further leads to $w_{11}(t) = w_{22}(t)$.*

*Proof.* By Lemmas F.6 and E.1, we may conclude that for all $t \geq T_1$, equation (48) holds, and therefore $w_{11}(t) = w_{22}(t)$. $\qquad\square$

By Lemma E.2, all entries of $\boldsymbol{B}(t)$ can be expressed in terms of the entries of $\boldsymbol{A}(t)$ for all $t \geq T_1$. From this point onward, we will represent $\boldsymbol{W_{A,B}}(t)$ solely using the elements of $\boldsymbol{A}(t)$. We begin by simplifying the time derivative of $\boldsymbol{A}(t)$ as follows:

$$\dot{a}_{11}(t) = a_{22}(t)(w^* - w_{11}(t)) + a_{12}(t)(w_{12}^* - w_{12}(t)),$$
$$\dot{a}_{12}(t) = a_{21}(t)(w^* - w_{11}(t)) + a_{11}(t)(w_{12}^* - w_{12}(t)),$$
$$\dot{a}_{21}(t) = a_{12}(t)(w^* - w_{22}(t)),$$
$$\dot{a}_{22}(t) = a_{11}(t)(w^* - w_{22}(t)).$$
$$(49)$$

Rewriting $\boldsymbol{W_{A,B}}(t)$ in terms of the elements of $\boldsymbol{A}(t)$ yields:

$$\boldsymbol{W_{A,B}}(t) = \boldsymbol{A}(t)\boldsymbol{B}(t)$$
$$= \begin{pmatrix} a_{11}(t) & a_{12}(t) \\ a_{21}(t) & a_{22}(t) \end{pmatrix} \begin{pmatrix} a_{22}(t) & a_{12}(t) \\ a_{21}(t) & a_{11}(t) \end{pmatrix}$$
$$= \begin{pmatrix} a_{11}(t)a_{22}(t) + a_{12}(t)a_{21}(t) & 2a_{11}(t)a_{12}(t) \\ 2a_{21}(t)a_{22}(t) & a_{11}(t)a_{22}(t) + a_{12}(t)a_{21}(t) \end{pmatrix}.$$
$$(50)$$

We can also simplify the time derivative of $\boldsymbol{W_{A,B}}(t)$ as follows:

$$\dot{w}_{11}(t) = (w^* - w_{11}(t)) \left( a_{11}^2(t) + a_{12}^2(t) + a_{21}^2(t) + a_{22}^2(t) \right)$$
$$+ (w_{12}^* - w_{12}(t)) \left( a_{11}(t)a_{21}(t) + a_{12}(t)a_{22}(t) \right),$$
$$\dot{w}_{12}(t) = 2(w_{12}^* - w_{12}(t)) \left( a_{11}^2(t) + a_{12}^2(t) \right)$$
$$+ 2(w^* - w_{11}(t)) \left( a_{11}(t)a_{21}(t) + a_{12}(t)a_{22}(t) \right),$$
$$\dot{w}_{21}(t) = 2(w^* - w_{11}(t))(a_{11}(t)a_{21}(t) + a_{12}(t)a_{22}(t)),$$
$$\dot{w}_{22}(t) = \dot{w}_{11}(t).$$
$$(51)$$

Using (50), we state the basic conservation law from Arora et al. (2018): if the matrices are initialized in a balanced manner, this balancedness is preserved throughout the training process. That is,

$$\boldsymbol{A}(T_1)^\top \boldsymbol{A}(T_1) = \boldsymbol{B}(T_1)\boldsymbol{B}(T_1)^\top,$$

holds at initialization, this leads to

$$a_{11}^2(t) + a_{21}^2(t) = a_{12}^2(t) + a_{22}^2(t), \quad \forall t \geq T_1.$$
$$(52)$$

Now, we are going to examine the time derivative of the loss:

$$\frac{d}{dt}\ell(\boldsymbol{W_{A,B}}(t)) = \left\langle \nabla\ell(\boldsymbol{W_{A,B}}(t)), \dot{\boldsymbol{W}}(t) \right\rangle$$
$$= \left\langle \nabla\ell(\boldsymbol{W_{A,B}}(t)), \dot{\boldsymbol{A}}(t)\boldsymbol{B}(t) + \boldsymbol{A}(t)\dot{\boldsymbol{B}}(t) \right\rangle$$
$$= \mathrm{Tr}\left( \nabla\ell^\top(\boldsymbol{W_{A,B}}(t)) \left( \dot{\boldsymbol{A}}(t)\boldsymbol{B}(t) + \boldsymbol{A}(t)\dot{\boldsymbol{B}}(t) \right) \right)$$
$$= \mathrm{Tr}\left( \nabla\ell^\top(\boldsymbol{W_{A,B}}(t))\dot{\boldsymbol{A}}(t)\boldsymbol{B}(t) \right) + \mathrm{Tr}\left( \nabla\ell^\top(\boldsymbol{W_{A,B}}(t))\boldsymbol{A}(t)\dot{\boldsymbol{B}}(t) \right)$$
$$= -\mathrm{Tr}\left( \nabla\ell^\top(\boldsymbol{W_{A,B}}(t))\nabla\ell(\boldsymbol{W_{A,B}}(t))\boldsymbol{B}^\top(t)\boldsymbol{B}(t) \right)$$
$$- \mathrm{Tr}\left( \nabla\ell^\top(\boldsymbol{W_{A,B}})\boldsymbol{A}(t)\boldsymbol{A}^\top(t)\nabla\ell(\boldsymbol{W_{A,B}}(t)) \right)$$
$$= -\mathrm{Tr}\left( \underbrace{\nabla\ell(\boldsymbol{W_{A,B}}(t))\boldsymbol{B}^\top(t)\boldsymbol{B}(t)\nabla\ell(\boldsymbol{W_{A,B}^\top}(t))}_{:=\boldsymbol{L}_1(t)} \right)$$
$$- \mathrm{Tr}\left( \underbrace{\nabla\ell(\boldsymbol{W_{A,B}^\top}(t))\boldsymbol{A}(t)\boldsymbol{A}^\top(t)\nabla\ell(\boldsymbol{W_{A,B}}(t))}_{:=\boldsymbol{L}_2(t)} \right).$$
$$(53)$$

The third equality follows from the fact that for any two matrices $\boldsymbol{A}$ and $\boldsymbol{B}$ of the same size, $\langle \boldsymbol{A}, \boldsymbol{B} \rangle = \mathrm{Tr}(\boldsymbol{A}^\top \boldsymbol{B})$. The last equation holds due to the cyclic property of the trace. Combining (53) with Lemma F.7, we can ensure $\boldsymbol{L}_1(t)$ and $\boldsymbol{L}_2(t)$ are both positive semidefinite, which implies the loss is monotonically non-increasing for all $t \geq T_1$.

With Lemma E.2 and the monotonicity of the loss, we can guarantee positiveness of $a_{11}, a_{22}, w_{11}$, and $w_{22}$ after the pre-train phase:

**Lemma E.3.** *For a product matrix $W_{A,B}(t) = A(t)B(t) \in \mathbb{R}^{2\times 2}$, if $a_{11}(T_1), a_{22}(T_1), w_{11}(T_1)$, and $w_{22}(T_1)$ have all positive values, following inequalities hold for all $t \geq T_1$:*

$$a_{11}(t), a_{22}(t) > 0, \quad a_{12}(t) \geq 0.$$

*Furthermore,*

$$w_{11}(t), w_{22}(t) > 0$$

*holds for all $t \geq T_1$.*

*Proof.* We will prove the inequalities step by step.

**Positiveness of $a_{11}(t)$.** For the sake of contradiction, assume that there exists a timestep $\tau_1 > T_1$ where $a_{11}(\tau_1) = 0$ holds. From (50) and Lemma F.3, we must have $\det(A(\tau_1)) > 0$, which implies that $a_{12}(\tau_1)a_{21}(\tau_1) < 0$. Given the monotonicity of $\ell$, $W_{A,B}(t)$ must satisfy:

$$\ell(W_{A,B}(t)) \leq \ell(W_{A,B}(T_1)). \tag{54}$$

for all $t \geq T_1$. However, $W_{A,B}(\tau_1)$ cannot satisfy (54) because $w_{11}(\tau_1), w_{22}(\tau_1) < 0$ and $w_{12}(\tau_1) = 0$ for any $\tau_1 \geq 0$. This contradiction implies that such a $\tau_1$ cannot exist.

**Positiveness of $a_{22}(t)$.** Similarly, let's assume there exists a time $\tau_2 > T_1$ such that $a_{22}(\tau_2) = 0$ for the first time. We can express $W_{A,B}(\tau_2)$ as:

$$W_{A,B}(\tau_2) = \begin{pmatrix} a_{12}(\tau_2)a_{21}(\tau_2) & 2a_{11}(\tau_2)a_{12}(\tau_2) \\ 0 & a_{12}(\tau_2)a_{21}(\tau_2) \end{pmatrix}.$$

where the diagonal entries are negative due to the condition $\det(A(\tau_2)) > 0$. Therefore, the time derivative of $a_{22}$ at timestep $\tau_2$ is positive:

$$\dot{a}_{22}(\tau_2) = a_{11}(\tau_2)(w^* - w_{11}(\tau_2)) > 0.$$

Since $a_{22}(t)$ is increasing at point $\tau_2$, there exists time $t' < \tau_2$ such that $a_{22}(t') < 0$ (since $a_{22}(t)$ is continuous and differentiable), which is contradictory. Consequently, there cannot exist a $\tau_2$ such that $a_{22}(\tau_2) = 0$.

**Positiveness of $a_{12}(t)$.** Given that $\ell$ is non-decreasing, we can state:

$$\ell(W_{A,B}(t)) = \frac{1}{2}\left[(w^* - w_{11}(t))^2 + (w_{12}^* - w_{12}(t))^2 + (w^* - w_{22}(t))^2\right]$$

$$\leq \ell(W_{A,B}(T_1)) = \frac{1}{2}w_{12}^{*}{}^2,$$

for all $t \geq T_1$. Since $(w^* - w_{11}(t))^2$ and $(w^* - w_{22}(t))^2$ are non-negative, $w_{12}(t)$ must be non-negative for all $t \geq T_1$. From (50), we know $w_{12}(t) = 2a_{11}(t)a_{12}(t)$, which implies $a_{12}(t) \geq 0$ for all $t \geq T_1$ with the above conclusion which states $a_{11}(t) > 0$.

**Positiveness of $w_{11}(t), w_{22}(t)$.** Likewise, assume for the sake of contradiction that there exists a time $\tau_3 \geq T_1$ when $w_{11}(\tau_3) = 0$ is first satisfied. This directly implies that $a_{11}(\tau_3)a_{22}(\tau_3) = -a_{12}(\tau_3)a_{21}(\tau_3)$. Squaring both sides of the equation yields:

$$a_{11}^2(\tau_3)a_{22}^2(\tau_3) = a_{12}^2(\tau_3)a_{21}^2(\tau_3).$$

Subtracting $a_{12}^2(\tau_3)a_{22}^2(\tau_3)$ from both sides:

$$a_{11}^2(\tau_3)a_{22}^2(\tau_3) - a_{12}^2(\tau_3)a_{22}^2(\tau_3) = a_{12}^2(\tau_3)a_{21}^2(\tau_3) - a_{12}^2(\tau_3)a_{22}^2(\tau_3).$$

Factoring:

$$a_{22}^2(\tau_3)\left(a_{11}^2(\tau_3) - a_{12}^2(\tau_3)\right) = a_{12}^2(\tau_3)\left(a_{21}^2(\tau_3) - a_{22}^2(\tau_3)\right).$$

By the conservation law in (52), we have $a_{11}^2(\tau_3) + a_{21}^2(\tau_3) = a_{12}^2(\tau_3) + a_{22}^2(\tau_3)$, which leads to $a_{11}^2(\tau_3) - a_{12}^2(\tau_3) = a_{22}^2(\tau_3) - a_{21}^2(\tau_3)$. Replacing $a_{11}^2(\tau_3) - a_{12}^2(\tau_3)$ with $-(a_{21}^2(\tau_3) - a_{22}^2(\tau_3))$:

$$-a_{22}^2(\tau_3)\left(a_{21}^2(\tau_3) - a_{22}^2(\tau_3)\right) = a_{12}^2(\tau_3)\left(a_{21}^2(\tau_3) - a_{22}^2(\tau_3)\right).$$

This gives us:

$$\left(a_{12}^2(\tau_3) + a_{22}^2(\tau_3)\right)\left(a_{21}^2(\tau_3) - a_{22}^2(\tau_3)\right) = 0.$$

Since $a_{22}(\tau_3) > 0$ from the previous result, we can conclude that $a_{21}(\tau_3) = \pm a_{22}(\tau_3)$. To determine the sign of $a_{21}(\tau_3)$, recall that $\boldsymbol{W_{A,B}}(\tau_3)$ is written as:

$$\boldsymbol{W_{A,B}}(\tau_3) = \begin{pmatrix} 0 & 2a_{11}(\tau_3)a_{12}(\tau_3) \\ 2a_{21}(\tau_3)a_{22}(\tau_3) & 0 \end{pmatrix}.$$

Since $a_{11}(\tau_3) > 0, a_{12}(\tau_3) \geq 0$ from the previous result, $2a_{11}(\tau_3)a_{12}(\tau_3) \geq 0$ holds. Also, given that $\det(\boldsymbol{W_{A,B}}(\tau_3)) > 0$, we can determine that $a_{21}(\tau_3)$ is negative, which implies $a_{21}(\tau_3) = -a_{22}(\tau_3)$. Additionally, by the conservation law, we have $a_{11}^2(\tau_3) = a_{12}^2(\tau_3)$, which leads to $a_{11}(\tau_3) = a_{12}(\tau_3) > 0$.

Finally, consider the time derivative of $w_{11}$ at timestep $\tau_3$, substituting $a_{11}(\tau_3)$ and $a_{21}(\tau_3)$ with $a_{12}(\tau_3)$ and $-a_{22}(\tau_3)$, respectively:

$$\begin{aligned}
\dot{w}_{11}(\tau_3) &= (w^* - w_{11}(\tau_3))(a_{11}^2(\tau_3) + a_{12}^2(\tau_3) + a_{21}^2(\tau_3) + a_{22}^2(\tau_3)) \\
&\quad + (w_{12}^* - w_{12}(\tau_3))(a_{11}(\tau_3)a_{21}(\tau_3) + a_{12}(\tau_3)a_{22}(\tau_3)) \\
&= 2w^*(a_{12}^2(\tau_3) + a_{22}^2(\tau_3)) \\
&> 0,
\end{aligned}$$

which contradicts our initial assumption.

$\square$

Given that the time derivative in the (51) includes the term $a_{11}(t)a_{21}(t) + a_{12}(t)a_{22}(t)$, we need to verify the sign of $a_{11}a_{21} + a_{12}a_{22}$ in order to proceed with the analysis. Below lemma shows that as long as $w_{12}(t) \leq w_{12}^*$ holds, $a_{11}(t)a_{21}(t) + a_{12}(t)a_{22}(t)$ is always lower bounded by zero.

**Lemma E.4.** *For a product matrix $\boldsymbol{W_{A,B}}(t) = \boldsymbol{A}(t)\boldsymbol{B}(t) \in \mathbb{R}^{2\times2}$, if at any point $t \in [T_1, T_2]$ we have $w_{12}(t) \leq w_{12}^*$, then the following inequality holds throughout the entire interval $[T_1, T_2]$:*

$$a_{11}(t)a_{21}(t) + a_{12}(t)a_{22}(t) \geq 0.$$

*Proof.* We first define $g(t) \triangleq a_{11}(t)a_{21}(t) + a_{12}(t)a_{22}(t)$. Recall that at $T_1$, we have $a_{12}(T_1) = a_{21}(T_1) = 0$, which implies $g(T_1) = 0$ as well. Note that by (49), at timestep $T_1$, we have

$$\dot{a}_{12}(T_1) = a_{11}(T_1)(w_{12}^* - w_{12}(T_1)) + a_{21}(T_1)(w^* - w_{11}(T_1)) > 0,$$

while other elements remain unchanged. This indicates that $g(t) > 0$ immediately after $T_1$. We now show that if $g(\tau) > 0$ for any $\tau \in (T_1, T_2]$, then there is no $\tau' \in [\tau, T_2]$ which satisfies both $g(\tau') = 0$ and $\frac{d}{dt}g(t)\big|_{t=\tau'} < 0$. This implies that $g(t)$ never becomes negative under the assumption of $w_{12}(t) \leq w_{12}^*$.

Suppose, for the sake of contradiction, that there exists a $\tau' \in [\tau, T_2]$ where $g(\tau') = 0$ and $\frac{d}{dt}g(t)\big|_{t=\tau'} < 0$. Given $g(\tau') = 0$ and the conservation law in (52), and the inequalities from Lemma E.3, we can determine that there exist two combinations of the solution:

1. $a_{11}(\tau') = a_{22}(\tau'), \ a_{12}(\tau') = -a_{21}(\tau'), \ a_{11}(\tau') > a_{12}(\tau').$

2. $a_{11}(\tau') = a_{22}(\tau'), \ a_{12}(\tau') = a_{21}(\tau') = 0.$

We take the time derivative of $g(t)$ at timestep $\tau'$ and substitute the values from (49) as follows:

$$\begin{aligned}
\frac{d}{dt}g(t)\Big|_{t=\tau'} &= \dot{a}_{11}(\tau')a_{21}(\tau') + a_{11}(\tau')\dot{a}_{21}(\tau') + \dot{a}_{12}(\tau')a_{22}(\tau') + a_{12}(\tau')\dot{a}_{22}(\tau') \\
&= 2(w^* - w_{11}(\tau'))(a_{11}(\tau')a_{12}(\tau') + a_{21}(\tau')a_{22}(\tau')) \\
&\quad + (w_{12}^* - w_{12}(\tau'))(a_{11}(\tau')a_{22}(\tau') + a_{12}(\tau')a_{21}(\tau')).
\end{aligned} \tag{55}$$

For the first case, substituting equations $a_{11}(\tau') = a_{22}(\tau')$ and $a_{12}(\tau') = -a_{21}(\tau')$ to (55) leads to:

$$\frac{d}{dt}g(t)\Big|_{t=\tau'} = (w_{12}^* - w_{12}(\tau'))w_{11}(\tau').$$

Since $w_{11}(t) > 0$ for all $t \geq T_1$, if $w_{12}(\tau') \leq w_{12}^*$ holds, then $g(t)$ cannot take negative values at time $\tau'$.

For the second case, substituting equations $a_{11}(\tau') = a_{22}(\tau')$ and $a_{12}(\tau') = a_{21}(\tau') = 0$ to (55) leads to:

$$\frac{d}{dt}g(t)\Big|_{t=\tau'} = (w_{12}^* - w_{12}(\tau'))a_{11}^2(\tau'),$$

which is again a non-negative value if $w_{12}(\tau') \leq w_{12}^*$, leading to a contradiction. $\qquad\square$

**Lemma E.5.** *For a product matrix $\mathbf{W}_{\mathbf{A},\mathbf{B}}(t) = \mathbf{A}(t)\mathbf{B}(t) \in \mathbb{R}^{2\times 2}$, the following inequalities holds for all timestep $t \geq T_1$:*

$$w_{12}(t) \leq w_{12}^*,$$
$$w_{11}(t), w_{22}(t) \geq w^*,$$
$$w_{21}(t) \leq 0.$$

*Proof.* We will prove this lemma in several steps:

**Step 1:** $w_{12}(t) \leq w_{12}^*$ for all $t \geq T_1$.

We know $w_{12}(T_1) = 0 \leq w_{12}^*$. Assume, for the sake of contradiction, that there exists a time $t' > T_1$ where $t'$ is the first timestep such that $w_{12}(t') > w_{12}^*$. If this were true, there must exist a time $s$ where $T_1 \leq s < t'$ such that:

$$w_{12}(s) = w_{12}^*, \quad \dot{w}_{12}(s) > 0.$$

For these conditions to be met, $w_{12}(s)$ must satisfy:

$$\dot{w}_{12}(s) = 2(w^* - w_{11}(s))(a_{11}(s)a_{21}(s) + a_{12}(s)a_{22}(s)) > 0. \tag{56}$$

To satisfy (56), there are two possibilities:

$$(w^* - w_{11}(s)) > 0 \quad \text{and} \quad (a_{11}(s)a_{21}(s) + a_{12}(s)a_{22}(s)) > 0, \tag{57}$$
$$\text{or} \quad (w^* - w_{11}(s)) < 0 \quad \text{and} \quad (a_{11}(s)a_{21}(s) + a_{12}(s)a_{22}(s)) < 0. \tag{58}$$

However, neither of these can be true:

1. Equation (58) contradicts Lemma E.4, given that $s < t'$.

2. Equation (57) cannot be satisfied because there is no $s$ where $w^* > w_{11}(s)$. If there were, there would be a time $s'$ where $T_1 \leq s' < s$ both satisfying $w_{11}(s') = w^*$, and $\dot{w}_{11}(s') < 0$. But we find:

$$\dot{w}_{11}(s') = (w_{12}^* - w_{12}(s'))(a_{11}(s')a_{21}(s') + a_{12}(s')a_{22}(s')) \geq 0.$$

   This is because $w_{12}(s') < w_{12}^*$, and thus $a_{11}(s')a_{21}(s') + a_{12}(s')a_{22}(s') \geq 0$ by Lemma E.4. Therefore, our initial assumption must be false, implying that $w_{12}(t) \leq w_{12}^*$ for all $t \geq T_1$.

**Step 2:** Prove $w_{11}(t) \geq w_{11}^*$ and $w_{22}(t) \geq w_{22}^*$ for all $t \geq T_1$.

Given $w_{12}(t) \leq w_{12}^*$ for all $t \geq T_1$, Lemma E.4 implies $a_{11}(t)a_{21}(t) + a_{12}(t)a_{22}(t) \geq 0$ for all $t \geq T_1$. The evolution of $w_{11}$ is given by:

$$\dot{w}_{11}(t) = (w^* - w_{11}(t))(a_{11}^2(t) + a_{12}^2(t) + a_{21}^2(t) + a_{22}^2(t)) + (w_{12}^* - w_{12}(t))(a_{11}(t)a_{21}(t) + a_{12}(t)a_{22}(t)).$$

By above equation, if there exists a time $t' \geq T_1$ where $w_{11}(t') = w^*$, we can conclude $\dot{w}_{11}(t') \geq 0$, and thus $w_{11}(t) \geq w^*$ for all $t \geq T_1$. By Lemma E.2, $w_{22}$ has the same value as $w_{11}$, so $w_{22}(t) \geq w^*$ for all $t \geq T_1$.

**Step 3:** Prove $w_{21}(t) \leq 0$ for all $t \geq T_1$.

The evolution of $w_{21}$ is given by:

$$\dot{w}_{21}(t) = 2(w^* - w_{11}(t))(a_{11}(t)a_{21}(t) + a_{12}(t)a_{22}(t)).$$

Since $w_{11}(t) \geq w^*$ and $a_{11}(t)a_{21}(t) + a_{12}(t)a_{22}(t) \geq 0$ for all $t \geq T_1$, we can conclude $w_{21}(t) \leq 0$ for all $t \geq T_1$.

$\square$

### E.2.1 PROOF OF LOSS CONVERGENCE

Recall that the time derivative of the loss function is written as:

$$\frac{d}{dt}\ell(\boldsymbol{W_{A,B}}(t)) = -\operatorname{Tr}(\boldsymbol{L}_1(t)) - \operatorname{Tr}(\boldsymbol{L}_2(t)),$$

where $\boldsymbol{L}_1(t)$ and $\boldsymbol{L}_2(t)$ are defined in (53). To further our analysis, we can expand the time derivative of the loss by calculating the trace of $\boldsymbol{L}_1(t)$ and $\boldsymbol{L}_2(t)$. We omit the time index $t$ when clear from context.

$$\boldsymbol{L}_1 = \begin{pmatrix} r_{11} & r_{12} \\ 0 & r_{22} \end{pmatrix} \begin{pmatrix} a_{21}^2 + a_{22}^2 & a_{11}a_{21} + a_{12}a_{22} \\ a_{11}a_{21} + a_{12}a_{22} & a_{11}^2 + a_{12}^2 \end{pmatrix} \begin{pmatrix} r_{11} & 0 \\ r_{12} & r_{22} \end{pmatrix}$$

$$= \begin{pmatrix} r_{11}^2(a_{21}^2 + a_{22}^2) + 2r_{11}r_{12}(a_{11}a_{21} + a_{12}a_{22}) + r_{12}^2(a_{11}^2 + a_{12}^2) & C_1 \\ C_1 & r_{22}^2(a_{11}^2 + a_{12}^2) \end{pmatrix},$$

for some time-dependent value $C_1$. Following a similar process, we calculate $\boldsymbol{L}_2$:

$$\boldsymbol{L}_2 = \begin{pmatrix} r_{11} & 0 \\ r_{12} & r_{22} \end{pmatrix} \begin{pmatrix} a_{11}^2 + a_{12}^2 & a_{11}a_{21} + a_{12}a_{22} \\ a_{11}a_{21} + a_{12}a_{22} & a_{21}^2 + a_{22}^2 \end{pmatrix} \begin{pmatrix} r_{11} & r_{12} \\ 0 & r_{22} \end{pmatrix}$$

$$= \begin{pmatrix} r_{11}^2(a_{11}^2 + a_{12}^2) & C_2 \\ C_2 & r_{12}^2(a_{11}^2 + a_{12}^2) + 2r_{12}r_{22}(a_{11}a_{21} + a_{12}a_{22}) + r_{22}^2(a_{21}^2 + a_{22}^2) \end{pmatrix},$$

again for the time-dependent value $C_2$. With these expressions for $\boldsymbol{L}_1$ and $\boldsymbol{L}_2$, we can now rewrite equation (53) in a more explicit form:

$$\begin{aligned} \frac{d}{dt}\ell(\boldsymbol{W_{A,B}}(t)) = & -\operatorname{Tr}(\boldsymbol{L}_1(t)) - \operatorname{Tr}(\boldsymbol{L}_2(t)) \\ = & -r_{11}^2(t)\left(a_{11}^2(t) + a_{12}^2(t) + a_{21}^2(t) + a_{22}^2(t)\right) \\ & - 2r_{12}^2(t)\left(a_{11}^2(t) + a_{12}^2(t)\right) \\ & - r_{22}^2(t)\left(a_{11}^2(t) + a_{12}^2(t) + a_{21}^2(t) + a_{22}^2(t)\right) \\ & - 2r_{12}(t)r_{22}(t)\left(a_{11}(t)a_{21}(t) + a_{12}(t)a_{22}(t)\right) \\ & - 2r_{11}(t)r_{12}(t)\left(a_{11}(t)a_{21}(t) + a_{12}(t)a_{22}(t)\right). \end{aligned} \tag{59}$$

Note that the (59) is the non-positive term. Given that $\boldsymbol{L}_1$ and $\boldsymbol{L}_2$ are positive semi-definite, we can analyze each diagonal entry separately. This leads us to the following inequalities:

$$r_{11}^2(a_{21}^2 + a_{22}^2) + 2r_{11}r_{12}(a_{11}a_{21} + a_{12}a_{22}) + r_{12}^2(a_{11}^2 + b_{12}^2) \geq 0,$$

$$r_{12}^2(a_{11}^2 + a_{12}^2) + 2r_{12}r_{22}(a_{11}a_{21} + a_{12}a_{22}) + r_{22}^2(a_{21}^2 + a_{22}^2) \geq 0.$$

By rearranging the above inequalities, we obtain:

$$-2r_{11}r_{12}(a_{11}a_{21} + a_{12}a_{22}) \leq r_{11}^2(a_{21}^2 + a_{22}^2) + r_{12}^2(a_{11}^2 + a_{12}^2),$$

$$-2r_{12}r_{22}(a_{11}a_{21} + a_{12}a_{22}) \leq r_{12}^2(a_{11}^2 + a_{12}^2) + r_{22}^2(a_{21}^2 + a_{22}^2).$$

Substituting these inequalities into equation (59), we derive:

$$\frac{d}{dt}\ell(\boldsymbol{W_{A,B}}(t)) \leq -r_{11}^2(t)\left(a_{11}^2(t) + a_{12}^2(t)\right) - r_{22}^2(t)\left(a_{11}^2(t) + a_{12}^2(t)\right). \tag{60}$$

This provides a tighter upper bound on the time derivative of the loss. However, it is still insufficient to guarantee convergence, as the bound does not depend on the term $r_{12}(t)$. As a result, even though the right-hand side converges to zero, this alone does not imply that the loss itself converges.

To further tighten the bound, we leverage the positive semidefiniteness of $\boldsymbol{L}_1$ and $\boldsymbol{L}_2$. Specifically, note that for both $\boldsymbol{QKQ}^\top$ and $\boldsymbol{Q}^\top \boldsymbol{KQ}$ to be positive semi-definite, the only necessary condition is $\boldsymbol{K} \succcurlyeq 0$. Therefore, we modify $\boldsymbol{L}_1(t)$ to $\widetilde{\boldsymbol{L}_1}(t) \triangleq \nabla\ell(\boldsymbol{W_{A,B}}(t))\left(\boldsymbol{B}^\top(t)\boldsymbol{B}(t) - \mu(t) \cdot \boldsymbol{e}_2\boldsymbol{e}_2^\top\right)\nabla\ell^\top(\boldsymbol{W_{A,B}}(t))$, where $\mu(t)$ is chosen to ensure that the matrix $\boldsymbol{B}^\top(t)\boldsymbol{B}(t) - \mu(t) \cdot \boldsymbol{e}_2\boldsymbol{e}_2^\top$ remains positive semidefinite. This guarantees that $\widetilde{\boldsymbol{L}_1}(t) \succcurlyeq 0$. To ensure this condition, $\mu(t)$ must satisfy:

$$
\left|\boldsymbol{B}(t)^\top \boldsymbol{B}(t) - \mu(t) \cdot \boldsymbol{e}_2\boldsymbol{e}_2^\top)\right| = \left|\begin{pmatrix} a_{21}^2(t) + a_{22}^2(t) & a_{11}(t)a_{21}(t) + a_{12}(t)a_{22}(t) \\ a_{11}(t)a_{21}(t) + a_{12}(t)a_{22}(t) & a_{11}^2(t) + a_{12}^2(t) - \mu(t) \end{pmatrix}\right|
$$
$$
= -\left(a_{21}^2(t) + a_{22}^2(t)\right)\mu(t) + \left(a_{11}(t)a_{22}(t) - a_{12}(t)a_{21}(t)\right)^2
$$
$$
\geq 0.
$$

Rearranging this inequality with respect to $\mu(t)$, we get:

$$
\mu(t) \leq \frac{(a_{11}(t)a_{22}(t) - a_{12}(t)a_{21}(t))^2}{a_{21}^2(t) + a_{22}^2(t)} \tag{61}
$$
$$
= \frac{\det(\boldsymbol{B}(t))^2}{a_{21}^2(t) + a_{22}^2(t)}.
$$

Therefore, if we set $\mu(t)$ to satisfy the above inequality, we can guarantee $\widetilde{\boldsymbol{L}_1}$ to be a positive semidefinite matrix. Now, $\widetilde{\boldsymbol{L}_1}(t)$ can be calculated as:

$$
\widetilde{\boldsymbol{L}_1} = \begin{pmatrix} r_{11} & r_{12} \\ 0 & r_{22} \end{pmatrix}\begin{pmatrix} a_{21}^2 + a_{22}^2 & a_{11}a_{21} + a_{12}a_{22} \\ a_{11}a_{21} + a_{12}a_{22} & a_{11}^2 + a_{12}^2 - \mu \end{pmatrix}\begin{pmatrix} r_{11} & 0 \\ r_{12} & r_{22} \end{pmatrix}
$$
$$
= \begin{pmatrix} r_{11}^2(a_{21}^2 + a_{22}^2) + 2r_{11}r_{12}(a_{11}a_{21} + a_{12}a_{22}) + r_{12}^2(a_{11}^2 + a_{12}^2 - \mu) & \tilde{C} \\ \tilde{C} & r_{22}^2(a_{12}^2 + a_{22}^2 - \mu) \end{pmatrix},
$$

for some $\tilde{C}$. Since the matrix $\boldsymbol{B}^\top\boldsymbol{B} - \mu \cdot \boldsymbol{e}_2\boldsymbol{e}_2^\top$ is positive semi-definite, we can ensure $a_{12}^2 + a_{22}^2 - \mu \geq 0$. This leads to the following inequality from $\left(\widetilde{\boldsymbol{L}_1}\right)_{11}$:

$$
-2r_{11}r_{12}(a_{11}a_{21} + a_{12}a_{22}) \leq r_{11}^2(a_{21}^2 + a_{22}^2) + r_{12}^2(a_{11}^2 + a_{12}^2 - \mu).
$$

Finally, substituting this inequality into (59), we arrive at:

$$
\frac{d}{dt}\ell(\boldsymbol{W_{A,B}}(t)) \leq -\left(r_{11}^2(t) + r_{22}^2(t)\right)\left(a_{11}^2(t) + a_{12}^2(t)\right) - r_{12}^2(t)\mu(t). \tag{62}
$$

To prove the convergence of the loss, our main remaining goal is to establish a time-invariant lower bound for

$$
\min\left\{a_{11}^2(t) + a_{12}^2(t), \ \mu(t)\right\}
$$

to apply Grönwall's inequality.

**Lemma E.6.** *For a solution matrix $\boldsymbol{W_{A,B}}(t)$ initialized as $\boldsymbol{W_{A,B}}(T_1)$, which represents the state of the matrix after pre-training up to time $T_1$, the inequality*

$$
\det\left(\boldsymbol{W_{A,B}}(t)\right) \geq w^{*2}
$$

*holds for all $t \geq T_1$.*

*Proof.* Since $w_{12}(t)$ must satisfy $|w_{12}(t) - w_{12}^*| \leq \sqrt{2\ell(\boldsymbol{W_{A,B}}(t))} \leq w_{12}^*$ by the monotonicity of the loss, we can ensure that $w_{12}(t) \geq 0$ for all $t \geq T_1$. Also, by Lemma E.5, we have $w_{11}(t), w_{22}(t) \geq w^*$, and $w_{21}(t) \leq 0$ for all $t \geq T_1$. Under these conditions, $\det(\boldsymbol{W_{A,B}}(t))$ can be lower bounded as:

$$
\det(\boldsymbol{W_{A,B}}(t)) = w_{11}(t)w_{22}(t) - w_{12}(t)w_{21}(t) \geq w^{*2},
$$

for all timesteps $t \geq T_1$. $\qquad\square$

**Lemma E.7.** *For $\mu(t)$ defined to satisfy (61) and the entries in $\boldsymbol{A}(t)$, the following inequality holds for all timesteps $t \geq T_1$:*

$$\min \left\{ a_{11}^2(t) + a_{12}^2(t), \ \mu(t) \right\} \geq w^*.$$

*Proof.* To prove the lower bound of $a_{11}^2(t) + a_{12}^2(t)$, Our goal is to demonstrate that $a_{11}^2(t) + a_{12}^2(t) \geq w^*$ for all timesteps $t$ after $T_1$. By Lemma E.7, we have $\|\boldsymbol{W}_{\boldsymbol{A},\boldsymbol{B}}(t)\|_F \geq \sqrt{2}w^*$, which leads to:

$$\sqrt{2}w^* \leq \|\boldsymbol{W}_{\boldsymbol{A},\boldsymbol{B}}(t)\|_F$$
$$= \sqrt{\sigma_1^2\left(\boldsymbol{W}_{\boldsymbol{A},\boldsymbol{B}}(t)\right) + \sigma_2^2\left(\boldsymbol{W}_{\boldsymbol{A},\boldsymbol{B}}(t)\right)}.$$

By applying Lemma F.4, we have:

$$\sqrt{\sigma_1^2(\boldsymbol{W}_{\boldsymbol{A},\boldsymbol{B}}(t)) + \sigma_2^2(\boldsymbol{W}_{\boldsymbol{A},\boldsymbol{B}}(t))} = \sqrt{\sigma_1^4(\boldsymbol{A}(t)) + \sigma_2^4(\boldsymbol{A}(t))}$$
$$= \sqrt{\left(\sigma_1^2\left(\boldsymbol{A}(t)\right) + \sigma_2^2\left(\boldsymbol{A}(t)\right)\right)^2 - 2\sigma_1^2(\boldsymbol{A}(t))\sigma_2^2(\boldsymbol{A}(t))}$$
$$= \sqrt{\|\boldsymbol{A}(t)\|_F^4 - 2\det(\boldsymbol{A}(t))^2}. \tag{63}$$

Rewriting (63) while applying Lemmas F.4 and E.6 leads to:

$$\|\boldsymbol{A}(t)\|_F^4 \geq 2w^{*2} + 2\det(\boldsymbol{A}(t))^2$$
$$= 2w^{*2} + 2\det(\boldsymbol{W}_{\boldsymbol{A},\boldsymbol{B}}(t))$$
$$\geq 4w^{*2}.$$

Thus, $\boldsymbol{A}(t)$ have to satisfy $\|\boldsymbol{A}(t)\|_F^2 \geq 2w^*$ for all timesteps $t \geq T_1$. Now, assume that there exists a time $t' > T_1$ such that $a_{11}^2(t') + a_{12}^2(t') < w^*$. To satisfy inequality $\|\boldsymbol{A}(t')\|_F^2 \geq 2w^*$, we would need at least $a_{21}^2(t') + a_{22}^2(t') > w^*$ to hold. To verify the value of $a_{21}^2(t') + a_{22}^2(t')$, we take its time derivative using (49):

$$\frac{d}{dt}(a_{21}^2(t) + a_{22}^2(t)) = 2a_{21}(t)\dot{a_{21}}(t) + 2a_{22}(t)\dot{a_{22}}(t)$$
$$= -2a_{12}(t)a_{21}(t)r_{22}(t) - 2a_{11}(t)a_{22}(t)r_{22}(t)$$
$$= -2r_{22}(t)(a_{11}(t)a_{22}(t) + a_{12}(t)a_{21}(t))$$
$$= 2w_{11}(t)(w^* - w_{11}(t)).$$

Since $w_{11}(t) \geq w^*$ holds by Lemma E.5 for all $t \geq T_1$, we conclude $a_{21}^2(t) + a_{22}^2(t)$ is monotonically non-increasing from time $t \geq T_1$. Since $a_{12}^2(T_1) + a_{22}^2(T_1)$ is initialized as $w^*$, this implies that $a_{21}^2(t') + a_{22}^2(t') \leq w^*$. Consequently, there cannot exist a $t' > T_1$ such that $a_{11}^2(t') + a_{12}^2(t') < w^*$ holds, which leads to contradiction.

Next, we are now showing that the term $\frac{\det(\boldsymbol{B}(t))^2}{a_{21}^2(t) + a_{22}^2(t)}$ is lower bounded by $w^*$. Therefore, if we set $\mu(t)$ as $w^*$, we can guarantee the positive semidefiniteness of $\widetilde{\boldsymbol{L}_1}(t)$.

By applying Lemma F.4 and the lower bound of $\det(\boldsymbol{W}_{\boldsymbol{A},\boldsymbol{B}}(t))$ by Lemma E.6, we have

$$\frac{\det\left(\boldsymbol{B}(t)\right)^2}{a_{21}^2(t) + a_{22}^2(t)} = \frac{\det\left(\boldsymbol{W}_{\boldsymbol{A},\boldsymbol{B}}(t)\right)}{a_{21}^2(t) + a_{22}^2(t)} \geq \frac{w^{*2}}{a_{21}^2(t) + a_{22}^2(t)}.$$

Also, from the previous result, we have an upper bound on $a_{21}^2(t) + a_{22}^2(t)$, which is $a_{21}^2(t) + a_{22}^2(t) \leq w^*$. Combining these results, the following inequality holds:

$$\frac{\det\left(\boldsymbol{W}_{\boldsymbol{A},\boldsymbol{B}}(t)\right)}{a_{21}^2(t) + a_{22}^2(t)} \geq w^*.$$

Therefore, if we set $\mu(t)$ to be $w^*$, $\mu(t)$ can satisfy the positive semidefiniteness condition. By combining the results, we can finally guarantee:

$$\min \left\{ a_{11}^2(t) + a_{12}^2(t), \ \mu(t) \right\} \geq w^*.$$

$\square$

Using the results of Lemma E.7, we can rewrite (62) as follows:

$$\frac{d}{dt}\ell(\boldsymbol{W_{A,B}}(t)) \leq -\left(r_{11}^2(t) + r_{22}^2(t)\right)\left(a_{11}^2(t) + a_{12}^2(t)\right) - r_{12}^2(t)\mu(t)$$
$$\leq -\left(r_{11}^2(t) + r_{12}^2(t) + r_{22}^2(t)\right)w^*$$
$$\leq -2w^*\ell(\boldsymbol{W_{A,B}}(t)).$$

Applying Grönwall's inequality to our previous result, we can now demonstrate loss convergence where $t \geq T_1$:

$$\ell(\boldsymbol{W_{A,B}}(t)) \leq \ell(\boldsymbol{W_{A,B}}(T_1))e^{-2w^*(t-T_1)}$$
$$= \frac{1}{2}w_{12}^{*\ 2}e^{-2w^*(t-T_1)}. \tag{64}$$

This inequality allows us to conclude that $\ell(\boldsymbol{W_{A,B}}(t))$ converges to zero exponentially.

### E.2.2   PROOF OF STABLE RANK BOUND

From (64), we know that at convergence, $w_{11}(\infty) = w_{22}(\infty) = w^*$ and $w_{12}(\infty) = w_{12}^*$. Although a closed-form expression for $w_{21}(\infty)$ is unavailable, Lemma E.5 shows that $w_{21}(t) \leq 0$ for $t \geq T_1$, which implies $w_{21}(\infty) \leq 0$. This indicates that the test loss remains strictly positive, as the ground-truth value $w_{21}^* = \frac{w^{*\ 2}}{w_{12}^*}$ is assumed to be strictly positive.

In this section, we leverage the fast convergence rate detailed in (64) to establish bounds on the singular values of the converged matrix $\boldsymbol{W_{A,B}}(\infty)$. Subsequently, these singular value bounds are used to further bound the stable rank of $\boldsymbol{W_{A,B}}(\infty)$.

**Lemma E.8.** *The singular values of $\boldsymbol{W_{A,B}}(\infty)$ fulfill:*

$$\sigma_1(\boldsymbol{W_{A,B}}(\infty)) \leq w^* \cdot \exp\left(2\frac{w_{12}^*}{w^*}\right),$$
$$\sigma_2(\boldsymbol{W_{A,B}}(\infty)) \geq w^* \cdot \exp\left(-2\frac{w_{12}^*}{w^*}\right).$$

*Proof.* We denote the singular values of $\boldsymbol{W_{A,B}}(t)$ as $\sigma_r(t)$ for simplicity. By Lemma F.1, we can get general solution of each singular value $\sigma_r(t)$ by solving linear differential equation:

$$\sigma_r(t) = \sigma_r(s) \cdot \exp\left(-2\int_{t'=s}^{t} \langle \nabla\ell(\boldsymbol{W_{A,B}}(t')), \boldsymbol{u}_r(t')\boldsymbol{v}_r^\top(t')\rangle dt'\right), \quad r = 1, 2, \tag{65}$$

where $\boldsymbol{u}_r(t)$ and $\boldsymbol{v}_r(t)$ denotes left and right singular vector of corresponding $r$-th singular value, respectively. Since $\boldsymbol{u}_r(t)$ and $\boldsymbol{v}_r(t)$ are both unit vectors, applying Cauchy-Schwartz inequality, we can bound $\langle \nabla\ell(\boldsymbol{W_{A,B}}(t)), \boldsymbol{u}_r(t)\boldsymbol{v}_r^\top(t)\rangle$ by:

$$\left|\langle \nabla\ell(\boldsymbol{W_{A,B}}(t)), \boldsymbol{u}_r(t)\boldsymbol{v}_r^\top(t)\rangle\right| \leq \|\nabla\ell(\boldsymbol{W_{A,B}}(t))\|_F \cdot \left\|\boldsymbol{u}_r(t)\boldsymbol{v}_r^\top(t)\right\|_F$$
$$= \|\nabla\ell(\boldsymbol{W_{A,B}}(t))\|_F$$
$$= \sqrt{2\ell(\boldsymbol{W_{A,B}}(t))}.$$

we can get bound $\sigma_r(t)$ as following:

$$\sigma_r(s) \cdot \exp\left(-2\sqrt{2}\int_{t'=s}^{t} \sqrt{\ell(\boldsymbol{W_{A,B}}(t'))}dt'\right) \leq \sigma_r(t) \leq \sigma_r(s) \cdot \exp\left(2\sqrt{2}\int_{t'=s}^{t} \sqrt{\ell(\boldsymbol{W_{A,B}}(t'))}dt'\right)$$
$$\tag{66}$$

With the setting above, in the pre-train section, after $T_1$ timesteps, we prove that $\sigma_1(T_1) = \sigma_2(T_1) = w^*$. Starting from $T_1$ with pre-trained weights, we can lower bound $\sigma_2(\boldsymbol{W_{A,B}}(t))$ with equations (64)

and (66) when $t \geq T_1$ as follows:

$$
\begin{aligned}
\sigma_2(t) &\geq \sigma_2(T_1) \cdot \exp\left(-2\sqrt{2} \int_{t'=T_1}^{t} \sqrt{\ell(\boldsymbol{W_{A,B}}(t'))}dt'\right) \\
&\geq w^* \cdot \exp\left(-2w_{12}^* \int_{t'=T_1}^{t} e^{-w^*(t'-T_1)}dt'\right) \\
&= w^* \cdot \exp\left(-\frac{2w_{12}^*}{w^*}\left(1 - e^{-w^*(t-T_1)}\right)\right).
\end{aligned}
$$

and when $t \to \infty$, $\sigma_2(\infty)$ can be lower bounded by:

$$
\sigma_2(\infty) \geq w^* \cdot e^{-2 \cdot \frac{w_{12}^*}{w^*}}.
$$

In the same way, we can upper bound $\sigma_1(\infty)$ by:

$$
\sigma_1(\infty) \leq w^* \cdot e^{2 \cdot \frac{w_{12}^*}{w^*}}.
$$

$\square$

By Lemma E.8, we can now lower bound the stable rank of a matrix $\boldsymbol{W_{A,B}}(\infty)$:

$$
\begin{aligned}
\frac{\|\boldsymbol{W_{A,B}}(\infty)\|_F^2}{\|\boldsymbol{W_{A,B}}(\infty)\|_2^2} &= \frac{\sigma_1^2(\boldsymbol{W_{A,B}}(\infty)) + \sigma_2^2(\boldsymbol{W_{A,B}}(\infty))}{\sigma_1^2(\boldsymbol{W_{A,B}}(\infty))} \\
&= 1 + \frac{\sigma_2^2(\boldsymbol{W_{A,B}}(\infty))}{\sigma_1^2(\boldsymbol{W_{A,B}}(\infty))} \\
&\geq 1 + \exp\left(-8\frac{w_{12}^*}{w^*}\right),
\end{aligned}
$$

which concludes the proof of Theorem 4.2.

### E.3 FORMAL STATEMENT AND PROOF OF THEOREM 4.3

We now extend the preceding analysis to the general case involving a ground truth matrix $\boldsymbol{W}^* \in \mathbb{R}^{d \times d}$. The solution matrix $\boldsymbol{W_{A,B}} \in \mathbb{R}^{d \times d}$ is again factorized as $\boldsymbol{W_{A,B}} = \boldsymbol{AB}$, where both $\boldsymbol{A}, \boldsymbol{B} \in \mathbb{R}^{d \times d}$. In this section, our detailed presentation and proof of Theorem 4.3 (from the main text) are structured as follows: we first introduce and prove Theorem E.2, which is then followed by its direct consequence, Corollary E.3.

We use the slightly modified loss function:

$$\mathcal{L}(\boldsymbol{A}, \boldsymbol{B}) = \frac{1}{2} \sum_{n=1}^{N} \left( \langle \boldsymbol{AB}, \boldsymbol{X}_n \rangle - y_n \right)^2, \tag{67}$$

where the measurement matrix $\boldsymbol{X}_n = \boldsymbol{e}_{i_n} \boldsymbol{e}_{j_n}^\top$ represents a masking matrix, with the $n$-th observed entry set to one and all other entries set to zero, and $y_n \in \mathbb{R}$ denotes the ground truth value of the $n$-th observation. Then, by defining $\boldsymbol{\Theta} = \begin{bmatrix} \boldsymbol{A} \\ \boldsymbol{B}^\top \end{bmatrix} \in \mathbb{R}^{2d \times d}$ and $\bar{\boldsymbol{X}}_n = \frac{1}{2} \begin{bmatrix} \boldsymbol{0} & \boldsymbol{X}_n \\ \boldsymbol{X}_n^\top & \boldsymbol{0} \end{bmatrix} \in \mathbb{R}^{2d \times 2d}$, we can rewrite the (67) as:

$$\mathcal{L}(\boldsymbol{A}, \boldsymbol{B}) = \tilde{\mathcal{L}}(\boldsymbol{\Theta}) = \frac{1}{2} \sum_{n=1}^{N} \left( \langle \boldsymbol{\Theta}\boldsymbol{\Theta}^\top, \bar{\boldsymbol{X}}_n \rangle - y_n \right)^2$$

$$= \frac{1}{2} \| F(\boldsymbol{\Theta}) - \boldsymbol{y} \|_2^2. \tag{68}$$

Here, $F(\boldsymbol{\Theta})$ and $\boldsymbol{y}$ represent vectors defined as:

$$F(\boldsymbol{\Theta}) \triangleq \begin{bmatrix} \langle \boldsymbol{\Theta}\boldsymbol{\Theta}^\top, \bar{\boldsymbol{X}}_1 \rangle \\ \langle \boldsymbol{\Theta}\boldsymbol{\Theta}^\top, \bar{\boldsymbol{X}}_2 \rangle \\ \vdots \\ \langle \boldsymbol{\Theta}\boldsymbol{\Theta}^\top, \bar{\boldsymbol{X}}_N \rangle \end{bmatrix} \in \mathbb{R}^N, \quad \boldsymbol{y} \triangleq \begin{bmatrix} y_1 \\ y_2 \\ \vdots \\ y_N \end{bmatrix} \in \mathbb{R}^N. \tag{69}$$

By reparameterizing $\boldsymbol{A}, \boldsymbol{B}$ to $\boldsymbol{\Theta}$, and $\boldsymbol{X}_n$ to $\bar{\boldsymbol{X}}_n$, we can reduce the parameter matrices into a single matrix $\boldsymbol{\Theta}$ while ensuring the symmetry of $\boldsymbol{\Theta}\boldsymbol{\Theta}^\top$. We train the model $\boldsymbol{\Theta}$ via gradient flow, where the loss evolution is given by:

$$\dot{\tilde{\mathcal{L}}}(\boldsymbol{\Theta}(t)) = (F(\boldsymbol{\Theta}(t)) - \boldsymbol{y})^\top \dot{F}(\boldsymbol{\Theta}(t))$$

$$= (F(\boldsymbol{\Theta}(t)) - \boldsymbol{y})^\top \begin{bmatrix} \frac{d}{dt} \langle \boldsymbol{\Theta}(t)\boldsymbol{\Theta}(t)^\top, \bar{\boldsymbol{X}}_1 \rangle \\ \frac{d}{dt} \langle \boldsymbol{\Theta}(t)\boldsymbol{\Theta}(t)^\top, \bar{\boldsymbol{X}}_2 \rangle \\ \vdots \\ \frac{d}{dt} \langle \boldsymbol{\Theta}(t)\boldsymbol{\Theta}(t)^\top, \bar{\boldsymbol{X}}_N \rangle \end{bmatrix}$$

$$= 2 (F(\boldsymbol{\Theta}(t)) - \boldsymbol{y})^\top \begin{bmatrix} \langle \bar{\boldsymbol{X}}_1 \boldsymbol{\Theta}(t), \dot{\boldsymbol{\Theta}}(t) \rangle \\ \langle \bar{\boldsymbol{X}}_2 \boldsymbol{\Theta}(t), \dot{\boldsymbol{\Theta}}(t) \rangle \\ \vdots \\ \langle \bar{\boldsymbol{X}}_N \boldsymbol{\Theta}(t), \dot{\boldsymbol{\Theta}}(t) \rangle \end{bmatrix}$$

$$= 2 (F(\boldsymbol{\Theta}(t)) - \boldsymbol{y})^\top \begin{bmatrix} \text{vec} \left( \bar{\boldsymbol{X}}_1 \boldsymbol{\Theta}(t) \right)^\top \\ \text{vec} \left( \bar{\boldsymbol{X}}_2 \boldsymbol{\Theta}(t) \right)^\top \\ \vdots \\ \text{vec} \left( \bar{\boldsymbol{X}}_N \boldsymbol{\Theta}(t) \right)^\top \end{bmatrix} \text{vec} \left( \dot{\boldsymbol{\Theta}}(t) \right) \tag{70}$$

$$= (F(\boldsymbol{\Theta}(t)) - \boldsymbol{y})^\top J(\boldsymbol{\Theta}(t)) \text{vec} \left( \dot{\boldsymbol{\Theta}}(t) \right). \tag{71}$$

Here, the Jacobian matrix $J(\boldsymbol{\Theta}(t))$ is defined as:

$$J(\boldsymbol{\Theta}(t)) \triangleq \frac{\partial F(\boldsymbol{\Theta}(t))}{\partial \mathrm{vec}(\boldsymbol{\Theta}(t))} = \begin{bmatrix} \mathrm{vec}\left(\nabla_{\boldsymbol{\Theta}}\langle\boldsymbol{\Theta}(t)\boldsymbol{\Theta}(t)^\top, \bar{\boldsymbol{X}}_1\rangle\right)^\top \\ \mathrm{vec}\left(\nabla_{\boldsymbol{\Theta}}\langle\boldsymbol{\Theta}(t)\boldsymbol{\Theta}(t)^\top, \bar{\boldsymbol{X}}_2\rangle\right)^\top \\ \vdots \\ \mathrm{vec}\left(\nabla_{\boldsymbol{\Theta}}\langle\boldsymbol{\Theta}(t)\boldsymbol{\Theta}(t)^\top, \bar{\boldsymbol{X}}_N\rangle\right)^\top \end{bmatrix} = 2 \begin{bmatrix} \mathrm{vec}\left(\bar{\boldsymbol{X}}_1\boldsymbol{\Theta}(t)\right)^\top \\ \mathrm{vec}\left(\bar{\boldsymbol{X}}_2\boldsymbol{\Theta}(t)\right)^\top \\ \vdots \\ \mathrm{vec}\left(\bar{\boldsymbol{X}}_N\boldsymbol{\Theta}(t)\right)^\top \end{bmatrix} \in \mathbb{R}^{N \times 2d^2}.$$

(72)

With the notations defined above, we state the following theorem:

**Theorem E.2.** *Let the combined weight matrix be*

$$\boldsymbol{\Theta} \triangleq \begin{bmatrix} \boldsymbol{A} \\ \boldsymbol{B}^\top \end{bmatrix} \in \mathbb{R}^{2d \times d},$$

*and consider the loss function $\tilde{\mathcal{L}}$ defined in (67). Denote*

$$\sigma_{\min} \triangleq \sigma_{\min}(J(\boldsymbol{\Theta}(0))), \quad \sigma_{\max} \triangleq \sigma_{\max}(J(\boldsymbol{\Theta}(0))).$$

*If the initialization satisfies:*

$$\tilde{\mathcal{L}}(\boldsymbol{\Theta}(0)) \leq \frac{\sigma_{\min}^6}{1152 d \sigma_{\max}^2},$$

*then for every $t \geq 0$ the following hold:*

$$\tilde{\mathcal{L}}(\boldsymbol{\Theta}(t)) \leq \tilde{\mathcal{L}}(\boldsymbol{\Theta}(0)) \exp\left(-\frac{1}{2}\sigma_{\min}^2 t\right),$$

$$\|\boldsymbol{\Theta}(t) - \boldsymbol{\Theta}(0)\|_F \leq \frac{6\sqrt{2}\sigma_{\max}}{\sigma_{\min}^2}\sqrt{\tilde{\mathcal{L}}(\boldsymbol{\Theta}(0))}.$$

The above theorem tells us that, if the model is initialized with a sufficiently small loss, the model's loss will converge to zero quickly, and the parameters will not move significantly from the initialization. With the above theorem, we can state the following corollary:

**Corollary E.3.** *Suppose $\boldsymbol{A}$ and $\boldsymbol{B}$ are initialized as balanced, i.e.:*

$$\boldsymbol{A}(0)^\top \boldsymbol{A}(0) = \boldsymbol{B}(0)\boldsymbol{B}(0)^\top.$$

*Under the conditions of Theorem E.2, for every singular index $i \in [d]$ and all $t \geq 0$:*

$$\sigma_i(\boldsymbol{A}(t)) = \sigma_i(\boldsymbol{B}(t)) \quad \text{and} \quad |\sigma_i(\boldsymbol{A}(t)) - \sigma_i(\boldsymbol{A}(0))| \leq \frac{\sigma_{\min}}{4\sqrt{2d}}.$$

*Consequently, the stable rank of $\boldsymbol{A}(t)$ remains bounded below by*

$$\frac{\|\boldsymbol{A}(t)\|_F^2}{\|\boldsymbol{A}(t)\|_2^2} \geq \left(\frac{\|\boldsymbol{A}(0)\|_F - \frac{\sigma_{\min}}{4\sqrt{2d}}}{\|\boldsymbol{A}(0)\|_2 + \frac{\sigma_{\min}}{4\sqrt{2d}}}\right)^2.$$

### E.3.1 PROOF OF THEOREM E.2

We begin the proof of the theorem by noting that the Jacobian $J(\cdot)$ is a Lipschitz function, as stated in the following lemma:

**Lemma E.9.** *The Jacobian matrix $J(\boldsymbol{W})$, as defined in (72), is $\sqrt{d}$-Lipschitz. Specifically, for any matrices $\boldsymbol{W}, \boldsymbol{V} \in \mathbb{R}^{2d \times d}$, the following inequality holds:*

$$\|J(\boldsymbol{W}) - J(\boldsymbol{V})\| \leq \sqrt{d}\|\mathrm{vec}(\boldsymbol{W}) - \mathrm{vec}(\boldsymbol{V})\|. \tag{73}$$

*Proof.* Note that for each $n$-th observation,

$$J_n(\boldsymbol{\Theta}) = 2\mathrm{vec}\left(\bar{\boldsymbol{X}}_n\boldsymbol{\Theta}\right)^\top$$

$$= \mathrm{vec}\left(\begin{pmatrix} 0 & \boldsymbol{X}_n \\ \boldsymbol{X}_n^\top & 0 \end{pmatrix}\begin{pmatrix} \boldsymbol{A} \\ \boldsymbol{B}^\top \end{pmatrix}\right)^\top$$

$$= \mathrm{vec}\left(\begin{pmatrix} \boldsymbol{X}_n\boldsymbol{B}^\top \\ \boldsymbol{X}_n^\top\boldsymbol{A} \end{pmatrix}\right)^\top \in \mathbb{R}^{2d^2}.$$

Let $\boldsymbol{M}_l$ denote the $l$-th row of a matrix $\boldsymbol{M}$, and let $\boldsymbol{M}_{.,l}$ denote its $l$-th column. We have

$$\|J_n(\boldsymbol{\Theta})\|_F^2 = \|\boldsymbol{X}_n^\top\boldsymbol{A}\|_F^2 + \|\boldsymbol{X}_n\boldsymbol{B}^\top\|_F^2$$

$$= \|\boldsymbol{e}_{j_n}\boldsymbol{e}_{i_n}^\top\boldsymbol{A}\|_F + \|\boldsymbol{e}_{i_n}\boldsymbol{e}_{j_n}^\top\boldsymbol{B}^\top\|_F$$

$$= \|\boldsymbol{A}_{i_n}\|_2^2 + \|\boldsymbol{B}_{.,j_n}\|_2^2.$$

Now, suppose we observe all entries, i.e., $N = d^2$. Then for any fixed $n$, $i_n = i_m$ can be satisfied for all $m \in [d]$, meaning each element of $\boldsymbol{A}$ is observed $d$ times. Similarly, each element of $\boldsymbol{B}$ is also observed $d$ times.

Therefore, we can upper bound the Frobenius norm of the Jacobian matrix by the Frobenius norm of the Jacobian under full observation:

$$\|J(\boldsymbol{\Theta})\|_F^2 \leq \sum_{n=1}^{d^2}\left(\|\boldsymbol{X}_n^\top\boldsymbol{A}\|_F^2 + \|\boldsymbol{X}_n\boldsymbol{B}^\top\|_F^2\right)$$

$$= d\left(\|\boldsymbol{A}\|_F^2 + \|\boldsymbol{B}\|_F^2\right)$$

$$= d\|\boldsymbol{\Theta}\|_F^2.$$

By upper-bounding the spectral norm of the difference between two Jacobian matrices and applying the inequality above, we obtain:

$$\|J(\boldsymbol{W}) - J(\boldsymbol{V})\|^2 = \|J(\boldsymbol{W} - \boldsymbol{V})\|^2$$

$$\leq \|J(\boldsymbol{W} - \boldsymbol{V})\|_F^2$$

$$\leq d\|\boldsymbol{W} - \boldsymbol{V}\|_F^2,$$

which concludes the proof. $\qquad\square$

Next, we borrow a lemma from Telgarsky (2021), which states that for a Lipschitz function $J$, if we consider a sufficiently small neighborhood around the initialization $\boldsymbol{\Theta}(0)$, then the singular values of the Jacobian $J(\boldsymbol{\Theta})$ remain close to those at initialization:

**Lemma E.10** (Lemma 8.3 in Telgarsky (2021))**.** *If we suppose* $\|\mathrm{vec}(\boldsymbol{\Theta}) - \mathrm{vec}(\boldsymbol{\Theta}(0))\| \leq \frac{\sigma_{\min}}{2\sqrt{d}}$, *we have the following:*

$$\sigma_{\min}(J(\boldsymbol{\Theta})) \geq \frac{\sigma_{\min}}{2}, \quad \sigma_{\max}(J(\boldsymbol{\Theta})) \leq \frac{3\sigma_{\max}}{2},$$

*where we denote* $\sigma_{\min} \triangleq \sigma_{\min}(J(\boldsymbol{\Theta}(0)))$, *and* $\sigma_{\max} \triangleq \sigma_{\max}(J(\boldsymbol{\Theta}(0)))$.

For simplicity, we denote $\boldsymbol{\theta}$ as the vectorized version of $\boldsymbol{\Theta}$, i.e., $\boldsymbol{\theta} \triangleq \mathrm{vec}(\boldsymbol{\Theta})$. We define the time step $\tau$, which is the first time step when the trajectory of $\boldsymbol{\theta}(t)$ touches the boundary:

$$\tau \triangleq \inf_{t \geq 0}\left\{t \mid \|\boldsymbol{\theta}(t) - \boldsymbol{\theta}(0)\| \geq \frac{\sigma_{\min}}{2\sqrt{d}}\right\}.$$

We now demonstrate the convergence of the loss when $t \in [0, \tau]$ using the following lemma.

**Lemma E.11.** *For all* $t \in [0, \tau]$, *the loss defined in (67) converges as follows:*

$$\tilde{\mathcal{L}}(\boldsymbol{\Theta}(t)) \leq \tilde{\mathcal{L}}(\boldsymbol{\Theta}(0))\exp\left(-\frac{1}{2}\sigma_{\min}^2 t\right),$$

*where we define* $\sigma_{\min} \triangleq \sigma_{\min}(J(\boldsymbol{\Theta}(0)))$.

*Proof.* Recall that the time derivative of the loss can be written as follows, according to (71):

$$\dot{\tilde{\mathcal{L}}}(\boldsymbol{\Theta}(t)) = -\left(F(\boldsymbol{\Theta}(t)) - \boldsymbol{y}\right)^{\top} J(\boldsymbol{\Theta}(t))\, \dot{\boldsymbol{\theta}}(t)$$
$$= -\left(F(\boldsymbol{\Theta}(t)) - \boldsymbol{y}\right)^{\top} J(\boldsymbol{\Theta}(t)) J(\boldsymbol{\Theta}(t))^{\top} \left(F(\boldsymbol{\Theta}(t)) - \boldsymbol{y}\right),$$

noting that

$$\dot{\boldsymbol{\theta}}(t) = -\nabla_{\boldsymbol{\theta}(t)}\tilde{\mathcal{L}}(\boldsymbol{\Theta}(t)) = -J(\boldsymbol{\Theta}(t))^{\top} \left(F(\boldsymbol{\Theta}(t)) - \boldsymbol{y}\right).$$

By Lemma E.10, for any $t \in [0, \tau]$, we can upper bound the above term as follows:

$$\dot{\tilde{\mathcal{L}}}(\boldsymbol{\Theta}(t)) \le -\lambda_{\min}\left(J(\boldsymbol{\Theta}(t))J(\boldsymbol{\Theta}(t))^{\top}\right) \|F(\boldsymbol{\Theta}(t)) - \boldsymbol{y}\|^2$$
$$\le -\frac{1}{2}\sigma_{\min}^2 \tilde{\mathcal{L}}(\boldsymbol{\Theta}(t)).$$

Applying Grönwall's inequality gives:

$$\tilde{\mathcal{L}}(\boldsymbol{\Theta}(t)) \le \tilde{\mathcal{L}}(\boldsymbol{\Theta}(0)) \exp\left(-\frac{1}{2}\sigma_{\min}^2 t\right) \quad \text{for } t \in [0, \tau].$$

$\square$

The above lemma shows that the loss decays rapidly to zero if $\boldsymbol{\theta}(t)$ stays within a small neighborhood around the initialization. We now show that if the loss converges quickly near initialization, then $\boldsymbol{\theta}(t)$ does not move far from its initial value:

**Lemma E.12.** *Let $\sigma_{\min} \triangleq \sigma_{\min}(J(\boldsymbol{\Theta}(0)))$ and $\sigma_{\max} \triangleq \sigma_{\max}(J(\boldsymbol{\Theta}(0)))$. For all $t \in [0, \tau]$, the distance between the weight vector at time $t$ and the initial weight vector is bounded by:*

$$\|\boldsymbol{\theta}(t) - \boldsymbol{\theta}(0)\| \le \frac{6\sqrt{2}\sigma_{\max}}{\sigma_{\min}^2} \sqrt{\tilde{\mathcal{L}}(\boldsymbol{\Theta}(0))}.$$

*Proof.* We start by evaluating the distance between $\boldsymbol{\theta}(t)$ and $\boldsymbol{\theta}(0)$ using Lemma E.10:

$$\|\boldsymbol{\theta}(t) - \boldsymbol{\theta}(0)\| = \left\|\int_0^t \dot{\boldsymbol{\theta}}(s)\, \mathrm{d}s\right\|$$
$$= \int_0^t \left\|J(\boldsymbol{\Theta}(s))^{\top} \left(F(\boldsymbol{\Theta}(s)) - \boldsymbol{y}\right)\right\| \mathrm{d}s$$
$$\le \int_0^t \sigma_{\max}(J(\boldsymbol{\Theta}(s))) \|F(\boldsymbol{\Theta}(s)) - \boldsymbol{y}\| \, \mathrm{d}s$$
$$\le \frac{3}{2}\sigma_{\max} \int_0^t \|F(\boldsymbol{\Theta}(s)) - \boldsymbol{y}\| \, \mathrm{d}s.$$

By Lemma E.11, we know that the objective function $\tilde{\mathcal{L}}(\boldsymbol{\Theta})$ satisfies:

$$\|F(\boldsymbol{\Theta}(t)) - \boldsymbol{y}\|^2 \le \|F(\boldsymbol{\Theta}(0)) - \boldsymbol{y}\|^2 \exp\left(-\frac{1}{2}\sigma_{\min}^2 t\right).$$

Taking the square root of both sides, we obtain:

$$\|F(\boldsymbol{\Theta}(t)) - \boldsymbol{y}\| \le \|F(\boldsymbol{\Theta}(0)) - \boldsymbol{y}\| \exp\left(-\frac{1}{4}\sigma_{\min}^2 t\right).$$

Substituting this into the previous inequality:

$$\|\boldsymbol{\theta}(t) - \boldsymbol{\theta}(0)\| \le \frac{3}{2}\sigma_{\max}\|F(\boldsymbol{\Theta}(0)) - \boldsymbol{y}\| \int_0^t \exp\left(-\frac{1}{4}\sigma_{\min}^2 s\right) \mathrm{d}s$$
$$\le \frac{6\sigma_{\max}}{\sigma_{\min}^2}\|F(\boldsymbol{\Theta}(0)) - \boldsymbol{y}\|,$$

where we used the fact that:

$$\int_0^t \exp(-Cs)\, \mathrm{d}s \le \frac{1}{C}, \quad \text{for } C > 0.$$

$\square$

By combining Lemmas E.11 and E.12, we obtain the following results:

$$\tilde{\mathcal{L}}(\boldsymbol{\Theta}(t)) \leq \tilde{\mathcal{L}}(\boldsymbol{\Theta}(0)) \exp\left(-\frac{1}{2}\sigma_{\min}^2 t\right), \tag{74}$$

$$\|\boldsymbol{\theta}(t) - \boldsymbol{\theta}(0)\| \leq \frac{6\sqrt{2}\sigma_{\max}}{\sigma_{\min}^2}\sqrt{\tilde{\mathcal{L}}(\boldsymbol{\Theta}(0))}, \tag{75}$$

which hold for $t \in [0, \tau]$. If we can demonstrate that $\tau = \infty$, the proof is complete.

Actually, if we initialize $\boldsymbol{\Theta}(0)$ to satisfy the condition:

$$\tilde{\mathcal{L}}(\boldsymbol{\Theta}(0)) \leq \frac{\sigma_{\min}^6}{1152d\sigma_{\max}^2},$$

and substitute this condition into (75), we obtain an upper bound for $\|\boldsymbol{\theta}(t) - \boldsymbol{\theta}(0)\|$:

$$\|\boldsymbol{\theta}(t) - \boldsymbol{\theta}(0)\| \leq \frac{6\sqrt{2}\sigma_{\max}}{\sigma_{\min}^2}\frac{\sigma_{\min}^3}{\sqrt{1152d}\sigma_{\max}} = \frac{\sigma_{\min}}{4\sqrt{d}}.$$

Recall the definition of $\tau$, which is the first time when $\boldsymbol{\theta}(t)$ touches the boundary of the small ball around the initialization:

$$\tau \triangleq \inf_{t \geq 0}\left\{t \mid \|\boldsymbol{\theta}(t) - \boldsymbol{\theta}(0)\| \geq \frac{\sigma_{\min}}{2\sqrt{d}}\right\}.$$

However, with the condition $\tilde{\mathcal{L}}(\boldsymbol{\Theta}(0)) \leq \frac{\sigma_{\min}^6}{1152d\sigma_{\max}^2}$, $\boldsymbol{\theta}(t)$ cannot ever touch the boundary. This is because $\|\boldsymbol{\theta}(t) - \boldsymbol{\theta}(0)\|$ is bounded above by $\frac{\sigma_{\min}}{4\sqrt{d}}$, which is strictly less than $\frac{\sigma_{\min}}{2\sqrt{d}}$. Therefore, the parameter will remain inside the ball indefinitely, meaning $\tau = \infty$. This completes the proof of the theorem.

### E.3.2 PROOF OF COROLLARY E.3

First, we establish the equality $\sigma_i(\boldsymbol{A}(t)) = \sigma_i(\boldsymbol{B}(t))$ for all $i \in [d]$. Corollary E.3 assumes that $\boldsymbol{A}(0)$ and $\boldsymbol{B}(0)$ are initialized as "balanced", satisfying $\boldsymbol{A}(0)^\top\boldsymbol{A}(0) = \boldsymbol{B}(0)\boldsymbol{B}(0)^\top$. By Lemma F.4, this balanced condition ensures that the singular values of $\boldsymbol{A}(t)$ and $\boldsymbol{B}(t)$ remain identical for all $t \geq 0$:

$$\sigma_i(\boldsymbol{A}(t)) = \sigma_i(\boldsymbol{B}(t)).$$

Second, we address the change in the singular values of a combined parameter matrix $\boldsymbol{\Theta}(t)$ (related to $\boldsymbol{A}(t)$ and $\boldsymbol{B}(t)$). Theorem E.2 states that under a specified condition on the initial loss, $\tilde{\mathcal{L}}(\boldsymbol{\Theta}(0)) \leq \frac{\sigma_{\min}^6}{1152d\sigma_{\max}^2}$, the deviation of $\boldsymbol{\Theta}(t)$ from its initialization $\boldsymbol{\Theta}(0)$ is bounded for all $t \geq 0$ by:

$$\|\boldsymbol{\Theta}(t) - \boldsymbol{\Theta}(0)\|_F \leq \frac{\sigma_{\min}}{4\sqrt{d}}.$$

Let $K = \frac{\sigma_{\min}}{4\sqrt{d}}$. By Weyl's inequality, $|\sigma_i(\boldsymbol{X}) - \sigma_i(\boldsymbol{Y})| \leq \|\boldsymbol{X} - \boldsymbol{Y}\|_2$, and noting that $\|\cdot\|_2 \leq \|\cdot\|_F$, we have for all $i \in [d]$:

$$\begin{aligned}
|\sigma_i(\boldsymbol{\Theta}(t)) - \sigma_i(\boldsymbol{\Theta}(0))| &\leq \|\boldsymbol{\Theta}(t) - \boldsymbol{\Theta}(0)\|_2 \\
&\leq \|\boldsymbol{\Theta}(t) - \boldsymbol{\Theta}(0)\|_F \\
&\leq K.
\end{aligned}$$

This inequality allows us to establish bounds for $\|\boldsymbol{\Theta}(t)\|_F$ (using reverse triangle inequality) and its largest singular value $\sigma_1(\boldsymbol{\Theta}(t)) = \|\boldsymbol{\Theta}(t)\|_2$:

$$\begin{aligned}
\|\boldsymbol{\Theta}(t)\|_F &\geq \|\boldsymbol{\Theta}(0)\|_F - K, \\
\sigma_1(\boldsymbol{\Theta}(t)) &\leq \sigma_1(\boldsymbol{\Theta}(0)) + K.
\end{aligned}$$

This yields the following lower bound on the stable rank of $\boldsymbol{\Theta}(t)$:

$$\frac{\|\boldsymbol{\Theta}(t)\|_F^2}{\|\boldsymbol{\Theta}(t)\|_2^2} \geq \left(\frac{\|\boldsymbol{\Theta}(0)\|_F - K}{\sigma_1(\boldsymbol{\Theta}(0)) + K}\right)^2 = \left(\frac{\|\boldsymbol{\Theta}(0)\|_F - \frac{\sigma_{\min}}{4\sqrt{d}}}{\|\boldsymbol{\Theta}(0)\|_2 + \frac{\sigma_{\min}}{4\sqrt{d}}}\right)^2.$$

Furthermore, the balancedness condition implies $A(t)^\top A(t) = B(t)B(t)^\top$. By the definition of $\Theta(t)$, $\Theta(t)^\top \Theta(t) = A(t)^\top A(t) + B(t)B(t)^\top$, this leads to $\Theta(t)^\top \Theta(t) = 2A(t)^\top A(t)$. This relationship implies $\sigma_i(\Theta(t)) = \sqrt{2}\sigma_i(A(t))$ for all $i$. Substituting this into the bounds for $\Theta(t)$, we have

$$\|A(t)\|_F \geq \|A(0)\|_F - K/\sqrt{2},$$
$$\|A(t)\|_2 \leq \|A(0)\|_2 + K/\sqrt{2}.$$

This leads to the final lower bound on the stable rank of $A(t)$ (which, by balancedness, is equal to that of $B(t)$):

$$\frac{\|A(t)\|_F^2}{\|A(t)\|_2^2} \geq \left( \frac{\|A(0)\|_F - K/\sqrt{2}}{\|A(0)\|_2 + K/\sqrt{2}} \right)^2 = \left( \frac{\|A(0)\|_F - \frac{\sigma_{\min}}{4\sqrt{2d}}}{\|A(0)\|_2 + \frac{\sigma_{\min}}{4\sqrt{2d}}} \right)^2.$$

# F  USEFUL LEMMAS

**Lemma F.1** (Adaptation of Lemma 1 and Theorem 3 in Arora et al. (2019)). *For any time $t$, the product matrix $\boldsymbol{W}(t) \in \mathbb{R}^{d,d}$ can be decomposed into its singular value decomposition:*

$$\boldsymbol{W}(t) = \sum_{r=1}^{d} \sigma_r(t) \boldsymbol{u}_r(t) \boldsymbol{v}_r(t)^\top$$

*where $\sigma_r(t)$ are the singular values of $\boldsymbol{W}(t)$, and $\boldsymbol{u}_r(t)$, $\boldsymbol{v}_r(t)$ are the corresponding left and right singular vectors, respectively. Moreover, if $\boldsymbol{A}, \boldsymbol{B}$ are balanced at initialization, i.e.,*

$$\boldsymbol{A}^\top(0)\boldsymbol{A}(0) = \boldsymbol{B}(0)\boldsymbol{B}^\top(0),$$

*the time evolution of the singular values $\sigma_r(t)$ is represented as:*

$$\dot{\sigma}_r(t) = -2 \cdot \sigma_r(t) \cdot \left\langle \nabla\ell(\boldsymbol{W}(t)), \boldsymbol{u}_r(t)\boldsymbol{v}_r(t)^\top \right\rangle, \quad r = 1, \ldots, d \tag{76}$$

**Lemma F.2.** *For any real-valued square matrix $\boldsymbol{A} \in \mathbb{R}^{d\times d}$, the absolute value of its determinant equals the product of its singular values:*

$$|\det(\boldsymbol{A})| = \prod_{r=1}^{d} \sigma_r$$

*where $\sigma_r$ are the singular values of $\boldsymbol{A}$.*

*Proof.* We express $\boldsymbol{A}$ using SVD: $\boldsymbol{A} = \boldsymbol{U}\boldsymbol{\Sigma}\boldsymbol{V}^\top$. Applying the determinant to both sides, we get:

$$\det(\boldsymbol{A}) = \det(\boldsymbol{U}\boldsymbol{\Sigma}\boldsymbol{V}^\top)$$
$$= \det(\boldsymbol{U})\det(\boldsymbol{\Sigma})\det(\boldsymbol{V}^\top)$$

Here, $\boldsymbol{U}$ and $\boldsymbol{V}$ have orthonormal columns, and $\boldsymbol{\Sigma}$ is diagonal with singular values along its main diagonal. Since the determinant of an orthonormal matrix is either $\pm 1$,

$$|\det(\boldsymbol{A})| = \det(\boldsymbol{\Sigma}) = \prod_{r=1}^{d} \sigma_r.$$

$\square$

**Lemma F.3** (Determinant of $\boldsymbol{A}(t)$). *Consider a matrix $\boldsymbol{A}(t) \in \mathbb{R}^{d,d}$ initialized as $\det(\boldsymbol{A}(0)) > 0$. Then, $\det(\boldsymbol{A}(t)) > 0$ for all $t \geq 0$.*

*Proof.* This follows directly from Lemma F.1 and F.2. Since the singular values are initialized as positive, and their evolution is continuous according to the given differential equation, they cannot become zero or negative. Therefore, $\boldsymbol{A}(t)$ maintains its sign of the determinant at initialization throughout the optimization process. $\square$

**Lemma F.4** (Adaptation of Lemma 8 in Razin & Cohen (2020)). *Consider a product matrix $\boldsymbol{W}(t) = \boldsymbol{A}(t)\boldsymbol{B}(t) \in \mathbb{R}^{d\times d}$, where $\boldsymbol{A}(t)$ and $\boldsymbol{B}(t)$ are of equal size and balanced at initialization. Under these conditions, the following equality holds for all $t \geq 0$ and all singular values:*

$$\sigma_r\left(\boldsymbol{W}(t)\right) = \sigma_r\left(\boldsymbol{A}(t)\right)^2 = \sigma_r\left(\boldsymbol{B}(t)\right)^2$$

*where $\sigma_r(\cdot)$ denotes the $r$-th singular value of the respective matrix where $r \in [d]$. Moreover, if $\det\left(\boldsymbol{A}(0)\right)$ and $\det\left(\boldsymbol{B}(0)\right)$ are both positive, then by Lemma F.3, we can guarantee that for all $t \geq 0$:*

$$\det\left(\boldsymbol{W}(t)\right) = \det\left(\boldsymbol{A}(t)\right)^2 = \det\left(\boldsymbol{B}(t)\right)^2$$

**Lemma F.5** (Adaptation of Theorem 1 in Arora et al. (2019)). *Consider a product matrix $\boldsymbol{W}(t) = \boldsymbol{A}(t)\boldsymbol{B}(t) \in \mathbb{R}^{d\times d}$. We can guarantee $\boldsymbol{A}(t)$ and $\boldsymbol{B}(t)$ are analytic functions of $t$. As a result, $\boldsymbol{W}(t)$ is also an analytic function of $t$.*

**Lemma F.6** (Lemma 10 in Razin & Cohen (2020)). *Let $f, g : [0, \infty] \to \mathbb{R}$ be real analytic functions such that $f^{(k)}(0) = g^k(0)$ for all $k \in \mathbb{N} \cup \{0\}$. Then, $f(t) = g(t)$ for all $t \geq 0$.*

**Lemma F.7** (Positive Semidefiniteness of $\boldsymbol{ABA}^\top$). *For matrices $\boldsymbol{A}, \boldsymbol{B} \in \mathbb{R}^{d,d}$, if $\boldsymbol{B}$ is positive semi-definite, then both $\boldsymbol{ABA}^\top$ and $\boldsymbol{A}^\top \boldsymbol{BA}$ are positive semi-definite.*

*Proof.* For any vector $\boldsymbol{x} \in \mathbb{R}^d$:

$$\boldsymbol{x}^\top \boldsymbol{ABA}^\top \boldsymbol{x} = (\boldsymbol{A}^\top \boldsymbol{x})^\top \boldsymbol{B} (\boldsymbol{A}^\top \boldsymbol{x}) \geq 0$$

since $\boldsymbol{B}$ is a positive semi-definite matrix. In the same way, for any vector $\boldsymbol{x} \in \mathbb{R}^d$ we have:

$$\boldsymbol{x}^\top \boldsymbol{A}^\top \boldsymbol{BA} \boldsymbol{x} = (\boldsymbol{Ax})^\top \boldsymbol{B} (\boldsymbol{Ax}) \geq 0$$

which concludes the proof. $\square$

