# OpenReview forum: "Implicit Bias and Loss of Plasticity in Matrix Completion: Depth Promotes Low-Rankness"
_ICLR.cc/2026/Conference — ICLR 2026 Poster_

### Official Review · Reviewer_DsYE · 2025-10-25

**Soundness:** 3
**Presentation:** 3
**Contribution:** 2
**Rating:** 4
**Confidence:** 4

**Summary:**

The paper studies implicit low-rank bias in deep linear matrix factorization for matrix completion and connects it to loss of plasticity when warm-starting from partially trained solutions. The core thesis is that while L=2 models exhibit coupling only when the observation pattern connects entries, depth >= 3 architectures induce structural coupling regardless of observation connectivity, yielding a stronger implicit bias toward low rank. The paper provides theoretical results on coupled/decoupled regime and on plasticity loss under warm starts supported by toy experiments.

**Strengths:**

* Provides an intuitive coupled vs. decoupled framework explaining how depth alone can produce low-rank bias.
* Theoretical results are mathematically careful, with complete proofs and well-presented figures.
* The paper connects depth-induced bias to plasticity loss, an emerging topic in optimization and continual learning.
* Writing and presentation are clear and well-formatted.

**Weaknesses:**

The main theoretical contributions rely heavily on diagonal observation structures and specific initializations which limits generalizablity of said findings; it is also unclear how the results extend to generic or noisy data patterns. Much of the clean spectral characterization is within this stylized regime. Additionally, the finding that deeper (linear) networks inherently exhibit a stronger
low-rank bias than shallow networks is not an entirely novel finding though I appreciate that the novelty here is the coupling mechanism + solvable diagonal case + plasticity proofs. For other weaknesses and more details, see questions below.

**Questions:**

* Would it be possible to go beyond diagonals? Can the authors extend analysis/conclusions to non-diagonal, sparse patterns (even banded or block-diagonal) to demonstrate generality of the depth-induced bias? The main theorem(s) only seems to cover an extremely stylized regime which is much narrower than generic matrix completion settings. Right now, I believe this setup is quite limiting to the generalizability of said findings as this makes the conclusions hinge on a stylized regime.

* How robust are the results to generic random initialization and to noisy/unequal diagonal entries? Can the authors provide some characterization of formal continuity/stability arguments in these cases?

* Given that many works nowadays, both in theoretical analysis and empirical methods, focus on adaptive optimizers (e.g., Adam and its variants etc.), I wonder how these findings change under these optimizers? Many works ([1][2] to just name a few with [1] also focusing on matrix completion) have explored the implicit bias of these types of optimizers with some focusing on the role of how adaptive optimizers provide pre-conditoning in certain tasks/testbeds similar to that of this paper.
  * Can the related works section be made a bit clearer in terms of distinguishing implicit reg/bias between 1st order and adaptive, etc. 2nd order optimizers (see below)? I think mentioning these works that focus on the implicit bias of adaptive optimizers (beyond just GD) like [1] and [2], among others, are worth contextualizing (I see the paper already cites the "The implicit bias of adam on separable data" but it's clumped together with several other works under a generic sense of implicit regularization for GD etc.); this would better help frame this paper's relevance and contribution while improving the flow. Or one could better structure related works according to the testbed/problem (e.g., compressed sensing, matrix factorization, etc.). I'm also not sure why the related works is in the appendix.
  * Adding optimizer ablations (Adam/Adagrad/SGD) would clarify whether depth-induced coupling is robust across optimizers; for instance, core results (Prop. 3.2, Thm. 3.3; Thm. 4.2/4.3) are proved under gradient flow but practical claims and CNN experiments implicitly target regimes where training with Adam etc. is typically more standard. It's also unclear whether the depth-induced coupling and ensuing low-rank spectra hold beyond the GF/SGD limit. I view this optimizer-dependence gap as a key reason to temper claims of practical significance (despite the mention of practical tests using CNNs etc.).

* For figs. 10-13 in Appendix, for the CNN/resnet experiments, could the authors please detail how convolutional kernels are reshaped, which layers are measured, provide seeds/variance/CI to support trends, etc.

* The plasticity theorems are insightful but tend to be confined to toy regimes.

[1] "Combining Explicit and Implicit Regularization for Efficient Learning in Deep Networks" Zhao et al. (2022)

[2] "On the Implicit Bias of Adam" Cattaneo et al. (2024)

---

> ### Author Response · Authors · 2025-11-22
>
> We sincerely appreciate your thoughtful and constructive feedback. Below, we address your concerns and questions point by point.
>
> ## **W1. Q1. Main theoretical contributions rely too heavily on diagonal observations**
>
> Thank you for this comment and for the constructive suggestion. First, we would like to emphasize that **even the diagonal observation setting we study is already nontrivial**. The training dynamics are still nonlinearly coupled, so the analysis remains technically challenging. As we note in the paper, Menon (2024) points out that even the problem of completing a $2 \times 2$ matrix with identical diagonal observations using deep matrix factorization under gradient flow remains open. Our results, including Theorem 3.3, take a step toward resolving this problem in a more general $d \times d$ case. We hope this clarifies that the diagonal setup is not a purely toy simplification, but a nontrivial contribution in its own right.
>
> Motivated by this concern, we have extended Theorem 3.3 beyond purely diagonal observations in the revised version. In particular, as you suggested, Theorem D.3 in Appendix D.5 treats **block-diagonal observations**, and Figure 33 confirms the same qualitative depth dependent low-rank behavior as in the diagonal case. For a more detailed discussion of this extension, we refer the reviewer to the global response.
>
> ## **W2. Q2. Robustness to initialization and noisy/non-uniform diagonal entries**
>
> Thank you for this question. For generic random initialization, including Gaussian schemes, we discuss robustness and our choice of deterministic initialization in more detail in the Global Response under “Initialization dependence.” Here we briefly address the case of noisy or unequal diagonal entries.
>
> Regarding robustness to noisy or unequal diagonal entries, our current proofs rely on a certain amount of structure to make the nonlinearly coupled ODE system analytically tractable. When the observation pattern or diagonal structure is perturbed in a fully generic way, the symmetry that allows us to track the singular values in a fixed basis is lost, and the resulting coupled dynamics are substantially harder to analyze. Even in the structured diagonal and block-diagonal cases we study (Theorem 3.3 and Theorem D.3), the analysis already requires nontrivial work. The aforementioned open problem by Menon (2024) also concerns **identical diagonal observations**.
>
> To probe robustness empirically, **we added experiments with noisy diagonals (Figure 10) and non-equal diagonal entries (Figure 11) in Appendix C.1.** In both settings, we observe the same qualitative trend as in the idealized diagonal case: depth-2 models remain high-rank across initialization scales, while deeper models with small initialization exhibit a stronger low-rank bias, and the behavior changes continuously as the noise level is varied.
>
> From the ODE perspective, gradient flow depends continuously on the observations and initialization, so small perturbations of the diagonal values should lead to small changes in the limiting singular values. A full, quantitative continuity or stability theory for generic random initialization and arbitrary noisy diagonals is beyond the scope of the current paper, but we view our structured setting as a first step that isolates the coupled–decoupled mechanism in a regime where the analysis is still tractable.
>
> ## **Q3. Extension to adaptive optimizers**
>
> Thank you for raising this point. In the present paper we focus on gradient flow (gradient descent with small step size), which are the standard baseline in much of the implicit bias literature [1-6] and allow us to **isolate the effect of depth and coupling in a relatively clean dynamical system**.
>
> Regarding adaptive optimizers, we agree that understanding how our findings extend to methods such as Adam is an important direction for future work. [7], for example, derive an effective Adam flow for singular values in a matrix completion setting under strong assumptions, but their analysis does not characterize what happens at convergence or identify the limiting singular values, which is similar in spirit to the limitations of [1]. [8] likewise study the implicit bias of Adam, but in different problem setups. We briefly discuss and compare these works to ours in the Related Work section in Appendix A.
>
> Our mechanism is formulated at the level of gradient flow, where we can explicitly see how depth and coupling shape the limiting singular values. Introducing adaptive preconditioning changes the geometry of the dynamics and would likely modify the quantitative form of this mechanism. A careful extension of our analysis to Adam type flows in the same matrix completion setting would require substantial additional technical work and is beyond the scope of the current paper, but we view this as a natural and important avenue for future research.

---

> > ### Author Response · Authors · 2025-11-22
> >
> > ## **Q3.1. Structure and placement of Related work**
> >
> > We agree that a clearer separation between works on first order methods and those on adaptive or second order optimizers can help frame our contributions. In the revised version, we have addressed this by restructuring the related work in Appendix A to more clearly distinguish:
> >
> > 1. implicit regularization of first order methods (gradient flow / gradient descent), and
> > 2. implicit bias of adaptive optimizers such as Adam and its variants, including [7, 8].
> >
> > Due to space constraints in the main text, we keep the detailed literature review in the appendix, but we have added explicit pointers from the main paper to Appendix A (Line 128) and ensured that the main text contains enough context for readers to understand our contributions without reading the full appendix.
> >
> > ## **Q3.2. Adding optimizer ablations**
> >
> > Following your suggestion, we have extended our experiments, both in the matrix completion setting and in practical neural networks, to a broader set of optimizers.
> >
> > In the matrix completion setting, we now include results for SGD, GD with momentum, Adam, RMSProp, and Adagrad. Specifically, we run gradient based optimization with a small stepsize under Gaussian initialization with identical diagonal observations. The results in Figures 12–16 in Appendix C.1 align well with our theory: for depth-2, the model converges to high-rank solutions across initialization scales, whereas for depth $\geq 3$ the solutions become increasingly low-rank as the initialization scale decreases and as depth increases.
> >
> > We have also added optimizer ablations in neural network experiments. In addition to the SGD with momentum results in Figures 17–20, the revised Appendix C.1.1 now includes:
> >
> > - Figures 21–24: ResNet/VGG experiments trained with Adam, and
> > - Figures 25–28: ResNet/VGG experiments trained with RMSProp, a variant of AdaGrad with exponential discounting of past squared gradients.
> >
> > Across all of these settings, we observe the same qualitative trend as in the SGD with momentum case: increasing depth consistently drives the weight matrices toward lower-rank structure. These optimizer ablations suggest that the depth-induced low-rank behavior is robust across a range of commonly used optimizers, not just in the gradient-flow or SGD limit. We have also added a pointer to these experiments in the main text (Lines 348–351).
> >
> > ## **Q4. Experimental details for CNN/ResNet results**
> >
> > We have now provided these details in the revised appendix (Lines 1519-1533). Specifically,
> >
> > 1. we reshape each convolutional kernel $\mathbf{Z} \in \mathbb{R}^{c_{\rm in} \times c_{\rm out} \times k_1 \times k_2}$ into a matrix $\mathbf{W} \in \mathbb{R}^{c_{\rm in} \times (c_{\rm out}k_1k_2)}$ following [9].
> > 2. we measure the effective rank for all layers except the final one and average them into a single scalar.
> > 3. we report averages over five independent runs with 95% confidence intervals.
> >
> > ## **Q5. Plasticity theorems confined to toy regimes**
> >
> > We acknowledge that the depth-2 model we analyze does not fully capture all aspects of practical neural networks. At the same time, our plasticity analysis is not restricted to the $2 \times 2$ example in Theorem 4.2. In Theorem 4.3 **we provide a full characterization for general $d \times d$ matrices**, which already goes beyond a purely toy setting. In this general case, **the observations may also contain noise**, as long as they satisfy the loss conditions required for our analysis. Even within this simplified matrix completion framework, the training dynamics form a nonlinearly coupled system of ODEs, and obtaining a rigorous description of such coupled dynamics is already highly nontrivial.
> >
> > The loss of plasticity phenomenon has been reported in many empirical studies, but its underlying mechanism is still not well understood. Much of the existing empirical work interprets loss of plasticity through the lens of lazy training, without providing a formal, theoretically grounded mechanism [10-12]. In contrast, our analysis in Section 4 provides, to the best of our knowledge, **the first theoretically rigorous account of loss of plasticity** in a concrete learning task. Our results provide a theoretical guarantee that connects lazy training behavior to loss of plasticity, thereby offering insights into why loss of plasticity emerges in practical neural network models, even though the model we analyze is simplified.

---

> > > ### Author Response · Authors · 2025-11-22
> > >
> > > ## **W3. Novelty of depth-induced low-rank bias in matrix completion**
> > >
> > > As far as we are aware, there are only a few works that study depth induced low-rank bias in the **matrix completion** setting. Among them, [1] do consider depth, but they cannot fully track the dynamics to prove low-rank convergence as network depth increases. Their analysis is primarily restricted to a regime where $t \geq t_0$, after which the singular vectors are assumed to have stabilized. For $t \geq t_0$, they show that one singular value can be expressed as a function of another, with a unknown constant term determined by the state at $t_0$. From this the authors argue that the gap between singular values grows with depth. However, **this result is not fully rigorous in terms of characterizing the limit, and it does not identify where the singular values converge in the limit point**.
> > >
> > > [2] also analyze a depth $\geq 2$ matrix completion problem in a $2 \times 2$ setting with three observations (one diagonal and two off diagonal entries). Their Theorems 1 and 2 show that the effective rank converges to its infimum when the loss converges, but **their analysis does not distinguish between the depth-2 and depth $\geq 3$ regimes**. In particular, it does not show that **deeper models exhibit a stronger low-rank bias, nor does it identify the underlying mechanism**. In addition, **their bounds are independent of the initialization scale,** so they do not capture the empirical observation that low-rank bias becomes stronger as the initialization scale decreases.
> > >
> > > In contrast, our Theorem 3.3 and Corollary 3.4 explicitly analyze the **limit point** of the singular values and separate the depth-2 and depth $\geq 3$ cases. We show that depth-2 models do not converge to low-rank under our setting, whereas for depth $\geq 3$ the coupled dynamics do converge to low-rank solutions and we identify the underlying mechanism for this depth-induced bias. Moreover, our results are **initialization scale dependent, which aligns with the empirical finding that smaller initialization scales strengthen low-rank bias.**
> > >
> > > If the reviewer had in mind works on matrix factorization task such as [5,6] (i.e., matrix completion with full observation), then we view this as a different regime. In the fully observed case, the **limiting solution is fixed** and, as $t$ grows, gradient flow converges to that solution; much of the literature there focuses on how singular values or features are learned incrementally along the trajectory. **In our matrix completion setting, by contrast, the limit point is not predetermined**, because the training objective admits infinitely many global solutions. It is precisely in this regime that we show deeper networks converge to lower rank solutions.
> > >
> > > If we are overlooking other references that already establish depth-induced low-rank bias in the matrix completion setting, we would be very grateful if you could point us to them.
> > >
> > > Thanks for your time and consideration.
> > >
> > > Best regards,
> > >
> > > Authors
> > >
> > > ---
> > >
> > > ### References
> > >
> > > 1. Implicit Regularization in Deep Matrix Factorization, NeurIPS 2019
> > > 2. Implicit Regularization in Deep Learning May Not Be Explainable by Norms, NeurIPS 2020
> > > 3. Implicit Regularization in Matrix Factorization, NIPS 2017
> > > 4. Implicit Regularization in Matrix Sensing via Mirror Descent, NeurIPS 2021
> > > 5. The Implicit Bias of Depth: How Incremental Learning Drives Generalization, ICLR 2020
> > > 6. Towards Resolving The Implicit Bias of Gradient Descent for Matrix Factorization: Greedy Low-rank Learning, ICLR 2021
> > > 7. Combining Explicit and Implicit Regularization for Efficient Learning in Deep Networks, NeurIPS 2022
> > > 8. On the Implicit Bias of Adam, ICML 2024
> > > 9. SGD and Weight Decay Secretly Minimize the Rank of Your Neural Network, arXiv:2206.05794
> > > 10. DASH: Warm-Starting Neural Network Training in Stationary Settings without Loss of Plasticity, NeurIPS 2024
> > > 11. A study on the plasticity of neural networks, arXiv 2106.00042
> > > 12. Critical Learning Periods Emerge Even in Deep Linear Networks, ICLR 2024

---

### Official Review · Reviewer_U2hd · 2025-10-29

**Soundness:** 3
**Presentation:** 3
**Contribution:** 2
**Rating:** 4
**Confidence:** 5

**Summary:**

This paper explores the use of deep matrix factorization for matrix completion tasks, with a focus on how network depth influences training dynamics and the implicit bias of neural networks. Building on the prior work of Bai et al. (2024), the paper emphasizes that coupled dynamics are the core mechanism behind this low rank bias. For deep networks, the dynamics are coupled, meaning that even if the observed entries are decoupled, the training process strongly tends toward a low-rank solution. This bias arises from the inherent interdependence between layers in deep factorizations, which naturally favors low-rank solutions, regardless of the observation pattern. The paper also discusses the phenomenon of plasticity loss in matrix completion tasks, where models struggle to adapt to new information after initial training, a problem particularly evident in shallow networks. The paper shows that deep models mitigate plasticity loss through their low-rank bias, enabling them to maintain good adaptability even when trained on limited data.

**Strengths:**

- This paper is well-written and has a clear overall structure, making it relatively easy to read.
- The paper addresses a key issue in matrix completion, specifically the role of network depth in shaping training dynamics and the implicit bias toward low-rank solutions. This focus provides valuable insights into how deeper networks perform better in this context.
- The paper builds on and extends the work of Bai et al. (2024), with a emphasis on the coupling dynamics and their relation to low-rank solutions. The paper does well to highlight the phenomenon of low-rank bias in deep networks, showing how deeper networks (L ≥ 3) exhibit stronger tendencies to converge to low-rank solutions, even when trained with limited data. The use of the theoretical framework surrounding coupled and decoupled dynamics is a solid foundation for exploring the behavior of deep networks in matrix completion tasks.
- The analysis of plasticity loss, especially the argument that deeper networks avoid this issue better than shallower networks, provides new insights to this phenomenon.

**Weaknesses:**

- While the paper’s theoretical results are solid, some claims feel weak or underexplored. For instance, Theorem 3.1 is focused on a very specific case (2x2 matrices), and while it does provide some insights, it lacks generality. Additionally, Theorem 3.3 is reduced to an implicit equation that is not deeply analyzed.
- Although numerical experiments (e.g., Figure 2) are used to validate the low-rank results, the theoretical argumentation feels limited. The paper would benefit from a more detailed analysis of how the implicit equation correlates with low-rank solutions, particularly for deeper networks (L ≥ 3).
- The definitions of "couple" and "decouple" (in the context of training dynamics) are important for understanding the paper’s core argument, but they have been discussed in similar terms in Bai et al. (2024) (Definition A.5). The paper would benefit from clearer citations to this previous work to avoid redundancy and ensure clarity.
- The discussion of special initializations, especially in Eq. (7), is quite restrictive and doesn't align well with practical network initialization strategies, which are typically random. This narrow focus on a specific initialization limits the applicability of the results.

**Questions:**

1. It seems to be focused on a very specific case (2x2 matrix), and while it provides some detailed characterization, it is weak overall. Additionally, I believe Bai et al. (2024)'s Theorem 2 offers a more general result that already characterizes the explicit relationship between initialization scale and alignment. Could the authors clarify how Theorem 3.1 adds value over this more general result?
2. The definitions of couple and decouple seem to have similar characterizations in Bai et al. (2024) (Definition A.5). Would it be possible to cite this source to avoid redundancy and clarify the connection?
3. Why does Eq. (7) consider such a special initialization? In practice, we usually consider random initializations rather than the specific initialization used here. Could the authors provide a justification for using this specific initialization in their analysis?
4. What is the relationship between the implicit equation and low-rank? Why does satisfying the implicit equation imply low-rank? Could the authors analyze how this implicit equation precisely corresponds to low rank, especially for the case where $L = 3$? I think this is the core argument of the paper (that coupling leads to low-rank solutions), but Theorem 3.3 reduces this to an implicit equation without further analysis. The low-rank results are instead verified through numerical experiments (e.g., Figure 2), which weakens the theoretical contribution. Could the authors provide a more detailed theoretical explanation of this?
5. Regarding Line 306: I think the argument that increasing depth promotes low rank might have a trivial reason behind it. For example, in the depth-2 case, the origin is a strict saddle point, and a very small initialization is required for the escape time to be long, which enables alignment. However, in deeper networks, the origin is no longer a strict saddle point, and the escape time significantly increases. This suggests that stacking more layers inherently leads to a smaller initialization scale. Comparing the rank under the same initialization scale might not be fair.
6. How do the authors argue the differences between matrix completion tasks and real-world neural network tasks in terms of data characteristics? Real neural networks typically sample continuous data, which is almost surely connected. In contrast, matrix completion tasks involve discrete sampling. Is this discreteness unique to matrix completion tasks?
7. In the Loss of Plasticity paper (Kleinman et al. 2024), the authors compare networks of different depths, finding that deeper networks show more pronounced phenomena at critical learning stages (Fig. 1). This seems contradictory to the claim that 3-layer networks exhibit more coupling than 2-layer networks, leading to low rank early on and thus avoiding plasticity loss. Could the authors discuss this discrepancy further?

---

> ### Author Response · Authors · 2025-11-22
>
> Thank you for your valuable feedback and for highlighting several important aspects of our work. We address your comments one by one below.
>
> ## **W1. Q1. Role of Theorem 3.1 and comparison with Bai et al. (2024), Theorem 2 and 3.**
>
> Thank you for this important question. We first want to note that our goal with Theorem 3.1 is not to claim generality, but to obtain a sharp, fully explicit characterization in a coupled setting where even such a low dimensional example is nontrivial to analyze.
>
> Bai et al. (2024) study infinitesimal initialization for general $d \times d$ matrix completion. Their Theorem 2 describes the local dynamics near a critical point on a rank-$k$ invariant manifold $\Omega_k$: under a small perturbation of size $\alpha$, the gradient flow escapes along the top Hessian eigenspace and enters $\Omega_{k+1}$ in the limit $\alpha \to 0$, under the assumptions that the residual has a unique largest singular value and that the critical point is second-order stationary. In Theorem 3, they further state that, if the trajectory attains the optimal solution inside each rank-level invariant manifold $\Omega_k$, and if the limit $\mathbf{W}\_c = \lim\_{\alpha \to 0} \mathbf{W}\_\infty (\alpha \mathbf{W}\_0)$ exists for a full-rank initialization $\mathbf{W}_0$, then $\mathbf{W}_c$ is a minimum-rank weight. **In summary, their result holds under several technical assumptions and in the asymptotic regime $\alpha \to 0$.**
>
> These results characterize the asymptotic behavior in the infinitesimal initialization regime and do not provide a quantitative bound on how aligned the learned factors are for a fixed (small but finite) initialization scale. In contrast, Theorem 3.1 focuses on a concrete 2×2 connected completion problem and gives an explicit upper bound on the misalignment $\lVert \mathbf{a}_i^\perp (\infty) \rVert_2^2 / \lVert \mathbf{a}_i(\infty)\rVert_2^2$ as a function of the initial norm $\lVert \mathbf{A}(0) \rVert_F$ and the initial scale of $\mathbf{b}_1(0)$. **This is a global, non-asymptotic statement about the final solution at finite initialization scale, which cannot be obtained from Theorem 2 or Theorem 3 of Bai et al. (2024).**
>
> While Theorem 3.1 is indeed a 2×2 “toy” setting, analyzing it is still nontrivial precisely because the gradient flow induces nonlinearly coupled dynamics. Working through this concrete example allows us to make the dependence on initialization scale fully explicit and to directly see how coupling drives the emergence of low-rank behavior. In this sense, Theorem 3.1 is complementary to Bai et al. (2024): their work provides a general asymptotic picture under technical assumptions and infinitesimal initialization, whereas Theorem 3.1 offers a detailed, non asymptotic characterization in a specific coupled example that illustrates the mechanism we aim to highlight.
>
> ## **W2, Q4. Detailed analysis of implicit equations**
>
> Thank you for raising this important question. We provide a more detailed discussion of this point in the Global Response under “Additional insight from the implicit characterization (8) and (9).” Here we briefly summarize the main idea.
>
> In the revised version, we clarify the additional insight provided by the implicit system (8), (9) by proving **Corollary 3.4** following Theorem 3.3. Under the setting $1 < m < \infty, d \geq 2, w^*>0$, and $L \geq 3$, Corollary 3.4 shows that as $\alpha \to 0$, the stable rank of $\mathbf{W}_{L:1}(\infty)$ converges to 1, while in the decoupled regimes $L=2$ or $m=\infty$, the stable rank remains close to $d$.
>
> By combining Theorem 3.3 with Corollary 3.4, we can summarize the behavior of the stable rank across all regimes considered in Theorem 3.3 and explicitly see how the coupled dynamics governed by (8) and (9) control the limiting stable rank. We have incorporated this discussion into the revised manuscript (Lines 323–333), and we believe that this comparison provides concrete analytic insights into the implicit system (8) and (9) and further strengthens Theorem 3.3.
>
> ## **W3, Q2. Comparison with Bai et al. (2024)’s definition of (de)coupled dynamics**
>
> Thank you for pointing this out. Bai et al. (2024) indeed introduce closely related terminology in their Definition A.5, where they describe coupled and decoupled training dynamics for depth-2 networks. We acknowledge that we should have clearly stated that this viewpoint was already discussed in Bai et al. (2024). In our revised version, we have now added an explicit citation and clarification at the point where Definition 2 is introduced (see Lines 237–238). There, we explain that Bai et al. (2024) define coupled and decoupled behavior for depth-2 networks, **while Definition 2 in our paper extends this notion to networks of arbitrary depth.**

---

> > ### Author Response · Authors · 2025-11-22
> >
> > ## **W4, Q3. Initialization Dependence**
> >
> > Thank you for raising this concern. A detailed discussion is provided in the Global Response under “Initialization dependence”; here we briefly summarize our perspective.
> >
> > From a dynamical systems viewpoint, the training dynamics form a high-dimensional, non-linear system of ODEs, and solving this system for generic random initializations such as Gaussian in order to exactly characterize the limiting solution is highly nontrivial. We therefore work with a structured but still rich family of deterministic initializations that extends the scaled identity ($\alpha \mathbf{I}_d$) initialization widely used in prior work, and that remains analytically tractable while preserving the key coupled versus decoupled behavior. In the revised version, we added a brief remark after the initialization description to clarify this motivation and its relation to Gaussian initialization.
> >
> > ## **Q5. On comparing ranks under the same initialization scale**
> >
> > We agree that your escape-time intuition around the origin is insightful. In particular, the fact that deeper networks can have slower escape dynamics near the origin is an important part of the picture, and your explanation is consistent with our understanding.
> >
> > However, this effect alone does not fully account for our observations. In both our analysis (via numerical solutions of the implicit equations) and our experiments, we set the initialization scale so that the scale of the product $\mathbf{W}_{L:1}(0)$ is comparable across depths. Concretely, **we match the scale of $\alpha^L$ on the x-axis in Figures 1-3, so deeper models are not simply started “closer to zero” than shallower ones under our comparison.**
> >
> > Moreover, Theorem 3.3 shows that the depth-2 model always converges to a full-rank solution, independently of $\alpha$. In contrast, for sufficiently small $\alpha$ and $L \geq 3$, the coupled dynamics produce a strictly larger gap between $\sigma_1$ and the other singular values, corresponding to a lower stable rank. This indicates a genuine depth-induced low-rank bias that **cannot be explained solely as a trivial consequence of deeper networks being initialized with a smaller effective scale.**
> >
> > We have clarified this point in the revised version at Lines 339–341.
> >
> > ## **Q6. Discrete vs. Continuous data**
> >
> > We would first appreciate a bit of clarification on what is meant by “discrete sampling” versus “continuous data” in this context. Our understanding is that the concern is about the observation pattern in matrix completion (a discrete subset of entries) versus the fact that real neural networks are trained on data drawn from a continuous, connected input space.
> >
> > In our matrix completion setting, the values of the matrix entries are real valued (continuous). What is discrete is the *set of observed indices*: we only see entries on a finite subset $\Omega \subseteq [d] \times [d]$, so “discreteness” refers to which entries are observed. Similarly, in practical neural network training, the underlying data distribution is continuous, but training is still performed on a finite collection of samples. These samples are accessed in a discrete way, so the effective training data are also a discrete subset of the input space. **In our analogy, the observed entries in matrix completion play the role of training data, and the unobserved entries play the role of test data.** The goal in both cases is to learn a representation that captures some low dimensional structure that generalizes from the observed subset to the unobserved region.
> >
> > What is somewhat specific to matrix completion is that the sampling geometry can be expressed explicitly as an observation graph, so notions such as connectedness can be analyzed cleanly. We use this structure to isolate how the connectivity of the observation pattern and the coupling of the factors interact with depth to produce low-rank bias and loss of plasticity. This mechanism does not rely on the values themselves being discrete, and our neural network experiments (Figures 17-28 in Appendix C.1.1 with various optimizers) are intended to show that the same depth-induced low-rank behavior also appears in standard architectures trained with a variety of optimizers.
> >
> > If we have misunderstood your concern about “discreteness” here, we would be grateful for further clarification and would be happy to address it more directly.

---

> > > ### Author Response · Authors · 2025-11-22
> > >
> > > ## **Q7. Relation to Kleinman et al. (2024)**
> > >
> > > Thank you for raising this connection. For Figure 1 in Kleinman et al. (2024), the setup compares two linear network branches that start from different effective initialization scales due to an early training deficit. They observe that, as depth increases, this initial scale mismatch has a larger impact on the final singular values. We view this primarily as an effect of how depth amplifies scale differences between two paths, rather than as a direct measure of how much loss of plasticity a single network experiences. In our setting, we consider a single path rather than multiple branches, and the “deficit” is encoded in the data: the pretraining phase corresponds to training on a partially observed matrix, and the post training phase corresponds to adding new observations.
> > >
> > > If the reviewer instead refers to the matrix completion experiment in Figure 4 of Kleinman et al. (2024), there are two key differences from our setup:
> > >
> > > 1. They plot a ***relative* reconstruction error**, defined by comparing the reconstruction error to that of a network trained directly from random initialization, rather than plotting the absolute error itself.
> > > 2. Their training setup is run **for a fixed number of iterations** without waiting for convergence, so the training loss at the end of training can vary with both depth and phase. In our experiments, by contrast, we terminate all training phases once the loss falls below a fixed threshold.
> > >
> > > We have clarified these differences in the revised main text at Lines 364 to 367.
> > >
> > > Thanks for your time and consideration.
> > >
> > > Best regards,
> > >
> > > Authors

---

### Official Review · Reviewer_TLLi · 2025-10-30

**Soundness:** 3
**Presentation:** 3
**Contribution:** 3
**Rating:** 6
**Confidence:** 3

**Summary:**

The paper uses deep linear matrix factorization as a controlled setting to explain why depth strengthens the implicit bias toward low rank. It identifies coupled vs. decoupled training dynamics as the underlying mechanism: in depth-2, coupling depends on observation-graph connectivity; in depth $\ge3$, coupling arises generically and drives stronger rank-1 attraction. Using diagonal observations and a family of initializations, the authors derive limiting singular-value formulas and connect the mechanism to loss of plasticity (LoP) under warm starts.

**Strengths:**

1. A clear, mechanism-first account that unifies several scattered observations about low-rank bias and depth.

2. Nontrivial analytical traction via diagonal observations and limiting SVD characterizations, with numerics that mirror the theory.

3. Bridges the mechanism to LoP with formal statements (stable-rank lower bounds) rather than just empirical anecdotes.

**Weaknesses:**

1. Scope of formalism: The most rigorous theorems rely on diagonal or highly structured observation patterns and gradient-flow analysis. This leaves a substantial gap to practical regimes, for example finite-step SGD with noise/momentum and unstructured sparsity. Without non-asymptotic bounds in these regimes, it’s unclear how predictive the theory is for typical training runs.

2. Initialization dependence: Several results hinge on a specific initialization family that tunes coupling. While the intuition is compelling, robustness across standard random inits is argued more heuristically than proved. This creates ambiguity about when the rank-1 bias reliably manifests.

3. Transfer to nonlinear/realistic setups: The LoP story is precise for depth-2 linear models and toy expansions of the observation set. Generalization to deep nonlinear networks, noisy completion, and rectangular cases is suggestive but not sealed.

**Questions:**

See weakness

---

> ### Author Response · Authors · 2025-11-22
>
> We appreciate your time and insightful comments. Below, we address the concerns raised.
>
> ## **W1. Gap to Practical Regimes**
>
> Thanks for the insightful comment. We would like to clarify that, except for Theorem 3.3, our main results do not rely on diagonal or other highly structured observation patterns. For example, Theorem 3.1 treats the case where only the first column is observed, Theorem 4.2 considers a setting where both diagonal and off-diagonal entries are observed, and Theorem 4.3 holds without any specific assumptions on the observation pattern or the observed values. We would appreciate it if the reviewer could also consider this broader collection of settings when evaluating our theoretical contributions.
>
> **For Theorem 3.3, we have extended our analysis beyond purely diagonal observations** in the revised version. In particular, as described in the Global Response, we now treat block-diagonal observations, which strictly generalize the original diagonal setting.
>
> Our matrix completion setting is a commonly used testbed for understanding the implicit bias of neural networks, and much of the prior theoretical work similarly analyzes linear networks under gradient flow **without stochastic noise or momentum** [1–6]. In line with this literature, we also study depth-induced low-rank bias in this setting, focusing on how increasing depth shapes the learned low-rank structure.
>
> Moreover, gradient flow is not purely an abstract idealization. For sufficiently small step sizes, gradient flow closely track those of gradient descent, and in our experiments the numerical solutions of the limiting equations are in very good agreement with the actual gradient descent trajectories (see Figures 7 and 8 in Appendix C.1). We also include practical neural network experiments in Figures 17 to 28 in Appendix C.1.1. In architectures such as ResNets and VGG, trained with standard optimizers like SGD with momentum, Adam, and RMSProp, **increasing depth consistently drives the weight matrices toward lower rank structure.**
>
> Furthermore, we provide a fully connected ReLU network experiment using the CIFAR-10 dataset comparing Gaussian (coupled) and identity-based (more “decoupled”) initializations across different depths, which further supports that coupled training dynamics intensify low-rank bias in practical settings (see Figure 29 in Appendix C.1.1).
>
> Analyzing such realistic settings directly is extremely challenging from a theoretical perspective, which is why we rely on the matrix completion task as a controlled surrogate. While we acknowledge that this setting does not capture every aspect of practical neural network training, we believe it provides a suitable and interpretable framework for explaining why deeper models tend to exhibit a stronger low-rank bias.
>
> ## **W2. Initialization dependence**
>
> Thank you for raising this concern. A detailed discussion is provided in the Global Response under “Initialization dependence”; here we briefly summarize our perspective.
>
> From a dynamical systems viewpoint, the training dynamics form a high-dimensional, non-linear system of ODEs, and solving this system for generic random initializations such as Gaussian in order to exactly characterize the limiting solution is highly nontrivial. We therefore work with a structured but still rich family of deterministic initializations that extends the scaled identity ($\alpha \mathbf{I}_d$) initialization widely used in prior work, and that remains analytically tractable while preserving the key coupled versus decoupled behavior. In the revised version, we added a brief remark after the initialization description to clarify this motivation and its relation to Gaussian initialization.

---

> > ### Author Response · Authors · 2025-11-22
> >
> > ## **W3. Transfer to nonlinear/realistic setups**
> >
> > We acknowledge that the depth-2 model we analyze does not fully capture all aspects of practical neural networks. At the same time, we would like to emphasize that our loss of plasticity analysis is not limited to the $2 \times 2$ case in Theorem 4.2; we also provide a full characterization for general $d \times d$ matrices in Theorem 4.3, which goes beyond the toy setting. In the general $d \times d$ case, the observations **may also contain noise**, as long as they satisfy the loss conditions required for our analysis. Even in this simplified matrix completion framework, the training dynamics are governed by a non-linearly coupled system of ODEs, and **carrying out a rigorous analysis of such coupled dynamics is already highly nontrivial.**
> >
> > The loss of plasticity phenomenon has been reported in many empirical studies, but its underlying mechanism is still not well-understood. Much of the existing empirical work interprets loss of plasticity through the lens of “lazy training,” without providing a formal, theoretically grounded mechanism [2, 7, 8]. In contrast, **our analysis in Section 4 provides, to the best of our knowledge, the first theoretically rigorous account of loss of plasticity.** Our results provide a theoretical guarantee that connects lazy training behavior to loss of plasticity, thereby offering insights into why loss of plasticity emerges in practical neural network models.
> >
> > Thanks for your time and consideration.
> >
> > Best regards,
> >
> > Authors
> >
> > ---
> >
> > ### **References**
> >
> > 1. Deep Linear Networks for Matrix Completion-an Infinite Depth Limit, SIAM Journal on Applied Dynamical Systems
> > 2. Critical Learning Periods Emerge Even in Deep Linear Networks, ICLR 2024
> > 3. Implicit Regularization in Deep Matrix Factorization, NeurIPS 2019
> > 4. Implicit Regularization in Deep Learning May Not Be Explainable by Norms, NeurIPS 2020
> > 5. Implicit Regularization in Matrix Factorization, NIPS 2017
> > 6. Implicit Regularization in Matrix Sensing via Mirror Descent, NeurIPS 2021
> > 7. DASH: Warm-Starting Neural Network Training in Stationary Settings without Loss of Plasticity, NeurIPS 2024
> > 8. A study on the plasticity of neural networks, arXiv 2106.00042

---

### Official Review · Reviewer_Qs2f · 2025-10-31

**Soundness:** 3
**Presentation:** 2
**Contribution:** 2
**Rating:** 6
**Confidence:** 2

**Summary:**

Matrix completion by deep linear networks is an important model problem for understanding a variety of phenomena in the training of deep neural networks. It was previously observed empirically that the such models have a tendency to discover low-rank solutions. In the case of two-layer networks, it was observed that this is the case if and only if the observations are "connected" in a suitable sense.

The reviewed work attempts to extend this understanding to the case of deeper networks (which are known to provide low-rank solutions even for disconnected observations). As a starting point, it uses the fact that the characterization of low-rankness through connected observations proceeds by observing that disconnected observations lead to decoupled gradient dynamics of different degrees of freedom.

To this end, the reviewed work definines a notion of "coupled gradient flow dynamics" and shows that under diagonal (and thus disconnected) observations and a structured initialization scheme, the training dynamics are coupled if and only if the initialization is diagonal. It implicitly characterizes the singular values of the converged matrices in this case.

Finally, the authors aim to illucidate the loss of plasticity phenomenon by showing that fast convergence of a pretrained network prevents significant decay of the stable rank.

**Strengths:**

- This work addresses the important and thorny question of implicit bias in deep learning

- Generalizes the independent observations mechanism beyond the 2 layer case

- Provides an potentially useful perspective on the plasticity phenomenon

**Weaknesses:**

Maybe due to the challenging nature of the questions studied, the connection between the highly restricted theoretical setting and the claims about the behavior of the real algorithm (deep learning or even just deep matrix completion) is somewhat tenuous at times. I am especially doubtful about the additional insight of the implicit characterization (8), (9). If it is possible to derive additional insight from it rather than just solving it numerically, this would significantly strengthen the paper.

**Questions:**

Typos:
Line 306 "of of"

1. You solve the equations (8) (9) numerically. Can you show that this system has a unique solution? If not, how do you know that the solutions of this system of equations characterizes the training behavior?

2. On a related note, what is the benefit of solving (8) (9) over directly observing the training trajectories under the (fairly restrictive) conditions where Theorem 3.3 holds?

---

> ### Author Response · Authors · 2025-11-22
>
> We sincerely appreciate your insightful and valuable feedback. We address the given concerns and questions below.
>
> ## **W1. Additional insight of the implicit characterization (8), (9)**
>
> Thank you for raising this concern. We provide a more detailed discussion of this point in the Global Response under “Additional insight from the implicit characterization (8) and (9).” Here we briefly summarize the main idea.
>
> In the revised version, we clarify the additional insight provided by the implicit system (8), (9) by proving **Corollary 3.4** following Theorem 3.3. Under the setting $1 < m < \infty, d \geq 2, w^*>0$, and $L \geq 3$, Corollary 3.4 shows that as $\alpha \to 0$, the stable rank of $\mathbf{W}_{L:1}(\infty)$ converges to 1, while in the decoupled regimes $L=2$ or $m=\infty$, the stable rank remains close to $d$.
>
> By combining Theorem 3.3 with Corollary 3.4, we can summarize the behavior of the stable rank across all regimes considered in Theorem 3.3 and explicitly see how the coupled dynamics governed by (8) and (9) control the limiting stable rank. We have incorporated this discussion into the revised manuscript (Lines 323–333), and we believe that this comparison provides concrete analytic insight into the implicit system (8) and (9) and further strengthens Theorem 3.3.
>
> ## **Q1. Can you show that this system has a unique solution?**
>
> We appreciate the reviewer for bringing this to our attention. We demonstrate that the **implicit system defined in (8) and (9) has a unique solution** for the limiting singular values. Specifically, we provide a proof of uniqueness in **Proposition D.2 (Appendix D.3.4)**. Furthermore, to avoid any potential confusion, we have added a clarifying sentence in the discussion following Theorem 3.3 (Lines 338–339). We hope these updates clarify the uniqueness of the solution.
>
> Briefly, the uniqueness is guaranteed by the strict monotonicity of the scalar functions associated with (8) and (9). If we define
>
> $$
> \begin{align*}
> f_1(\sigma) &= \sigma^{\frac{2-L}{L}}  - \left(\frac{w^* d - \sigma}{d-1}\right)^{\frac{2-L}{L}},  \quad
> f_2(\sigma) = (w^* d - (d-1)\sigma)^{\frac{2-L}{L}} - \sigma^{\frac{2-L}{L}},
> \end{align*}
> $$
>
> then, since the exponent satisfies $(2-L)/L < 0$, we show that $f_1$ is strictly decreasing and $f_2$ is strictly increasing on their respective feasible domains. This strict monotonicity implies that each of the equations in (8) and (9) admits a unique root, and therefore **the limiting singular values solving the system are uniquely determined by our initialization and problem parameters.**
>
> ## **Q2. What is the benefit of solving (8) (9)?**
>
> Thank you for your valuable question. Our goal is to precisely understand *which* solution the training dynamics converge to in the infinite time limit. Direct numerical experiments only provide approximate snapshots at finite times, which do not formally identify the limiting solution or its dependence on $(L,m,d, \alpha, w^*).$
>
> The equations (8) and (9) characterize the limit point of the training dynamics theoretically, and they allow us to prove qualitative statements that go beyond what can be observed from empirical trajectories. For example, we can show that for $L \geq 3$ and finite $m$ the stable rank of the limit solution converges to one as $\alpha \to 0$, whereas in the decoupled regimes ($L=2$ or $m=\infty$) the stable rank remains close to $d$. We therefore view solving (8) and (9) not as a substitute for observing trajectories, **but as a way to analytically identify and compare the limiting solutions across different regimes**, which is difficult to achieve from simulations alone.
>
> Thanks for your time and consideration.
>
> Best regards,
>
> Authors

---

### Author Response · Authors · 2025-11-22

#

We express our gratitude for your time and valuable comments. We would like to address the concerns raised by multiple reviewers and to highlight several new results and clarifications that we have added in the revised version and rebuttal period.

## **Extension of Theorem 3.3 to block-diagonal observations**

Motivated by the insightful comment of Reviewer DsYE, we have extended our Theorem 3.3 beyond purely diagonal observations. During the rebuttal period, **we added Theorem D.3, which treats the case where the model observes block-diagonal entries**. In addition, Figure 33 in Appendix D.5 numerically solves the equations obtained from Theorem D.3 and shows the same qualitative behavior as Figure 2 in the diagonal case: for $L=2$ and for $L \geq 3$ with $m=\infty$, the solution remains high-rank, whereas for $L \geq 3$ with finite $m$ and small initialization, the dynamics converge to a low-rank solution. This result strictly generalizes our earlier diagonal observation setting in Theorem 3.3, since diagonal observations correspond to the special case where each block has size one.

In the block-diagonal setting, if we denote by $n$ the number of diagonal blocks, the singular values $\sigma_j$ for $j \in \\{ n+1, \dots, d\\}$ always converge to zero for any choice of parameters. Intuitively, even if the dynamics are decoupled at the level of the full matrix in the sense of Definition 2, they become coupled when we apply the notion of coupling from Definition 2 to each diagonal block separately, so the training dynamics are coupled within each block.

Therefore, the matrix can still converge to a comparatively low-rank solution even in the decoupled case. Since all rows (or columns) within a block share the same pattern, the row (or column) space is spanned by at most $n$ distinct block-wise patterns, so the overall rank is at most $n$, the number of blocks. **This block-diagonal example therefore further illustrates how coupled versus decoupled dynamics control the strength of the low-rank bias.**

Due to the short discussion period, this new result currently appears only in Appendix D.5, but we plan to replace the original diagonal version of Theorem 3.3 with this block-diagonal formulation in the main text. We also plan to derive and include the corresponding corollary in the main text.

## **Additional insight of the implicit characterization (8), (9)**

In response to several reviewers’ comments on what additional insight the implicit characterization (8) and (9) provides, we have clarified in the revised manuscript how these equations lead to concrete statements about the limiting stable rank. Specifically, we establish **Corollary 3.4** following Theorem 3.3. Under the setting $1 < m < \infty, d \geq 2, w^*>0$, and $L \geq 3$, Corollary 3.4 demonstrates that as the initialization scale approaches zero ($\alpha \to 0$), the stable rank of the limiting product matrix $\mathbf{W}_{L:1}(\infty)$ converges to one.

By combining Theorem 3.3 with Corollary 3.4, we can summarize the behavior of the stable rank across all considered regimes as follows:

- If $L=2$ (**decoupled dynamics**):

    $$
    {\rm srank}\big(\mathbf{W}_{2:1}(\infty)\big) = \frac{\lVert \mathbf{W}\_{2:1}(\infty) \rVert\_F^{2}}{\lVert\mathbf{W}\_{2:1}(\infty)\rVert\_2^{2}}
      = \frac{(m + d - 1)^{4} + (m - 1)^{4}(d - 1)}{(m + d - 1)^{4}}.
    $$

- If $L \geq 3$ and $1 < m < \infty$ (**coupled dynamics):**

    $$
    {\rm srank}\big( \mathbf{W}\_{L:1}(\infty) \big) =   \frac{\lVert \mathbf{W}\_{L:1}(\infty)\rVert \_F^{2}}{\lVert \mathbf{W}\_{L:1}(\infty)\rVert \_2^{2}}
      \to 1 \quad \text{as } \alpha \to 0 .
    $$

- If $L \geq 3$ and $m=\infty$ (**decoupled dynamics**):

    $$
    {\rm srank}\big( \mathbf{W}\_{L:1}(\infty) \big) =  \frac{\lVert \mathbf{W}\_{L:1}(\infty)\rVert \_F^{2}}{\lVert \mathbf{W}\_{L:1}(\infty)\rVert \_2^{2}} = d .
    $$


These results highlight **how coupled dynamics govern the limiting stable rank**. For the shallow network case ($L=2$), if we set a large enough value for $m$ (thereby increasing the initial rank), the stable rank of the product matrix eventually converges to approximately $d$ regardless of $\alpha$, a trend that is also corroborated by Figure 2. Similarly, in the other decoupled regime ($L \geq 3 \text{ and } m =\infty$), the system yields identical singular values, resulting in a stable rank of exactly $d$.

In contrast, in the coupled dynamics regime ($L \geq 3$ with finite $m$), we can force the stable rank of the product matrix to converge to 1 simply by choosing a sufficiently small initialization scale.

We have incorporated this discussion into the revised manuscript (Lines 323-333), and we believe that this comparison provides concrete analytic insight into the implicit system (8) and (9) and further strengthens Theorem 3.3.

---

> ### Author Response · Authors · 2025-11-22
>
> ## **Initialization dependence**
>
> Several reviewers asked how sensitive our results are to the specific deterministic initialization in (7) and how they relate to more realistic random initializations such as Gaussian schemes.
>
> We first note that deterministic initialization is neither trivial nor unrealistic. It has been widely used in prior work, and **many analyses are carried out under specific deterministic choices such as the scaled identity $\alpha \mathbf{I}_d$** [1-4]. The initialization $\alpha \mathbf{I}_d$ is simply one instance within the broader family of initializations we consider in the limit $m \to \infty$. In this sense, our setting cover a much larger class of initial conditions that includes $\alpha \mathbf{I}_d$ as a special case, rather than focusing on a single, highly tuned initialization.
>
> Within this broader family, our goal is to understand the mechanism that produces low-rank bias as depth increases. Under Gaussian initialization, the dynamics are almost surely coupled (Proposition B.1), so one may be tempted to attribute low-rank bias purely to depth. By working with a slightly more structured family of deterministic initializations, we can separate the effect of depth from the effect of the coupling structure. In particular, Theorem 3.3 shows that even for depth $L\geq 3$,  if the dynamics are decoupled, the model converges to a rank-$d$ solution. **This indicates that the dominant factor behind low-rank bias is the coupling structure of the dynamics, rather than depth by itself.**
>
> This perspective also sheds light on why Gaussian initialization often leads to low-rank solutions in practice. For deep networks with $L \geq 3$ with Gaussian initialization, the dynamics are almost surely coupled, so we expect their behavior to be similar to the coupled deterministic initializations that exhibit a strong low-rank bias in our analysis. In this sense, our results provide a theoretical mechanism that is consistent with the empirical observation that Gaussian initialization tends to converge to low-rank solutions in many settings, although a full formal treatment of Gaussian initialization is beyond the scope of this work.
>
> Finally, directly analyzing Gaussian initialization is challenging because the degrees of freedom are very large, which makes it difficult to track each trajectory analytically. We therefore choose a structured yet still rich family of deterministic initializations that is tractable enough to analyze, while still capturing the key coupled versus decoupled behavior that we aim to highlight.
>
> In the revised version, we have added a brief remark immediately after the initialization description to clarify this motivation and its relation to Gaussian initialization.
>
> We hope our response helps to resolve any concerns and confusion.
>
> Best regards,
>
> Authors
>
> ---
>
> ### **References**
>
> 1. Implicit Regularization in Deep Matrix Factorization, NeurIPS 2019
> 2. Implicit Regularization in Deep Learning May Not Be Explainable by Norms, NeurIPS 2020
> 3. Implicit Regularization in Matrix Factorization, NIPS 2017
> 4. Implicit Regularization in Matrix Sensing via Mirror Descent, NeurIPS 2021

---

### Author Response · Authors · 2025-11-28

Dear Reviewers,

Thank you for taking the time to review our work and for providing such thoughtful and constructive feedback. We have thoroughly reviewed all comments and prepared our best effort to address every concern clearly and transparently.

We understand that you may have a busy schedule, but we wanted to follow up to ensure that our responses have sufficiently addressed your concerns. If any questions remain or if there are additional points you would like us to clarify, we would be very happy to continue the discussion.

Thank you again for your expertise, time, and thoughtful suggestions.

Sincerely,

Authors

---

### Meta-Review · Area_Chair_UVM5 · 2025-12-13

**Summary:**

Initial concerns centered on the restricted observation model, reliance on structured initializations, limited analytical insight from implicit equations, optimizer dependence, and the gap between theory and practical regimes. During rebuttal, the authors added significant theoretical extensions, clarified novelty relative to prior work, strengthened analytical results, and provided extensive empirical validations across optimizers, initializations, and neural network architectures.

**Reviewer Concerns:**

Addressed

- Theory was extended
- Implicit equations were strengthened by proofs of uniqueness and convergece result
- The role of deterministic initialization was clarified
- New ablation studies with SGD, momentum, Adam, RMSProp, and Adagrad, as well as CNN/ResNet experiments, demonstrated robustness of the depth-induced low-rank bias beyond gradient flow
- The relationship to Bai et al. (2024) and related literature was clarified
- Theoretical results were expanded beyond toy cases

Remaining

- The most rigorous results still rely on stylized matrix completion settings and linear networks
- Adaptive optimizers are explored experimentally but not fully characterized theoretically

**Reviewer Scores:**

- Reviewer Qs2f: Initially cautious about the implicit equations; likely to maintain or slightly increase their marginally positive score after the added uniqueness proof and analytic corollary.
- Reviewer TLLi: Raised concerns about practicality, initialization, and scope; the block-diagonal extension, optimizer ablations, and clarifications likely support maintaining their acceptance-leaning score.
- Reviewer U2hd: Initially marginally negative due to novelty and rigor concerns; the detailed responses clarifying Theorem 3.1’s contribution, added citations, and analytic strengthening would likely move the score upward.
- Reviewer DsYE: Expressed concerns about generality, optimizers, and robustness; these were directly addressed through new theory and extensive experiments, suggesting an upward revision to at least borderline-positive.

---

### Decision · Program_Chairs · 2026-01-26

Accept (Poster)